# Simulation of a contrail formation and early life cycle for a realistic airliner geometry

Younes Bouhafid[1] and Nicolas Bonne[2]

[1]ONERA, DAAA, Université Paris-Saclay F-91123, Palaiseau, France
[2]ONERA, DMPE, Université Paris-Saclay F-91123, Palaiseau, France

**Correspondence:** Younes Bouhafid (younes.bouhafid@onera.fr)

**Abstract.** Contrails—ice clouds forming in aircraft wakes—may have a radiative impact up to twice that of $CO_2$ emissions from aviation, though significant uncertainties remain. Understanding the entire contrail life cycle, from initial ice crystal formation to potential evolution into persistent cirrus clouds, requires addressing the wide range of spatial and temporal scales involved.

This work presents a novel numerical methodology for simulating contrails from ice crystal formation onset to wingtip vortex dissipation. Unlike conventional methods relying on analytical initialization using Lamb-Oseen vortex pairs, our approach couples Reynolds-averaged Navier–Stokes (RANS) with Large Eddy Simulation (LES) and synthetic turbulence techniques. This enables more accurate capture of near-field effects and detailed consideration of how aircraft geometry influences aerodynamic wake and contrail evolution.

Applied to realistic aircraft geometry under standard atmospheric conditions, our methodology revealed that horizontal tailplane vortices trigger short-wavelength instabilities in the main wingtip vortices, significantly modifying secondary wake structure. Compared to conventional methods, contrails generated through our methodology are wider with larger cross-sectional areas in the first few minutes following ice crystals formation. Previous studies showed contrail spatial dimensions significantly affect contrail-cirrus properties; for instance, under identical initial ice crystal number and mass, an A380 contrail-

cirrus exhibits 20% greater total extinction than a CRJ200 due to different contrail sizes.

We propose a modified conventional method incorporating a quadripolar wake—two main wingtip vortices plus two secondary horizontal tailplane vortices—that more closely matches our methodology's simulations, which more precisely account for near-field aerodynamic effects.

## 1   Introduction

Contrails, short for condensation trails, are the long ice clouds visible in the wake of airplanes on clear days. Those anthropogenic clouds result from the condensation of the water vapor located in the gas mixture between exhaust gases emitted from aircraft engines and cold ambient air. In ice-supersaturated regions of the troposphere, contrails can evolve into cirrus clouds and persist for several hours over hundreds of kilometers before dissipating, thereby increasing regional cloudiness in the meantime. Like any cloud, cirrus clouds induced by contrails interact with both solar and terrestrial infrared radiation, thus

impacting Earth's radiative balance. The effect of such artificial clouds on Earth's radiative balance has been quantified in a recent compilation of studies Lee et al. (2021). This work provided an estimation of the Effective Radiative Forcing (ERF) of contrails and other forcing agents such as $CO_2$, $NO_x$, aerosols, and water vapor. The ERF of cirrus clouds induced by contrails has been estimated at $57.4 \, \mathrm{mW \cdot m^{-2}}$ which is almost twice as much as the ERF of $CO_2$, estimated at $34.3 \, \mathrm{mW \cdot m^{-2}}$.

However, the estimated value of contrails' ERF is currently debated due to the high uncertainty surrounding this value. This uncertainty primarily reflects the difficulty in accurately estimating the optical properties of contrails, which depend on their microphysical properties (ice crystal concentration, crystal size distribution, etc.), as well as their vertical and horizontal extent. Moreover, the microphysical properties of a contrail several hours old are strongly influenced by these same microphysical properties in the initial moments (see Lewellen and Lewellen (2001); Unterstrasser and Görsch (2014)). During these initial moments, that is, within the first minutes of its lifecycle, the evolution of contrails results from the interaction between wake aerodynamics, plume chemistry, and microphysical processes. This interaction and the underlying physics must be correctly understood to accurately predict contrails' radiative impact.

The formation and evolution of contrails can be divided into four successive regimes (see Gerz et al. (1998); Paoli et al. (2013); Misaka et al. (2015); Jacquin and Garnier (1996)), each governed by specific physical phenomena. The *jet regime*, lasting only a few seconds (see Jacquin and Garnier (1996)), is dominated by strong turbulent diffusion in the engine jet while the wing's vortex sheet rolls into a pair of counter-rotating wingtip vortices. As the jet plume cools down, ice crystals begin to form, making the contrail visible at about one to two wingspans behind the aircraft (see Khou et al. (2015); Ramsay et al. (2024)). In the case of soot-rich emissions, that is to say for a soot emission index greater than $5 \times 10^{14} \, \mathrm{kg^{-1}}$ as stated in Yu et al. (2024), ice crystals primarily form through the condensation of water vapor onto soot particles. Conversely, if soot concentration is low in the plume, ice crystals predominantly form on volatile particles that have formed earlier in the plume, such as sulfuric acid droplets (see Kärcher (1998)). Interaction between the jet and vortices becomes significant around ten wingspans for a typical airliner and can be quantified by looking at the $R_3$ ratio defined in Jacquin and Garnier (1996).

The *vortex regime* begins when the plume's evolution is dominated by the action of wake vortices, action characterized by the plume roll-up around the wingtip vortices. As time passes, the vortex pair descends due to mutual induction, carrying a significant fraction of the ice crystals downward. This descent results in an increase of saturation pressure as a result of the adiabatic heating of the surrounding air, potentially leading to ice crystal sublimation. However, the vortex pair interacts with atmospheric stratification, generating baroclinic vorticity and a secondary wake that redistributes some ice crystals back to flight altitude and beyond as observed in Unterstrasser (2014) and where ambient conditions are more favorable for crystals growth. After several tens of seconds, Crow instability, which was theorized in Crow (1970); Crow and Bate Jr (1976), deforms the vortex pair, marking the transition to the dissipation regime. This regime, lasting a few minutes, ends with the destruction of the vortices and the formation of a chain of vortex rings periodically distributed along the flight direction and that can often be observed in the sky on clear days. This marks the beginning of the *diffusion regime*. It is important to note that under strong atmospheric stratifications, short-wave vortex instabilities may dominate and destroy the vortices before Crow instability develops, preventing the formation of vortex rings. During the *diffusion regime*, contrail evolution is governed by a certain number of physical processes which are affected by atmospheric conditions such as atmospheric turbulence, wind

shear, and stratification. If the air remains supersaturated with respect to ice, the contrail may transform into cirrus clouds, potentially persisting for hours. This happens in the so-called ice super-saturated regions (ISSR) that can frequently be found in the tropopause. A significant part of the *diffusion regime* was previously simulated by Paoli et al. (2017) through a 3D temporal LES approach, by Lewellen (2014) with a Quasi-3D approach and by Unterstrasser (2014) through a 2D approach.

Simulating a contrail over its entire life-cycle with a satisfactory level of fidelity presents a significant challenge. This is primarily due to the wide range of physical phenomena and scales involved in the four regimes making up the life of a contrail. Directly simulating all the scales of a contrail while accounting for its full spatiotemporal development is currently prohibitive in terms of computational cost. Nevertheless, understanding the formation and the evolution of a contrail is crucial to developing mitigation techniques, such as the use of sustainable aviation fuels (SAF) or hydrogen fuel that leads to a significant reduction of nucleation sites (see Voigt et al. (2021)). A commonly adopted solution in the literature consists of performing so-called temporal LES for which periodic boundary conditions are prescribed perpendicularly to the flight direction. The temporal development of a contrail in the fixed ground reference frame can then be computed. Such an approach is valid at a distance from the airplane when jet/vortex interaction becomes significant. The problem that arises then is how to initialize such temporal simulations. In most studies in the literature, a proposed solution is to initialize the wake with analytical solutions modeling a pair of vortices and jet plumes. For instance, the vortices are often defined using the Lamb-Oseen solution while turbulence inside the jet is randomly generated with a normal distribution as in Holzäpfel et al. (2001); Paoli et al. (2013); Unterstrasser (2014); Unterstrasser and Görsch (2014); Lewellen and Lewellen (2001). Such an approach simplifies the aircraft wake by idealizing it. Consequently, some aspects of the near-field aerodynamics are neglected, and their potential effect on the far-field is not taken into account. The hypothesis that the effects of the near-field on the far-field can be neglected must be examined. This is all the more important given the increase in geometrical complexity of future aircraft concepts, such as blended wing bodies and rhomboidal wings.

Recently, a methodology has been developed in Bouhafid et al. (2024) in order to simulate the evolution of an aircraft wake from vortex formation, up to their dissipation several minutes later. This methodology was successively used to bridge the gap between the near-field aerodynamics of the jet regime and the far-field aerodynamics of the vortex and dissipation regimes without using analytical solution initialization for the temporal LES of the wake. This allowed for the full effects of the aircraft geometry on the wake aerodynamics to be taken into account. The same study also compared the wake dynamic when their methodology is used compared to the usual analytical initializations. This comparison has shown significant impact of the horizontal tailplane vortices on the wake evolution and its interaction with the plume.

Therefore, the objective of this work is to apply the methodology developed in Bouhafid et al. (2024) to simulate a contrail from the formation of the first ice crystals to the end of the dissipation regime and to compare the results with the usual analytical initialization in order to see how the differences observed in wake dynamic impact the properties of the contrail in its early life, that is to say before contrail-to-cirrus transition. In addition, the influence of horizontal tailplane vortices on contrails is investigated.

The outline of this paper is as follows. Section 2 describes the methodology and the corresponding numerical framework for simulating a contrail from the jet regime to the dissipation regime. Section 3 gives details about the different simulation

 setups and the corresponding numerical parameters. Section 4 presents the results of the methodology applied to the NASA Common Research Model (CRM) geometry in cruise flight conditions, geometry representative of Boeing B777 airliner. An examination is conducted regarding the influence of the far-field initialization strategy. Finally, Section 5 concludes this work and highlights its different perspectives.

## 2 Methodology and numerical framework

### 2.1 Governing equations

Classical compressible Reynolds-averaged Navier-Stokes (RANS) and compressible Large Eddy Simulation (LES) equations are solved for a multi-species gas mixture using ONERA in-house finite volumes solver CHARME as detailed in Refloch et al. (2011). The RANS and LES equations are detailed below.

#### 2.1.1 Reynolds-averaged Navier-Stokes (RANS) equations

Compressible Reynolds-averaged Navier-Stokes (RANS) equations are solved for a multi-species gas mixture using ONERA in-house finite volumes solver CHARME. Using Einstein summation convention, the equations are:

$$\frac{\partial}{\partial t}\left(\overline{\rho}\widetilde{Y_\alpha}\right) + \frac{\partial}{\partial x_i}\left(\overline{\rho}\widetilde{Y_\alpha}\widetilde{u_i}\right) = \frac{\partial}{\partial x_i}\left(D_\alpha\overline{\rho}\frac{\partial\widetilde{Y_\alpha}}{\partial x_i} - \overline{\rho}\widetilde{u_i''Y_\alpha''}\right) + \overline{\dot{\omega}_\alpha}$$

$$\frac{\partial}{\partial t}\left(\overline{\rho}\widetilde{u_i}\right) + \frac{\partial}{\partial x_j}\left[\overline{\rho}\widetilde{u_i}\widetilde{u_j} + \overline{p}\delta_{ij} + \overline{\rho}\widetilde{u_i''u_j''} - \widetilde{\tau_{ij}}\right] = \overline{\rho f_i} \tag{1}$$

$$\frac{\partial}{\partial t}\left(\overline{\rho}\widetilde{E}\right) + \frac{\partial}{\partial x_j}\left[\overline{\rho}\widetilde{u_j}\widetilde{E} + \widetilde{u_j}\overline{p} + c_p\overline{\rho}\widetilde{u_j''T} + \widetilde{u_i}\overline{\rho}\widetilde{u_i''u_j''} - K_T\frac{\partial\widetilde{T}}{\partial x_j} - D_\alpha\overline{\rho}\widetilde{h_\alpha}\frac{\partial\widetilde{Y_\alpha}}{\partial x_j} - \widetilde{u_i}\widetilde{\tau_{ij}}\right] = \overline{\rho f_i u_i}$$

with:

$$\widetilde{\tau_{ij}} = \mu\left(\frac{\partial\widetilde{u_i}}{\partial x_j} + \frac{\partial\widetilde{u_j}}{\partial x_i} - \frac{2}{3}\delta_{ij}\frac{\partial\widetilde{u_k}}{\partial x_k}\right) \tag{2}$$

The ideal gas equation is added to the system:

$$\overline{p} = R\overline{\rho}\widetilde{T} \tag{3}$$

$\overline{\Phi}$, $\widetilde{\Phi}$ and $\Phi''$ are respectively the Reynolds average, Favre average and Favre fluctuation of variable $\Phi$. Fourier's law is used for heat diffusion while Fick's law is used for matter diffusion. In these equations, $\rho$ represents the mass density of the multi-species mixture, $u_i$ the velocity vector, and $E = e + \frac{1}{2}u_i^2$ the total energy, with $e$ being the internal energy. $p$ is the static pressure, $\tau_{ij}$ the viscous stress tensor, $T$ is the static temperature, and $f_i$ the external body forces acting on the fluid (e.g., gravity). $Y_\alpha$ is the mass fraction of species $E_\alpha$. $\dot{\omega}_\alpha$ is the source or sink term for species $E_\alpha$. $-\widetilde{u_i''u_j''}$ is the so-called Reynolds stress tensor. $\mu$

is the dynamic viscosity, $c_p$ the heat capacity, $K_T$ the thermal conductivity coefficient, $D_\alpha$ the diffusion coefficient of species $E_\alpha$, and $R$ the specific gas constant of the mixture.

Closing the RANS equations requires a turbulence model to compute the correlation terms. The turbulence model used in this work is the DRSM (Differential Reynolds Stress Modeling) ATAAC model (see Schwamborn and Strelets (2012)) where a transport equation is solved for each of the six Reynolds stresses, with a modelization of the pressure-strain correlation term based on Wilcox's work (see Wilcox et al. (1998)). Two additional equations are solved: one for the specific dissipation $\omega$ and another elliptic equation to model near-wall effects on turbulence. Unlike most two-equations turbulence models, DRSM models do not make the Boussinesq hypothesis, which makes them more accurate for solving strong vortical flows (see Churchfield and Blaisdell (2009)). The remaining turbulent diffusion fluxes are modeled using the gradient hypothesis:

$$\widetilde{\overline{\rho u_i'' Y_\alpha''}} = -\frac{\mu_t}{\mathrm{Sc}_t} \frac{\partial \widetilde{Y_\alpha}}{\partial x_i} \tag{4}$$

$$\widetilde{\overline{\rho u_j'' T}} = -\frac{\mu_t c_p}{\mathrm{Pr}_t} \frac{\partial \widetilde{T}}{\partial x_i} \tag{5}$$

with $\mathrm{Sc}_t$ et $\mathrm{Pr}_t$ respectively the turbulent Schmidt number and turbulent Prandtl number, which are commonly set to 0.9. The turbulent viscosity $\mu_t$ is set to $\mu_t = \frac{\rho k}{\omega}$ with $k$ being the turbulent kinetic energy deduced from the Reynolds tensor.

### 2.1.2 Large Eddy Simulation (LES) equations

Compressible Large Eddy Simulation (LES) equations are solved for a multi-species gas mixture using ONERA in-house finite volumes solver CHARME. Using Einstein summation convention, the equations are:

$$\begin{aligned}
&\frac{\partial}{\partial t}\left(\overline{\rho}\widetilde{u}_j\right) + \frac{\partial}{\partial x_i}\left(\overline{\rho}\widetilde{u}_i\widetilde{u}_j\right) + \frac{\partial}{\partial x_i}\left[\overline{\rho}\left(\widetilde{u_i u_j} - \widetilde{u}_i\widetilde{u}_j\right)\right] = -\frac{\partial \overline{p}}{\partial x_j} + \frac{\partial \overline{\tau_{ij}}}{\partial x_i} + \overline{\rho f_i} \\
&\frac{\partial}{\partial t}\left(\overline{\rho}\widetilde{Y}_\alpha\right) + \frac{\partial}{\partial x_i}\left(\overline{\rho}\widetilde{u}_i\widetilde{Y}_\alpha\right) + \frac{\partial}{\partial x_i}\left[\overline{\rho}\left(\widetilde{u_i Y_\alpha} - \widetilde{u}_i\widetilde{Y}_\alpha\right)\right] = \frac{\partial}{\partial x_i}\left(D_\alpha\overline{\rho}\frac{\partial \widetilde{Y}_\alpha}{\partial x_i}\right) + \overline{\dot{\omega}_k} \\
&\frac{\partial \overline{\rho}\widetilde{h}_s}{\partial t} + \frac{\partial}{\partial x_i}\left(\overline{\rho}u_i\widetilde{h}_s\right) = \frac{\overline{Dp}}{Dt} + \frac{\partial}{\partial x_i}\left[K_T\frac{\overline{\partial T}}{\partial x_i} - \overline{\rho}\left(\widetilde{u_i h_s} - \widetilde{u}_i\widetilde{h}_s\right)\right] + \overline{\tau_{ij}\frac{\partial u_i}{\partial x_j}} \\
&\quad - \frac{\partial}{\partial x_i}\left(\overline{\rho\sum_{k=1}^{N} D_k\frac{\partial Y_k}{\partial x_i}h_{s,k}}\right) + \overline{\dot{\omega}_T} + \overline{\rho f_i u_i}
\end{aligned} \tag{6}$$

with:

$$\frac{\overline{Dp}}{Dt} = \frac{\partial \overline{p}}{\partial t} + \overline{u_i\frac{\partial p}{\partial x_i}} \tag{7}$$

$h_s$ is the specific enthalpy of the mixture and $h_{s,k}$ the specific enthalpy of species $E_k$. $\dot{\omega}_T$ is the heat production rate due to the production/destruction of species. This time, $\overline{\Phi}$ represents standard LES filtering of variable $\Phi$ with $\widetilde{\Phi} = \frac{\overline{\rho \Phi}}{\overline{\rho}}$. The subgrid terms, such as $\overline{\rho} \left( \widetilde{u_i u_j} - \widetilde{u}_i \widetilde{u}_j \right)$, are modeled using the Smagorinsky model. Species turbulent diffusion and enthalpy turbulent diffusion are modeled through the commonly used gradient hypothesis. For example, we have $\widetilde{u_i h_s} - \widetilde{u}_i \widetilde{h}_s = -\frac{\nu_t}{Sc_t} \frac{\partial h_s}{\partial x_i}$ with $\nu_t$ and $Sc_t$ being respectively the eddy viscosity and the turbulent Schmidt number. In the context of contrail simulation, heat production is due to latent heat release caused by water condensation or sublimation and can be neglected before aerodynamic phenomena when it comes to simulating the first minutes of contrails, as stated by Lewellen and Lewellen (2001) and Unterstrasser (2014). However, it is important to remember that latent heat release cannot be neglected for older contrails in the diffusion regime (see Paoli et al. (2017)), which are beyond the scope of this work.

For both RANS and LES equations, convective fluxes are computed using the HLLC scheme with second-order accuracy. A second-order scheme is used for the computation of the diffusive fluxes. Time integration is performed using Backward Euler method for RANS calculations and Implicit Second-Order Runge Kutta for the LES. The GMRES method is used to solve the linear system obtained after the finite volumes discretization of the RANS/LES equations.

## 2.2 Microphysical model

The microphysical model used for the formation of contrail ice crystals is a simplified version of the bulk model developed by Jean-Charles Khou et al. (see Khou (2016); Khou et al. (2015); Montreuil et al. (2018)). More precisely, the exhaust plume is modeled through a multi-species gas mixture approach and is composed of seven species: dry air, water vapor $H_2O$ (gas phase $H_2O$), ice $H_2O_s$ (ice phase $H_2O$), sulfur trioxide $SO_3$, sulfuric acid $H_2SO_4$, sulfur trioxide in the adsorbed state $SO_{3,\mathrm{ads}}$ and sulfuric acid in the adsorbed state $H_2SO_{4,\mathrm{ads}}$. A transport equation is solved for the mass fraction $Y_i$ of each species (see Eq.1 and Eq.6). Moreover, ice crystal formation is modeled through water vapor condensation done exclusively on soot particles previously activated by sulfur trioxide ($SO_3$) and sulfuric acid ($H_2SO_4$) molecules. In reality, water vapor first turns into liquid water before freezing to ice. However, freezing is not modeled, implying that any liquid water formed on the surface of a soot turns instantly into ice. The approach used by this model is purely Eulerian and as a result, the quantities associated with soot particles, ice crystals, and sulfur species are treated as fields whose values depend on spatial coordinates and time. In the complete version of Khou's model (see Khou (2016)), the formation chemistry of $SO_3$ and $H_2SO_4$ molecules is considered through a reaction scheme involving 23 species and 60 reactions. This reaction scheme is not included in this work to reduce computational costs. Instead, it is assumed that $SO_3$ and $H_2SO_4$ molecules form very quickly after the engine exhaust gases are emitted, allowing their mass fractions to be directly prescribed in the engine outlet boundary conditions. This assumption is supported by contrail simulations for similar cases using the complete model performed at ONERA.

Mathematically, the microphysical model first solves a transport equation for the soot density number $N_s$:

$$\frac{\partial}{\partial t}\left(\rho N_s\right) + \frac{\partial}{\partial x_i}\left(\rho v_i N_s\right) = \frac{\partial}{\partial x_i}\left(\rho D_{\mathrm{diff}} \frac{\partial N_s}{\partial x_i}\right) \tag{8}$$

Thus, the equation for $N_s$ is the equation of a passive scalar that does not influence flow properties (velocity, pressure, and temperature). $D_{\mathrm{diff}}$ is the diffusion coefficient of soot particles for which the corresponding Schmidt number Sc with respect to air kinematic viscosity is set to 1. This equation can easily be averaged or filtered. The turbulent Schmidt number $\mathrm{Sc}_t$ for the corresponding RANS and LES equation is set to 0.9. The corresponding LES equation is closed with Smagorinsky model and gradient hypothesis.

Soot activation occurs solely through the adsorption of $SO_3$ and $H_2SO_4$ molecules. Two new species are introduced: $SO_{3,\mathrm{ads}}$ and $H_2SO_{4,\mathrm{ads}}$, with their respective mass fractions denoted as $Y_{SO_{3,\mathrm{ads}}}$ and $Y_{H_2SO_{4,\mathrm{ads}}}$. These species represent $SO_3$ and $H_2SO_4$ molecules in the "adsorbed" state, as opposed to their "free" state. The transition from "free" to "adsorbed" states occurs through the reactions $SO_3 \rightarrow SO_{3,ads}$ and $H_2SO_4 \rightarrow H_2SO_{4,ads}$

We define the activated surface fraction $\theta_{\mathrm{act}}$ of a soot particle. This quantity is equal to the ratio of the hydrophilic surface area to the total surface area of a given soot particle. A mass conservation equation is solved for the adsorbed species $SO_{3,\mathrm{ads}}$ and $H_2SO_{4,\mathrm{ads}}$. The corresponding source term $\dot{\omega}_{i,\mathrm{ads}}$ used in Eq.1 is given by:

$$\dot{\omega}_{i,ads} = \frac{\pi D_s^2 \alpha_{ads} \rho Y_i N_s v_{th,i}}{4 M_i} \left(1 - \theta_{\mathrm{act}}\right) \tag{9}$$

where $i = \{H_2SO_4, SO_3\}$, $M_i$ the molar mass of species $i$, $v_{\mathrm{th},i} = \sqrt{\frac{8 k_b T}{\pi m_{\mathrm{mol},i}}}$ the thermal velocity of species $i$, $m_{\mathrm{mol},i}$ the molecular mass of species $i$, $D_s$ is the diameter of a spherical dry soot particle, and $\alpha_{\mathrm{ads}}$ is the sticking probability, equal to 1 if $T \leq 420$ K and 0 otherwise. $D_s$ is set to 54 nm in all the calculations performed in this work. This value is in the order of magnitude of experimental measurements of soot particles diameter of a turbofan engine (see Petzold and Schröder (1998)). Note that desorption is not accounted for, meaning that adsorbed molecules cannot leave the soot surface. To ensure mass conservation, the source term $-\dot{\omega}_{i,ads}$ is added to $H_2SO_4$ and $SO_3$ mass conservation equations.

The fraction $\theta_{\mathrm{act}}$ is defined as:

$$\theta_{\mathrm{act}} = \sum_{i,ads} \frac{\mathrm{Na} \rho Y_{i,ads}}{\pi D_s^2 N_s \sigma_0 M_i} \tag{10}$$

where Na is Avogadro constant and $\sigma_0 = \frac{4}{\pi (D_{\mathrm{H2SO4}})^2} \approx 5 \times 10^{18} \mathrm{\ m}^{-2}$ is the surface density of active sites, based on the molecular diameter of $H_2SO_4$.

Contrail ice is modeled by introducing a new species, $H_2O_s$, representing water in the solid phase formed through heterogeneous nucleation of water vapor on activated soot. More precisely, activated soot particles serve as condensation nuclei. Water vapor diffuses onto these particles and forms an ice cap if the ambient air is supersaturated with respect to liquid water. As stated above, the model assumes that freezing happens instantly once liquid water forms on the soot particle. Currently, the microphysical model makes the simplifying assumption that all the ice spreads evenly over the soot particle surface, creating spherical ice crystals. These newly formed ice crystals then continue to grow by condensation as long as the relative humidity with respect to ice exceeds 100 %. The transformation of water vapor to ice and vice-versa is modeled through the reaction $H_2O \leftrightarrow H_2O_s$. The source term $\dot{\omega}_{\mathrm{ice}}$, representing the mass production rate of ice crystals due to water vapor deposition around

soot particles, is given by:

$$\dot{\omega}_{\text{ice}} = D_{\text{con}} \frac{4\pi r_p \theta_{\text{act}} N_s D_{\text{H}_2\text{O}} G_\alpha (r_p) \left( p_v - p_{\text{sol}}^{\text{sat},S} (r_p, T) \right)}{R_{\text{H}_2\text{O}} T} \tag{11}$$

$$D_{\text{con}} = \begin{cases} 1 & \text{if } p_v > p_{liq}^{\text{sat},S} (r_p, T) \text{ or } r_p > r_s \\ 0 & \text{else} \end{cases} \tag{12}$$

$$p^{sat,S} (r_p, T) = \exp \left( \frac{2\sigma M_{\text{H}_2\text{O}}}{R \rho_{\text{H}_2\text{O}} r_p T} \right) p^{\text{sat}} (T) \tag{13}$$

$p_v$ is the partial pressure of water vapor, $T$ is the static temperature, $D_{\text{H2O}}$ is the diffusion coefficient of water vapor, $\sigma$ is the surface tension coefficient of liquid water, $R_{\text{H}_2\text{O}}$ is the specific gas constant for water vapor, and $\rho_{\text{H}_2\text{O}}$ is the density of water in the liquid phase. $r_p$ is the mean radius of ice crystals, locally defined as:

$$r_p^3 = \frac{3}{4\pi} \frac{\rho Y_{\text{H}_2\text{O}_s}}{\rho_{\text{sol}} N_s} + r_s^3 \tag{14}$$

where $\rho$ is the density of the gas mixture, $\rho_{\text{sol}}$ is the density of ice—not to be confused with $\rho_{\text{H}_2\text{O}s} = \rho Y_{\text{H}_2\text{O}_s}$ the density of the H$_2$O$_s$ species in the gas mixture which corresponds to ice water content—, and $r_s = \frac{1}{2} D_s$ is the radius of a dry soot particle. The saturation vapor pressures with respect to liquid water and ice are given by the semi-empirical formulas from Murphy and Koop (2005) and the density $\rho_{\text{sol}}$ is given by Tabazadeh et al. (2000). Therefore, in this model, every particle with $r_p > r_s$ can be considered to be an ice crystal as it necessarily implies that $Y_{\text{H}_2\text{O}_s} > 0$. Conversely, $r_p = r_s$ means that the particle is a dry soot particle.

$G_\alpha$ is a dimensionless function used to account for cases where the continuum assumption is no longer valid in the immediate vicinity of an ice crystal:

$$G_\alpha(r_p) = \left( \frac{1}{1 + \text{Kn}} + \frac{4\text{Kn}}{3} \right)^{-1} \tag{15}$$

where $\text{Kn} = \frac{\lambda_{\text{air}}}{r_p}$ is the Knudsen number, and $\lambda_{\text{air}}$ is the mean free path of an air molecule in the plume.

Several remarks can be made regarding the expression for $\dot{\omega}_{\text{ice}}$. First, the introduction of the multiplier $D_{\text{con}}$ prevents water vapor condensation on dry soot when the air is supersaturated with respect to ice but not with respect to liquid water. However, if ice has already formed around the soot, the nucleation process can continue even if the air is no longer saturated with respect to liquid water. Additionally, the sign of the term $p_v - p_{\text{sol}}^{\text{sat},S}$ indicates whether the air is supersaturated with respect to ice or not. In the first case, the source term is positive, and the crystals grow. In the second case, the source term is negative, and the

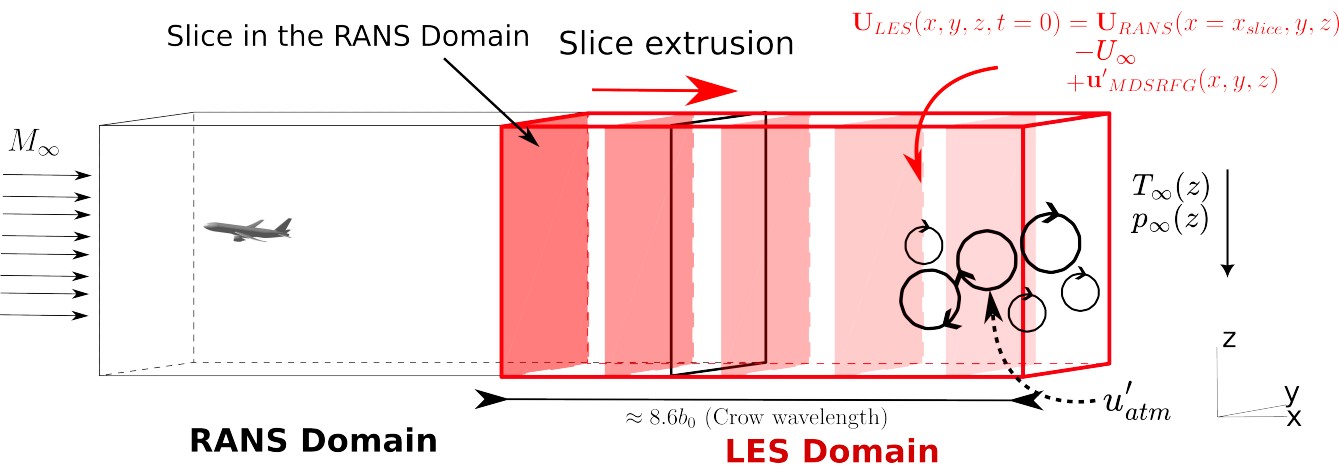

**Figure 1.** Diagram of the initialization methodology of a far-field calculation using a near-field calculation. ($M_\infty$, $U_\infty$, $T_\infty$ and $p_\infty$ are respectively flight Mach number, flight speed, ambient static temperature and ambient static pressure).

ice crystals sublimate, returning to water vapor. The conservation of the total mass of water is thus ensured by adding the term
$-\dot\omega_{\text{ice}}$ to the source term of the $H_2O$ species.

### 2.3    Bridging the gap between the near-field and the far-field

The methodology used to simulate the evolution of a contrail from the formation of the first ice crystals in the near-field to the wingtip vortices destruction in the far-field is the methodology described and validated in the work of Bouhafid et al. (2024). This methodology was used to simulate the CRM geometry aerodynamic wake from the jet regime to the dissipation regime.
A microphysical model, described above, is now added to take into account contrail formation and evolution. A diagram of the methodology is shown in Fig.1. The main ideas of the methodology are as follows. A RANS simulation of the jet regime of a realistic airliner geometry is carried out in the airplane reference frame. Anisotropic mesh adaptation techniques are used to ensure a sufficient resolution of the flow and conservation of wake circulation up to twenty wingspans $b$ behind the airplane. Next, a slice perpendicular to the flight direction is extracted from the RANS simulation. This slice is located far enough behind
the airliner that jet/vortex interaction becomes important, and axial gradients become negligible in comparison to tangential gradients. The slice and its corresponding flow field are then extruded following the flight direction with an extrusion distance equal to one Crow instability wavelength $\lambda_{\text{Crow}} = 8.6 \times \frac{\pi}{4}b$. This extrusion step generates the LES domain for which periodic boundary conditions are prescribed for the boundaries perpendicular to the flight direction. Moreover, the LES computation is now performed in the ground reference frame instead of the aircraft reference frame. Physically speaking, the temporal LES
simulation is equivalent to a certain extent to observing the evolution of a contrail portion located inside a region of the sky after an airliner went through it.

In order to transfer the information from RANS turbulence-related fields (Reynolds tensor, dissipation rate) to the LES domain, synthetic turbulence fluctuations noted $u'_{\text{MDSRFG}}$ are generated using the MDSRFG method developed in Castro and

Paz (2013) and added inside the plume, which is defined for a given position $x$ by the isosurface $U_x > \frac{1}{2}(U_j + U_\infty)$ with $U_j$ the maximum axial velocity in the plume and $U_\infty$ the flight speed. Another synthetic turbulence fluctuation field $u'_{\text{atm}}$ is generated in the entirety of the LES domain using the MDSRFG method. This fluctuation field aims to model atmospheric turbulence responsible for the vortices destruction. The mathematical formulation of the synthetic turbulence is fully described in Bouhafid et al. (2024). Finally, stratification is taken into account by including the effect of gravity in the LES and allowing both ambient static temperature and ambient static pressure to vary with altitude $z$ for a given Brunt-Väisälä frequency $N_b$.

## 3 Computational domain and numerical parameters

The methodology described in the previous section is now applied to the NASA Common Research Model geometry to which jet engines were added. First, a RANS calculation of the near-field is carried out and the results are discussed in Section 4.1. Afterward, the RANS calculation is extended to the far-field in Section 4.2 using a temporal LES according to the methodology developed in this work. The RANS and LES computational domains are described below.

### 3.1 RANS domain

The RANS domain is the same as the one in Bouhafid et al. (2024) and consists of the B777-like CRM geometry inside a box. In terms of design parameters for the CRM geometry, wingspan $b$ is 58.8 m, wing area $S$ is 383 m$^2$, and mean chord $c_m$ is 6.5 m. The box dimensions are $L_x \times L_y \times L_z$ with $L_x = 30b$, $L_y = 20b$ and $L_z = 20b$ as illustrated in Fig.2.

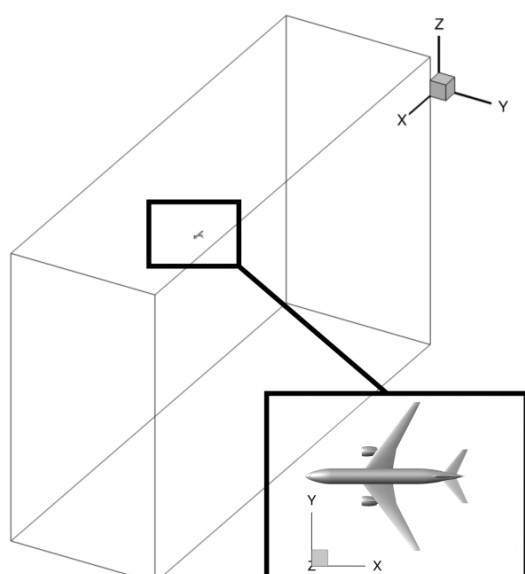

**Figure 2.** RANS computational domain with the CRM geometry. Only half of the geometry is taken into account in the computational domain.

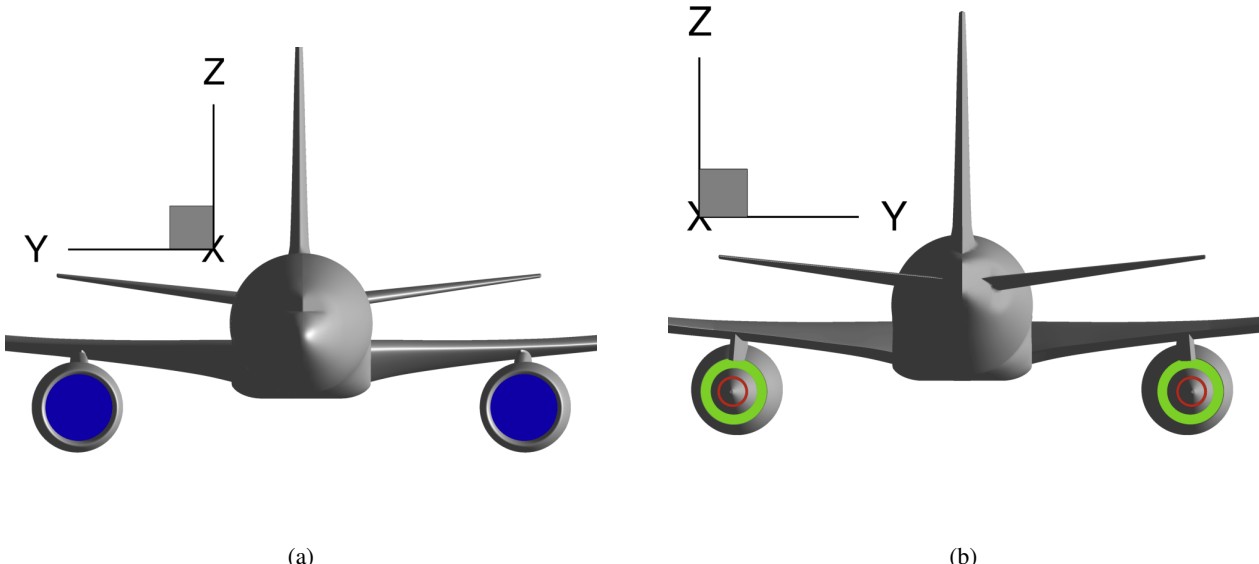

(a)                                           (b)

**Figure 3.** CRM geometry with jet engines. (a) Engine inlet. (b) Core engine outlet & Bypass engine outlet

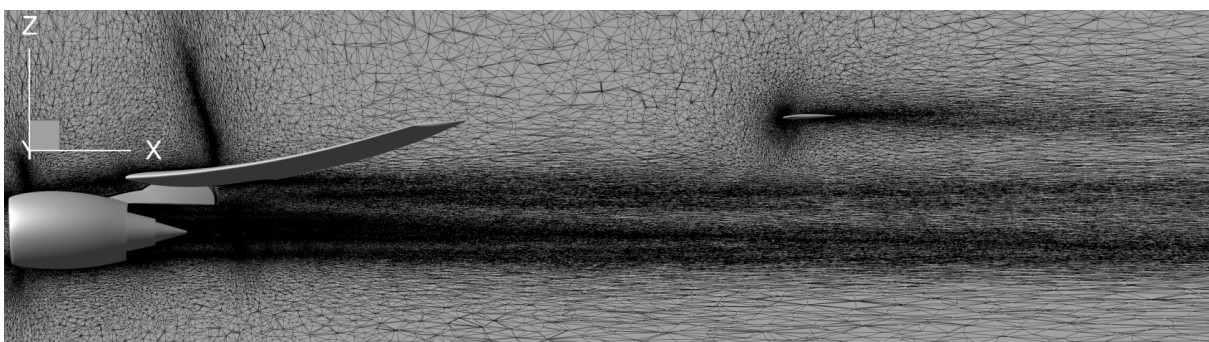

**Figure 4.** Mesh slice $y = 0.17b$

The distance behind the aircraft in the RANS domain is equal to $20b$. Only half of the aircraft is taken into account in the simulation as the CRM geometry is cut in half by the symmetry plane $y = 0$. This is a common hypothesis justified by the symmetry of the mean flow with respect to the plane $y = 0$.

The contrail is studied for cruise flight conditions with a flight Mach number $M_\infty = 0.85$ and an angle of attack $\alpha = 3°$ (see Tab.1). The ambient static pressure $p_\infty$ and ambient static temperature $T_\infty$ are typical for an altitude of 11 km inside the troposphere. The effect of stratification can be neglected in the jet regime and is thus not taken into account in the RANS calculation. Ambient water vapor mass fraction was chosen to impose $RH_{\text{ice}} = 109\ \%$ and $RH_{\text{liq}} = 68\ \%$ (see Tab.2), with $RH_{\text{ice}}$ and $RH_{\text{liq}}$ being respectively the ambient relative humidity with respect to ice and the ambient relative humidity with respect to liquid water.

Turbofan engines were added to the CRM geometry by ONERA, as seen in Fig.3. Their position relative to the wing-tip is $b_j/b = 34$ %. The engine inlet and engine outlet are modeled with inlet/outlet boundary conditions (outlet boundary conditions for the engine inlet and inlet boundary conditions for the engine outlet). The thermodynamic values used to define these boundary conditions were chosen to mimic an Ultra-high bypass ratio (UHBR) turbofan in terms of bypass ratio while ensuring mass conservation between the inlet and the outlet. The relevant microphysical parameters for the engine outlet are given in Tab.2 and were chosen to mimic the emissions of a turbofan burning kerosene *Jet A1*. The soot number density at the engine outlet and the dry soot particle radius are of the order of magnitude of previous studies (see Montreuil et al. (2018); Khou et al. (2015); Ramsay et al. (2024)) and is in the same order of magnitude of in-situ measurements performed by Petzold et al. (1999). The corresponding soot emission index is $\mathrm{EI}_{soot} = \frac{N_s A_{\mathrm{core}} U_{\mathrm{core}}}{Y_{\mathrm{H_2O}} \dot{m}_{\mathrm{core}}} \mathrm{EI}_{\mathrm{H_2O}}$ with $A_{\mathrm{core}}$, $U_{\mathrm{core}}$ and $\dot{m}_{\mathrm{core}}$ being respectively the surface area of the core flow, the core flow exit velocity and the core flow mass flow rate. The water vapor emissions are based on the study conducted by Garnier et al. (1997), which characterizes the emissions of a Boeing 767 equipped with Rolls-Royce RB211 engines. It is important to note that these emissions are unlikely to be representative of a modern UHBR engine, given that the RB211 has a significantly lower bypass ratio of approximately five. Consequently, it would be interesting for future work to use more realistic engine emissions. Using those water vapor emissions yields the value $\mathrm{EI}_{soot} \approx 10^{15} \ \mathrm{kg}^{-1}$.

The mesh of the RANS domain was generated using an anisotropic mesh adaptation technique based on the total kinetic energy (mean flow kinetic energy + turbulent kinetic energy) field. This was done thanks to the software Feflo.a from INRIA based on Loseille and Alauzet's work (see Loseille and Alauzet (2011a, b)). The final mesh contains around 110 million cells with around 40 million prisms and 70 million tetrahedra. It is the same mesh that was used in Bouhafid et al. (2024) for the study of the near-field aerodynamics of the CRM geometry in cruise flight conditions. Keeping the same mesh after activating the microphysical processes is equivalent to assuming that these processes are not energetic enough to significantly influence the aerodynamic properties of both the vortex wake and the jet. To illustrate the impact of anisotropic mesh adaptation, a mesh cut around the engine is shown in Fig.4. More details about the anisotropic mesh adaptation process and its application to the CRM geometry can be found in Bouhafid et al. (2024).

| **Atmospheric & engine conditions** | | | | |
| --- | --- | --- | --- | --- |
| Flight conditions | $\mathbf{M}_\infty$ <br> 0.85 | $\alpha$ <br> 3° | $\mathbf{p}_\infty$ <br> 264.37 hPa | $\mathbf{T}_\infty$ <br> 223.15 K |
| Core engine outlet | **Stagnation temperature** <br> 626.41 K | **Stagnation pressure** <br> 530.01 hPa | | |
| Bypass engine outlet | 297.23 K | 699.21 hPa | | |
| Engine inlet | **p** <br> 370.12 hPa | | | |

**Table 1.** Atmospheric & engine conditions at cruise flight

| Microphysical model parameters | | | |
|---|---|---|---|
| Atmospheric conditions at flight altitude | $RH_{liq}$ 68% | $RH_{ice}$ 109% | $N_s$ 0 |
| Core engine outlet | $Y_{air}$ $\approx 0.98$ | $Y_{H_2O}$ $\approx 0.02$ | $Y_{SO_3}$ $3 \cdot 10^{-10}$ |
| | $Y_{H_2SO_4}$ $1.4 \cdot 10^{-8}$ | $N_s$ $3.4 \cdot 10^{12}$ m$^{-3}$ | $r_s$ 27 nm |
| Bypass engine outlet | $Y_{air}$ $\approx 0.9999$ | $Y_{H_2O}$ $\approx 1 \cdot 10^{-4}$ | |

**Table 2.** Atmospheric & engine emissions at cruise flight conditions

## 3.2 LES domain

The LES domain is obtained by extruding along the flight direction $x$ the slice $x/b = 18$ extracted from the RANS calculation as described in Sec.2.3. The extrusion distance is equal to one Crow instability wavelength $\lambda_{Crow} = 8.6b_0 = 396$ m $\approx 400$ m. The LES domain is $20b \approx 1.2$ km long in the $y$ and $z$ direction in order to avoid spurious boundary effects. The LES mesh contains approximately 160 million hexahedral cells, with 100 cells in the flight direction. Such resolution in the axial direction is enough to correctly solve Crow instability but might not be enough if we aim to correctly resolve all short wavelength instabilities that could appear, as stated in Bouhafid et al. (2024). The mesh resolution in the $y$ and $z$ direction is maximum in the zone where the vortices come down because of their mutual induction. The length of the cells was chosen in order to provide 10 points per vortex core radius $r_c$, i.e. $dy = dz \approx 0.1r_c$.

Periodic boundary conditions are imposed for the boundaries perpendicular to the flight direction $x$ while far-field boundary conditions are chosen for the other boundaries where the fluid velocity is equal to 0 and the pressure and temperature fields are imposed to ambient values according to stratification. Temporal integration is performed using Implicit Runge-Kutta 2nd-order method with a time step chosen to verify $CFL < 0.5$, $CFL$ being the Courant number. Each LES calculation simulates about 200 s of physical time.

The synthetic turbulence in the jet region is generated using a Karman-pao spectrum. The corresponding turbulence dissipation rate $\epsilon_{jet}$ and integral length scale are deduced from the RANS calculation and are respectively equal to 50 m$^2$/s$^{-3}$ and $r_j = 6$ m, which is the jet radius at $x/b = 18$. The turbulence spectrum is discretized from the integral length scale to the smallest cell length.

Atmospheric conditions in the temporal LES are determined by atmospheric turbulence dissipation rate $\epsilon_{atm}$ and Brunt-Väisälä frequency $N_b$ characterizing stratification. Atmospheric turbulence is generated by adding synthetic turbulence fluctuations to the LES initial velocity field. Horizontal and vertical fluctuations are respectively modeled using the horizontal spectrum $E_h = C_k \epsilon_{atm}^{2/3} k^{-5/3}$ and vertical spectra $E_v = \frac{C_k}{r^2} \epsilon_{atm}^{2/3} k^{-5/3}$. $C_k$ is the Kolmogorov constant set at 1.5 (see Pope (2001)) and $r$ is the anisotropy factor defined by $r = \frac{u'_h}{u'_v}$ with $u'_h$ and $u'_v$ being respectively the root mean square values of the horizontal and vertical fluctuations of atmospheric turbulence. The two spectra are discretized from $k = k_{Crow} = \frac{2\pi}{\lambda_{Crow}}$ to

$k = \sqrt{8}k_{\text{Crow}}$. $r$ is set to 5/4 (see Holzäpfel et al. (2001)) and $\epsilon_{\text{atm}}$ to $6 \times 10^{-5}$ m$^2$/s$^{-3}$, a typical value in the upper troposphere as measured in Cho and Lindborg (2001). The correspond turbulent time scale is $\tau_b = \left( \frac{\lambda_{\text{Crow}}^2}{\epsilon_{\text{atm}}} \right)^{1/3}$. For the CRM geometry and the given value of $\epsilon_{\text{atm}}$, $\tau_b \approx 23$ min. Since the dissipation regime lasts only a few minutes, it is not necessary to force atmospheric turbulence to avoid turbulence decay.

Concerning stratification, two Brunt-Väisälä frequencies are considered in this work: $N_b = 0.012$ s$^{-1}$ and $N_b = 0.03$ s$^{-1}$.
The first case corresponds to a standard value for medium stratification while the second case corresponds to a strong stratification scenario. These values are consistent with previous observations made in Schumann et al. (2017). Knowing the value of $N_b$, the vertical profile for ambient pressure and ambient temperature can be deduced by the equations:

$$\frac{N_b^2}{g} = \frac{1}{\Theta_\infty} \frac{d\Theta_\infty}{dz} \tag{16}$$

$$\frac{dp_\infty}{dz} = -\rho_\infty(z)g \tag{17}$$

where $\Theta_\infty$ is ambient potential temperature, $g$ is gravitational acceleration, and $\rho_\infty$ ambient air density. The solution equations can be found in Paoli et al. (2013). Solving those equations for the case $N_b = 0.03$ s$^{-1}$ with the chosen ambient pressure and ambient temperature at flight altitude gives rise to a temperature inversion phenomenon. That is to say, the temperature decreases when altitude decreases instead of increasing as observed for $N_b = 0.012$ s$^{-1}$. This gives rise to an increase in relative
humidity when altitude decreases as seen in Fig.5 where the ice relative humidity contour is plotted. Temperature inversion layers in the tropopause were already observed in the past, and their characteristics were examined in previous works (see Birner (2006); Grise et al. (2010)). Note that it is assumed in this work that the ambient water vapor molar fraction is constant in the whole domain. Its value is equal to the ambient water vapor molar fraction at cruise flight set for the precursor near-field calculation. Note that with this hypothesis, very high values of $\text{RH}_{\text{ice}}$ are reached at the bottom of domain in the strong
stratification scenario. However, we will see below that no ice crystals can descend to such low altitudes for this scenario because of the intense baroclinic vorticity that develop in the secondary wake. The maximum ambient relative humidity with respect to ice reached by the contrail in the simulation is about 165%, as will be discussed later. Then, parts of the simulation domain where $\text{RH}_{\text{ice}} > 165\%$ are not expected to have any influence on the simulation.

## 4  Results and discussion

The results of the contrail simulations from the near-field to the far-field are discussed in this section for the two stratification scenarios. In order to evaluate the impact of the temporal LES initialization strategy, another contrail simulations of the far field is carried out using the classical analytical initialization strategy commonly used in literature, where the vortices are idealized by a pair of two Lamb-Oseen vortices as in Unterstrasser and Görsch (2014); Unterstrasser (2014); Holzäpfel et al. (2001); Paoli et al. (2013). A comparison is made between the RANS initialization strategy and the analytical initialization strategy.

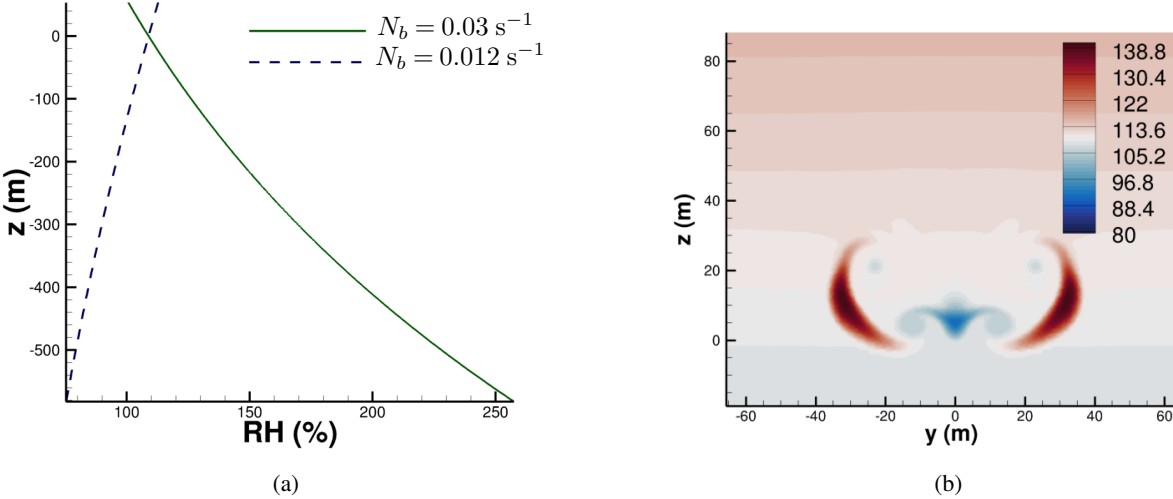

**Figure 5.** (a) Evolution of $RH_{ice}(z)$ in the LES domain and (b) $RH_{ice}$ field in the plume for $N_b = 0.012\ \text{s}^{-1}$ at $t = 0$

Explanations are given to understand the observed differences. In light of the obtained results, a new simulation is performed using an analytical initialization strategy with four Lamb-Oseen vortices, two for the main wing-tip vortices and two for the horizontal tailplane vortices. Finally, a LES simulation with two Lamb-Oseen vortices but with the plume obtained from the RANS simulation is carried out to investigate the effects of plume initialization. Note that all the analysis made below concern mainly the contrail evolution and its associated microphysics fields since the analysis of the aerodynamic fields was already done in Bouhafid et al. (2024).

### 4.1 Evolution of the contrail in the near-field

The contrail in the near-field can be visualized in Fig.6 where the cells satisfying the condition $\theta > 0.01$, $N_s > 10^7\ \text{m}^{-3}$ and $r_p > 1.5 r_s$ are blanked. For those conditions and the given atmospheric conditions, the contrail appears at around one wingspan behind the engines as seen in Fig.6b. Moreover, Fig.6a allows us to observe the jet deviation caused by the wing-tip vortex, a deviation that is followed by the beginning of the jet roll-up around the vortex.

In order to quantify the evolution of the contrail in the near-field, the relevant microphysical quantities are averaged in a cross-section of the contrail for a given position $x$. The contrail is identified by the cells satisfying the condition $r_p > r_s$ (strict inequality). The average is weighted by the local number of ice crystals in a given cell for ice crystal radius $r_p$ and activated surface fraction $\theta$. On the other hand, ice water content, ice crystals number concentration and relative humidity are averaged with weights equal to cell volume. The averaged microphysics quantities are plotted in Fig.7. Fig.7a displays the evolution of the average ice crystal number concentration $N_p$ (not to be mistaken for soot number density $N_s$) as a function of $x/b$. It can be observed that the dilution of the plume reduces ice crystal number concentration by almost two orders of magnitude by the end of the jet regime. The dilution is faster for the first few wingspans because of the jet strong turbulent diffusion. As turbulence

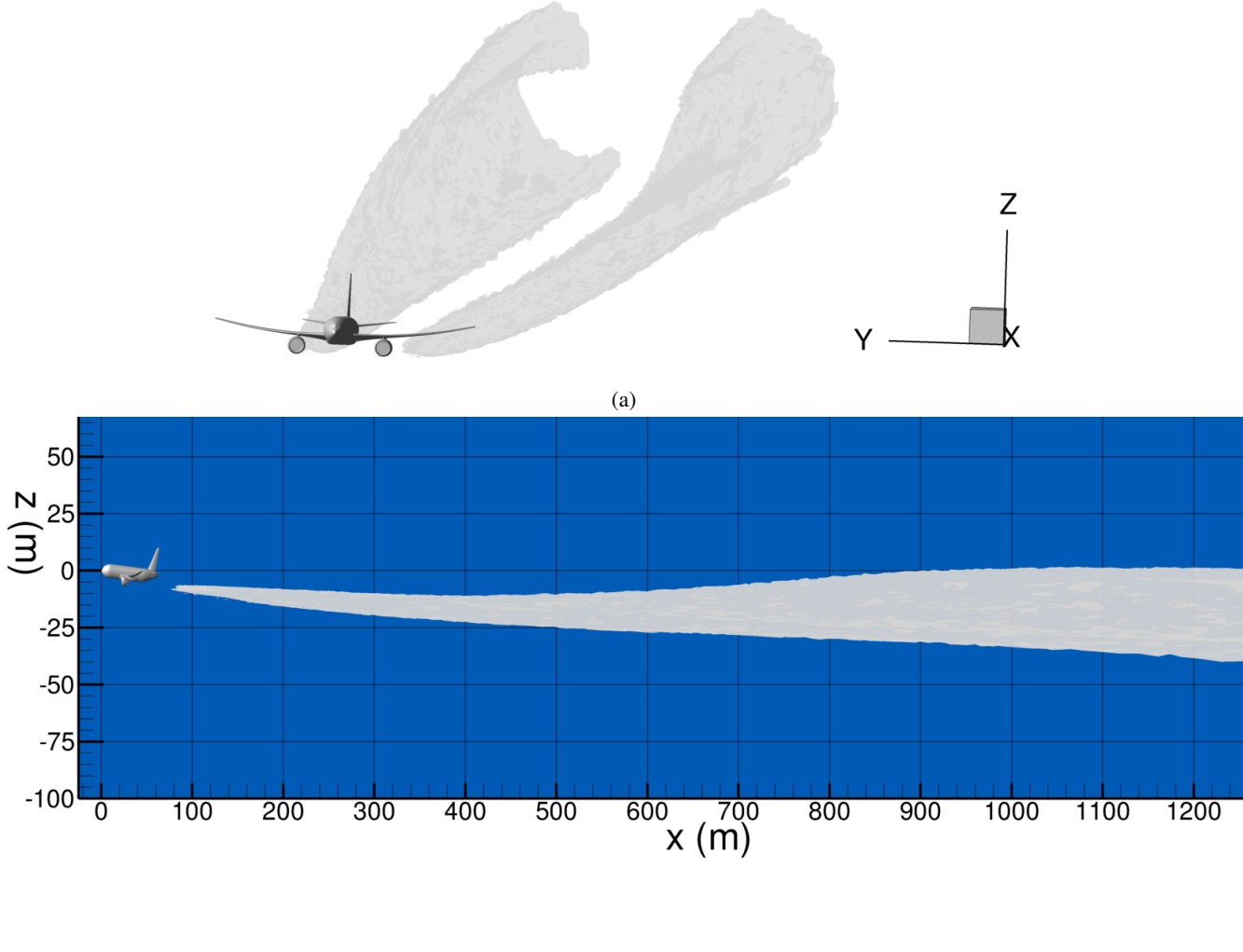

(a)

(b)

**Figure 6.** Mesh cells satisfying $\theta > 0.01$, $N_s > 10^7$ m$^{-3}$ and $r_p > 1.5r_s$. (a) 3D view. (b) $z - x$ view.

levels inside the plume decrease when the distance behind the aircraft increases, dilution becomes slower, and the influence of
the vortex on plume dilution becomes more important than turbulent diffusion.

The evolution of the average ice water content $IWC = \rho Y_{\mathrm{ice}}$ plotted in Fig.7b can be divided into three zones. First, the
IWC evolution shows a sharp growth for the first few wingspans, with an increase equal to four orders of magnitude compared
to $x/b = 0.5$. This strong ice production results from the condensation of the excess water vapor present in important quantities
in the plume in the early stage of the jet regime. This can be seen too in Fig.7e where relative humidity strongly increases for
the first three wingspans and where $\mathrm{RH}_{\mathrm{liq}}$ and $\mathrm{RH}_{\mathrm{ice}}$ reach values of 110% and 160%, respectively. Second, for $x/b$ between
5 and 15, the evolution of IWC is no longer monotonous as two maxima appear for $x/b = 4.5$ and $x/b = 11$. This can be

explained by the fact that the evolution of IWC results from the competition between two phenomena: ice water dilution and ice water production. Indeed, when plume dilution is greater than ice water production, IWC will decrease and vice-versa when plume dilution is smaller. For the first three wingspans, the effect of ice water production is superior to the effect of dilution

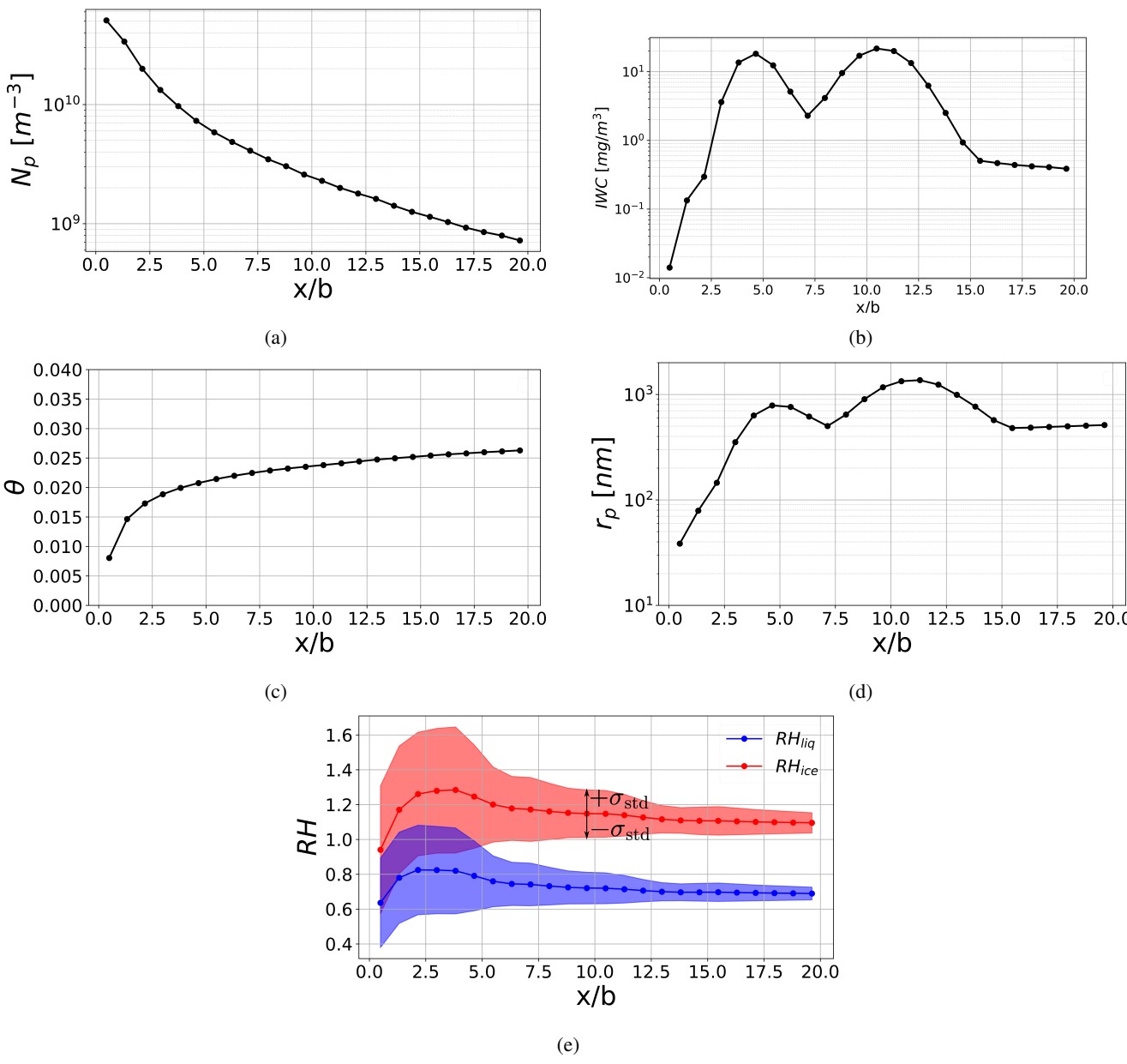

**Figure 7.** Microphysical quantities averaged in a cross-section of the contrail located at a distance $x$ behind the aircraft. (a) Ice crystal number concentration, (b) ice water content, (c) activated surface fraction, (d) mean crystal radius, (e) relative humidity with respect to ice water and liquid water. $\sigma_{\mathrm{std}}$ is the standard deviation.

because of the important quantity of excess water vapor in the plume. However, looking at Fig.7e, relative humidity reaches a maximum at $x/b = 3$, that is to say, shortly before IWC does. This results in a decrease in ice water production. Finally, for $x/b > 15$, the value of IWC reaches a plateau that can be interpreted as an equilibrium between ice water dilution and ice water production.

Fig.7c plots the evolution of the averaged activated surface fraction $\theta_{\mathrm{act}}$. It appears that soot activation occurs very quickly
after the exhaust gases are ejected from the engine. An asymptotic value $\theta_{\mathrm{act}} \approx 2.5 \%$ is reached at the end of the calculation domain. The existence of an asymptotic value was already observed in Khou (2016) and Kärcher 1D model developed in Kärcher (1998). This was explained by the effect of dilution on sulfur species and the absence of desorption in the used model, meaning that the activated surface fraction of a soot particle cannot decrease.

Regarding the evolution of the averaged mean ice crystal radius $r_p$ plotted in Fig.7d, we can make the same observations
and analysis as the ones made for IWC. Indeed, since $r_p \propto \sqrt[3]{\mathrm{IWC}/N_p}$ (see Eq.14), the evolution of $r_p$ will closely follow the evolution of IWC when IWC changes more rapidly than $N_p$. At the end of the domain, $r_p$ reaches an asymptotic value equal to $500\,\mathrm{nm}$. This value is of the same order of magnitude compared to values obtained in previous similar studies (see Khou et al. (2015); Ramsay et al. (2024); Paoli et al. (2013)). Note that the exact value will strongly depend on the ambient relative humidity, soot number density at the engine exit, and soot activation in the plume. Indeed, the work done by Ramsay et al.
(2024) used a very similar microphysical model and assumed that every soot particles is activated, which explains to some extent why the ice crystals obtained in their simulations are larger compared to this work's.

This may appear counterintuitive, as a higher number of available nucleation sites generally leads to increased competition for water vapor, resulting in smaller ice crystals. However, the source term for ice formation used in this work microphysical model (see Eq. 11) is directly proportional to the activated surface fraction $\theta_{\mathrm{act}}$ with $\theta_{\mathrm{act}} \leq 1$. Thus, a small value of $\theta_{\mathrm{act}}$ will
result in a slower rate of ice formation on soot particles which will in turn result in smaller crystals if we assume the same number of soot particles and same quantity of excess water vapor. The microphysics model used in this work assumes that the ice formed on the activated fraction of the soot is redistributed over the soot particle without increasing the activated surface fraction. In reality, initial ice formation creates a cap on the activated surface, increasing the hydrophilic surface fraction. Since the soot radius is negligible compared to the final ice crystal size, the effective hydrophilic surface fraction should reach unity.
This behavior is not captured by the current model, leading to underestimating ice crystals size which results in smaller ice crystals than in Ramsay et al., where $\theta_{\mathrm{act}} = 1$ is assumed from the start.

The mixing line of the plume is plotted in Fig.8 in $(p_v, T)$ diagram, where $p_v$ is the water vapor pressure and $T$ is the temperature. This line is obtained by averaging fluid variables over twenty streamlines originating from the engine core outlet. As the plume ages, both temperature and water vapor pressure decrease due to dilution with the cold ambient air. When the
mixing line goes above the saturation pressure curve for liquid water, ice crystals begin to form. These crystals will grow as long as the mixing line remains above the saturation pressure curve for ice, which occurs at approximately $T = 243$ K for the flight conditions considered.

To examine the distribution of $(p_v, T)$ within the plume, the $(p_v, T)$ values in every cell within the plume region — defined as those where $N_s > 0.1 N_{s,\mathrm{max}}$, with $N_{s,\mathrm{max}}$ being the maximum $N_s$ value at a given position $x$ — are plotted for various

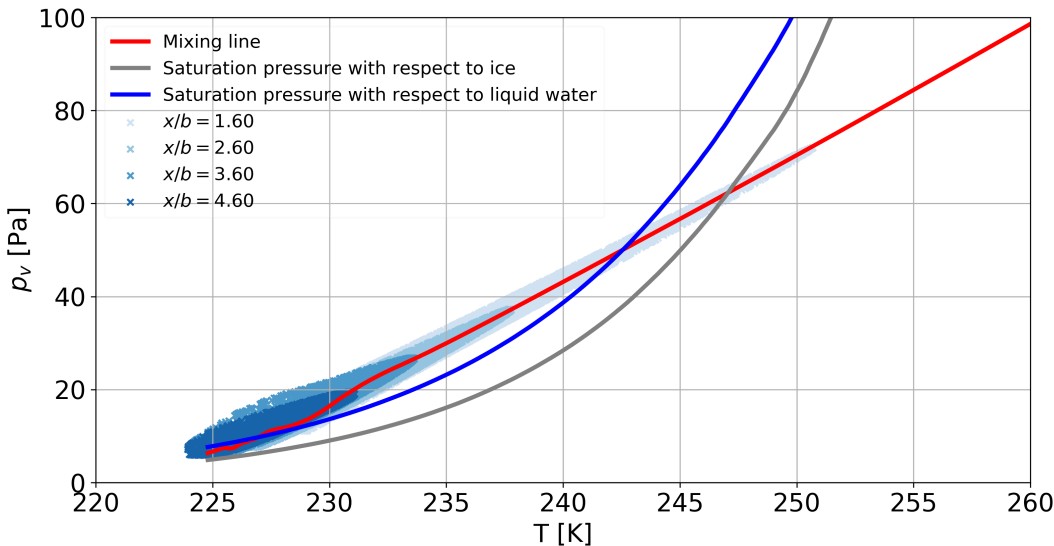

**Figure 8.** Mixing line of the plume in the near-field plotted in a $(p_v, T)$ diagram. The plume is obtained by averaging fluid variables on thirty streamlines originating from the engine core flow.

positions $x$ downstream of the aircraft. The obtained scatterplot provides a clear visualization of how temperature and water vapor pressure vary within the plume for a given plume age. Notably, the older the plume, the less scattered the points become, reflecting the increasing uniformity caused by plume dilution. Finally, we observe that the mixing line differs from a straight line for $T < 230$ K as $p_v$ decreases more abruptly. This is most likely due to the effect of water vapor condensation which decreases $p_v$ more rapidly than the plume isobaric mixing.

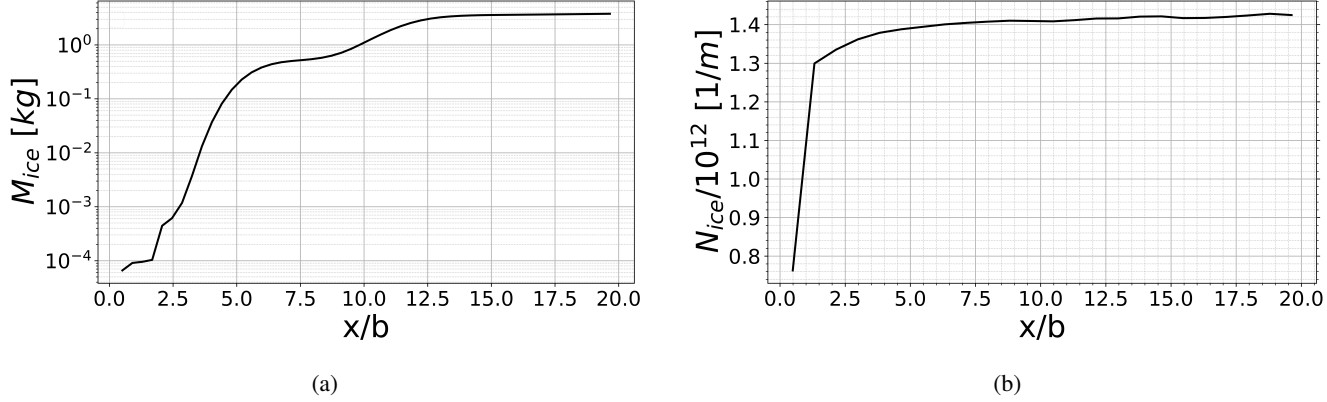

(a)          (b)

**Figure 9.** Evolution of the (a) total ice mass and (b) total ice crystal number per meter of flightpath

The evolution of the total ice mass $\mathcal{M}_{\text{ice}}$ and the total ice crystal number per meter of flightpath $N_{\text{ice}}$ are plotted in Fig.9 as a function of the distance $x/b$ behind the aircraft. More precisely, $\mathcal{M}_{\text{ice}}(x)$ is defined as $\iiint\limits_{V(x)} \text{IWC} \mathrm{d}V$ and $N_{\text{ice}}(x)$ as $\iint\limits_{S(x)} N_p \mathrm{d}S$. $V(x)$ is the volume between the domain inlet and $x$ and $S(x)$ is the contrail cross-section at a given position $x$. The integrals are calculated using the rectangle method. It is worth noting that only half of the geometry is represented in the RANS domain, meaning that only half of the contrail is included in the simulation. Concerning $\mathcal{M}_{\text{ice}}(x)$, we observe a sharp increase around $x/b = 2$, which corresponds to the moment when the averaged relative humidity with respect to liquid water is greater than $100\%$ in the plume. The total mass of ice grows before reaching a plateau around $x/b = 13$. When it comes to the total ice crystal number per meter of flightpath, the evolution of $N_{\text{ice}}$ shows a sharp increase for the first few wingspans before quickly reaching a plateau around $x/b = 5$.

## 4.2 Evolution of the contrail in the far-field

The results of four contrail far-field LES simulations are analyzed and discussed in this subsection. The first two simulations are initialized using the near-field RANS simulation. The other three simulations are initialized using an analytical initialization strategy which proceeds as follows. A vortex pair is initialized with two Lamb-Oseen vortices located at the same position as the RANS vortex pair. The velocity inside the vortices is purely tangential and is defined by:

$$V_{\theta,\text{LO}}(r) = \frac{\Gamma}{2\pi r}\left(1 - e^{-\beta \frac{r^2}{r_c^2}}\right) \tag{18}$$

$$\frac{\mathrm{d}p_{\text{LO}}}{\mathrm{d}r} = \rho \frac{V_{\theta,\text{LO}}}{r} \tag{19}$$

with $V_{\theta,\text{LO}}$ is the tangential velocity, $r$ the radial coordinate defined from the center of the vortex, $\Gamma$ the vortex circulation, $r_c$ the vortex core radius, $p_{\text{LO}}$ the static pressure inside the vortex and $\beta \approx 1.256$. $r_c$ and $\Gamma$ are deduced from the RANS calculation at $x/b = 18$. The plume is modeled by adding two circular jets with purely axial flow and located at the position of the jet engines in the CRM geometry. The radius of the jet is obtained from the RANS calculation as it was done in Bouhafid et al. (2024). All the flow field variables are uniform inside the jet, with a hyperbolic tangent decrease outside the jets. The value of the flow field variables in the jets is deduced from the RANS calculation. Moreover, the parameters of the hyperbolic tangent decrease are calibrated for the mass fraction fields in order for the total mass of each species to be identical to the total mass obtained from the RANS initialization at $t = 0$. The analytical initialization strategy with two Lamb-Oseen vortices will be noted 2LO from now on. Finally, time is scaled by the vortex pair characteristic time $t_b = \frac{2\pi b_0^2}{\Gamma} = 25.7$ s, which is the theoretical time for the vortex pair to descend a distance equal to $b_0$ as a result of their mutual induction.

Fig.10 shows at $t/t_b = 3.89$ and $t/t_b = 7.78$ the soot number density isosurface $N_s = 10^7$ m$^{-3}$ colored by mean crystal radius $r_p$ for the medium stratification scenario $N_b = 0.012$ s$^{-1}$. The important differences regarding the aerodynamic field observed and explained in Bouhafid et al. (2024) between the 2LO initialization and RANS initialization significantly impact

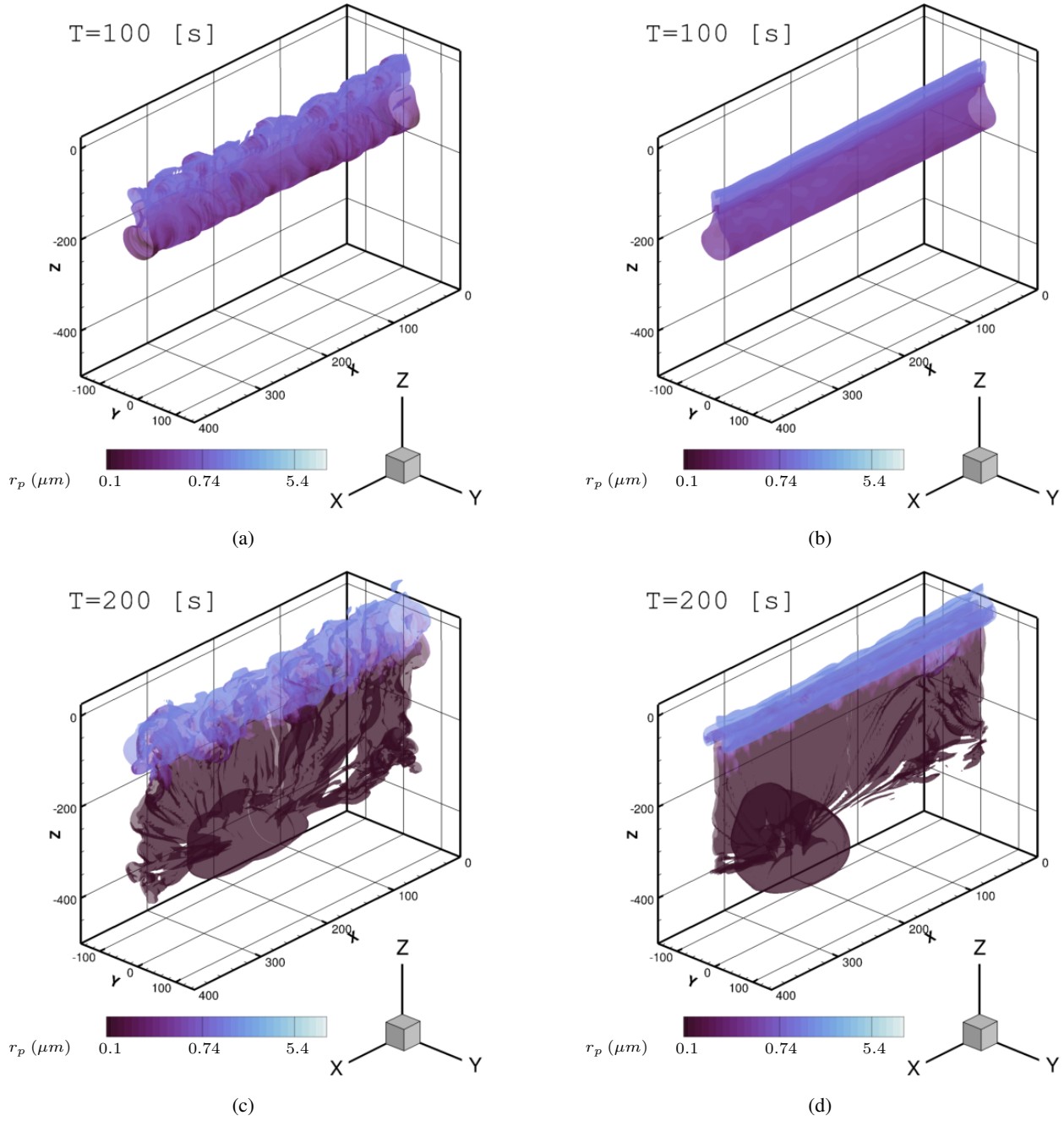

**Figure 10.** Soot number density isosurface $N_s = 10^7 \, \text{m}^{-3}$ colored by mean crystal radius $r_p$ in logarithmic scale. $N_b = 0.012 \, \text{s}^{-1}$ (medium stratification). Comparison between RANS initialization strategy and 2LO initialization strategy. (a) $t/t_b = 3.89$ RANS initialization, (b) $t/t_b = 3.89$ 2LO initialization, (c) $t/t_b = 7.78$ RANS initialization and (d) $t/t_b = 7.78$ 2LO initialization. Distances are in meters ($t_b = 25.7 \, \text{s}$).

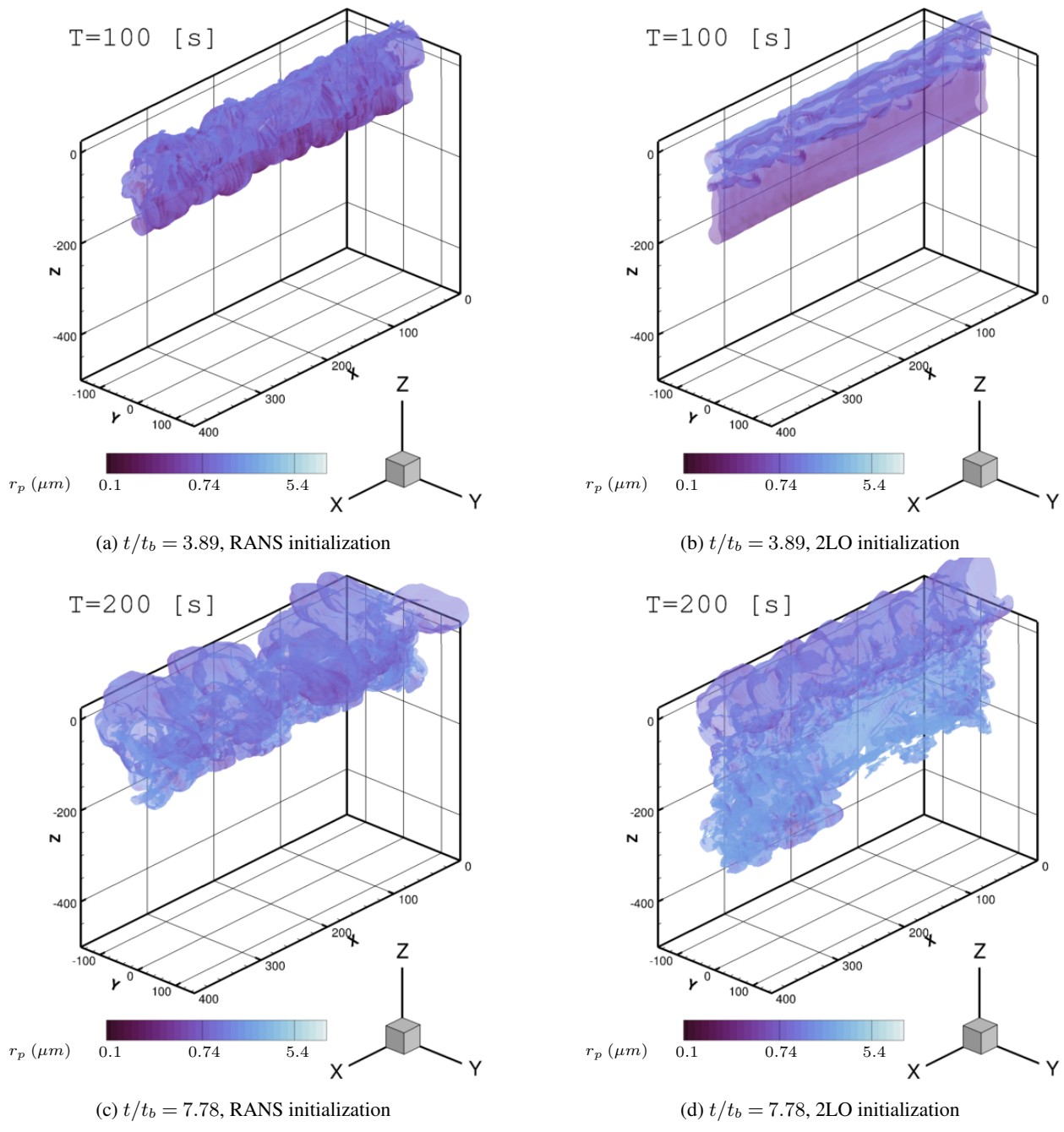

**Figure 11.** Soot number density isosurface $N_s = 10^7$ m$^{-3}$ colored by mean crystal radius $r_p$ in logarithmic scale. $N_b = 0.03$ s$^{-1}$ (strong stratification). Comparison between RANS initialization strategy and 2LO initialization strategy. (a) $t/t_b = 3.89$ RANS initialization, (b) $t/t_b = 3.89$ 2LO initialization, (c) $t/t_b = 7.78$ RANS initialization and (d) $t/t_b = 7.78$ 2LO initialization. Distances are in meters ($t_b = 25.7$ s).

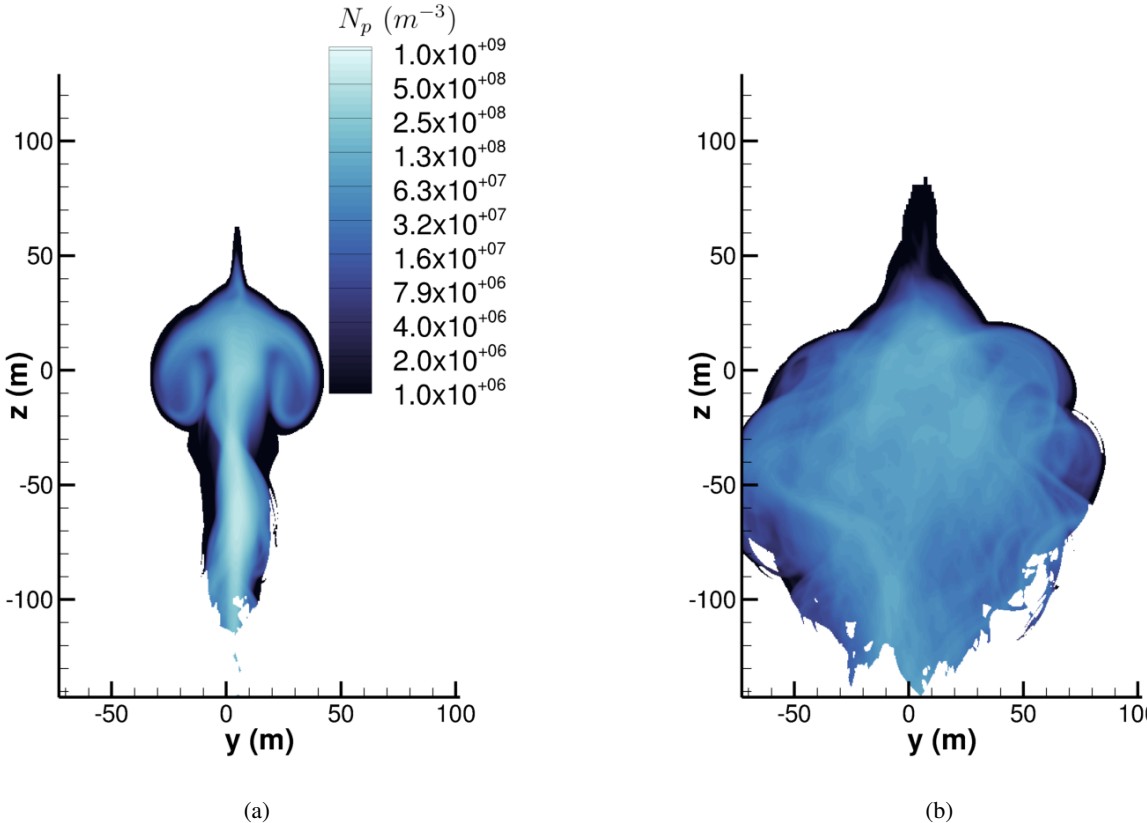

**Figure 12.** Spatially averaged ice crystal number concentration field $N_p$ in m$^{-3}$ at $t/t_b = 8$ ($t_b = 25.7$ s) and for $N_b = 0.012$ s$^{-1}$. Comparison between (a) *2LO initialization* and (b) *RANS initialization*. A logarithmic scale is used for the color levels.

the contrail shape and properties. While there are some similarities between the two cases, like the formation of a vortex ring following Crow instability, the contrail in the 2LO case appears much smoother. Conversely, the turbulent nature of the secondary wake in the RANS initialization case originating from short-wave vortex instabilities gives the contrail a more chaotic appearance, while increasing the size of the contrail at flight altitude.

The differences between the 2LO initialization case and RANS case are even greater for the strong stratification scenario $N_b = 0.03$ s$^{-1}$ for which the same soot number density isosurface is plotted in Fig.11 at $t/t_b = 3.89$ and $t/t_b = 7.78$. As expected, more ice crystals are found in the secondary wake for the two initialization strategies compared to the medium stratification scenario. Since atmospheric conditions are favorable for contrail persistence in the considered strong stratification scenario because of thermal inversion, the resulting contrail will contain a higher ice water content and larger crystals. However, we observe that the contrail goes significantly lower in the 2LO case. This results in a wider contrail for the RANS case. A possible explanation for this phenomenon could be that the vortices are more unstable in the RANS case and are destroyed sooner, thus preventing the vortices from going downwards and carrying ice crystals to lower altitudes. The unstable nature of the wake can be amplified either by the development of Windall instability in the RANS initialization—manifesting under cer-

tain conditions but absent in the 2LO initialization computation—or by the presence of horizontal tailplane vortices, resulting in a four-vortex system. This will be further examined in Section 4.3.

The differences observed can be quantified by spatially averaging the contrail along flight direction $x$, with the cells in the contrail being identified by $r_p > r_s$. The strict inequality ensures that only ice crystals are considered and not dry soot particles. The ice crystal number concentration field obtained in the averaged contrail cross section is plotted in Fig.12 and Fig.13 for $t/t_b = 8$ ($t = 200$ s). More precisely, the ice crystal number corresponds to the soot number density but only for soot particles with an ice layer on the surface. Thus, ice crystal number concentration is noted $N_p$ in order to distinguish it from the soot

number density $N_s$..

     We observe that, for the 2LO initialization under strong stratification ($N_b = 0.03$ s$^{-1}$), a small fraction of ice crystals reaches the minimum altitude of $z = -300$ m, where the ambient relative humidity with respect to ice is approximately $RH_{\text{ice}} \approx 165\%$. Such high values have previously been reported in the North Atlantic flight corridor, as documented by several measurement campaigns summarized in Ovarlez et al. (2000) (see Fig. 1 in the corresponding paper). Although a relative humidity of about

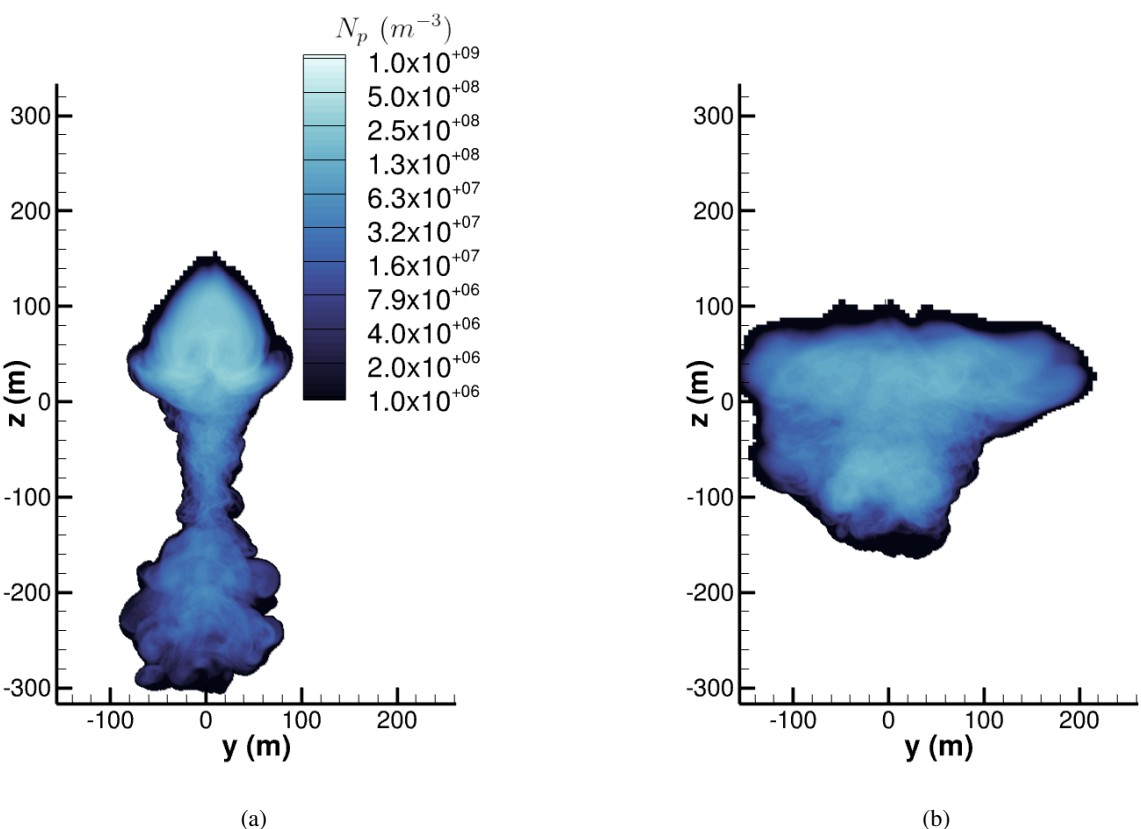

(a)                                                                                 (b)

**Figure 13.** Spatially averaged ice crystal number concentration field $N_p$ in m$^{-3}$ at $t/t_b = 8$ ($t_b = 25.7$ s) and for $N_b = 0.03$ s$^{-1}$. Comparison between (a) *2LO initialization* (b) *RANS initialization*. A logarithmic scale is used for the color levels.

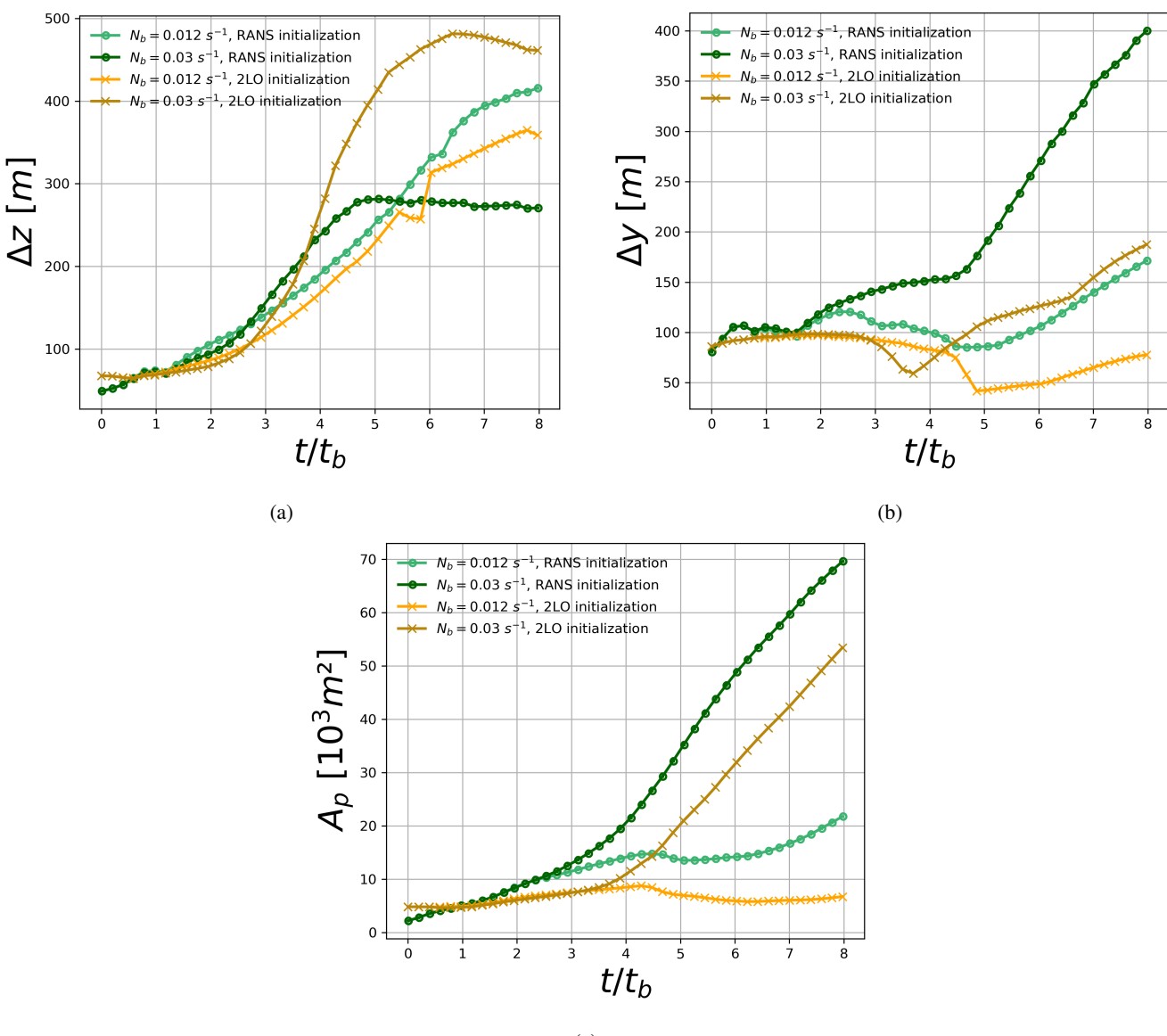

**Figure 14.** Contrail length scales for the different stratification scenarios. Comparison between the 2LO initialization strategy and the RANS initialization strategy. ($t_b = 25.7$ s). (a) Contrail vertical extent, (b) contrail width and (c) area of contrail cross-section.

165% is uncommon, it is not exceptional. Such high value of $RH_{\text{ice}}$ is very likely to increase the number of increasing ice crystals.

Fig.14 displays the relevant contrail length scales for the RANS initialization and 2LO initialization cases. Contrail vertical extent $\Delta z$, contrail width $\Delta y$, and contrail cross-sectional $A_p$ area are plotted versus scaled time $t/t_b$. Contrail vertical extent $\Delta z$ is computed by taking the difference between the maximum value of $z$ and the minimum value of $z$ in the contrail. Contrail

width is computed the same way but with the $y$ coordinate. Contrail cross-sectional area is calculated by summing the area of every cell inside the contrail.

Regarding contrail vertical extent, the differences between the RANS initialization and 2LO initialization remain small for the medium stratification scenario. However, for the strong stratification scenario, the contrail is $1.6$ times taller for the 2LO initialization case compared to the RANS initialization case. The differences between the four different cases are larger

for the contrail width. For both stratification scenarios, the contrail is two times wider for the RANS initialization compared to the equivalent 2LO initialization. For contrail cross-sectional area, two observations can be made. First, for the RANS initialization, the influence of stratification becomes visible at $t/t_b = 2.5$ while it happens later for the 2LO initialization, at around $t/t_b = 3.5$. Second, the contrail is bigger and grows more rapidly for the strong stratification scenario. This is simply due to the higher ambient relative humidity $RH_{ice}$ for $N_b = 0.03\ \mathrm{s}^{-1}$ resulting from thermal inversion. For $N_b = 0.012\ \mathrm{s}^{-1}$,

$RH_{ice}$ eventually decreases below $100\%$ as the vortices descend, resulting in ice water sublimation and a decrease of the contrail cross-sectional area at $t/t_b = 4.3$. Nonetheless, the contrail begins to grow again at later times because of the ice crystals in the secondary wake ascending back to flight altitude where air is supersaturated with respect to ice water. It is interesting to note that for the strong stratification scenario, contrail vertical extent does not increase anymore around $t/t_b = 4.5$ for RANS initialization and $t/t_b = 6.3$ for 2LO initialization while contrail width keeps increasing. This can be explained by the impact

of the vortex break-up which occurs sooner for $N_b = 0.03\ \mathrm{s}^{-1}$ compared to $N_b = 0.012\ \mathrm{s}^{-1}$. The ice crystals are no longer brought downward and the contrail will predominantly spread in the horizontal direction. We expect this phenomenon to occur at later times for $N_b = 0.012\ \mathrm{s}^{-1}$ as we can see that the inflection point for $\Delta z(t)$ is attained at $t/t_b = 6.5$ for the RANS initialization for $N_b = 0.012\ \mathrm{s}^{-1}$.

Contrail total ice mass per meter of flightpath $m_{ice} = \mathcal{M}_{ice}/L_x$, total ice crystal number per meter of flightpath $N_{ice} =$

$\mathcal{N}_{ice}/L_x$ and mean ice crystal radius $\overline{r_p} = \left(\frac{3m_{ice}}{4\pi N_{ice}\rho_{sol}}\right)^{1/3}$ are plotted Fig.15, with $\mathcal{N}_{ice}$, $\mathcal{M}_{ice}$ and $L_x$ being respectively the total number of ice crystals, the total ice mass and the length of the LES domain along flight direction. Concerning ice mass, we observe for every case an initial increase followed by a decrease which is in turn followed by another increase. The initial increase is due to ice formation continuing to occur at flight altitude where ambient relative humidity with respect to ice $RH_{ice}$ is greater to $100\%$.

The decrease that follows is due to a portion of the contrail reaching altitudes where $RH_{ice} < 100\%$. For the case $N_b = 0.012\ \mathrm{s}^{-1}$, this occurs for the ice crystals reaching lower altitude as they are dragged downward by the vortex pair. On the other hand, for the case $N_b = 0.03\ \mathrm{s}^{-1}$, a certain number of ice crystals quickly reach altitudes higher than flight altitude because of the strong baroclinic vorticity and where $RH_{ice} < 100\%$. Ice crystals reaching altitude significantly higher than flight altitude has already been observed in Saulgeot et al. (2023) for extreme stratification scenarios. However, as most ice crystals stay

in super-saturated region for $N_b = 0.03\ \mathrm{s}^{-1}$, ice mass eventually begins to rapidly increase and the influence of initialization strategy can begin to be neglected for $t/t_b > 5.5$. The smaller increase in ice mass at $N_b = 0.012\ \mathrm{s}^{-1}$ can be attributed to the ascent of some ice crystals back to the flight altitude, where conditions are favorable for their growth, while those remaining trapped in the descending vortices shrink.

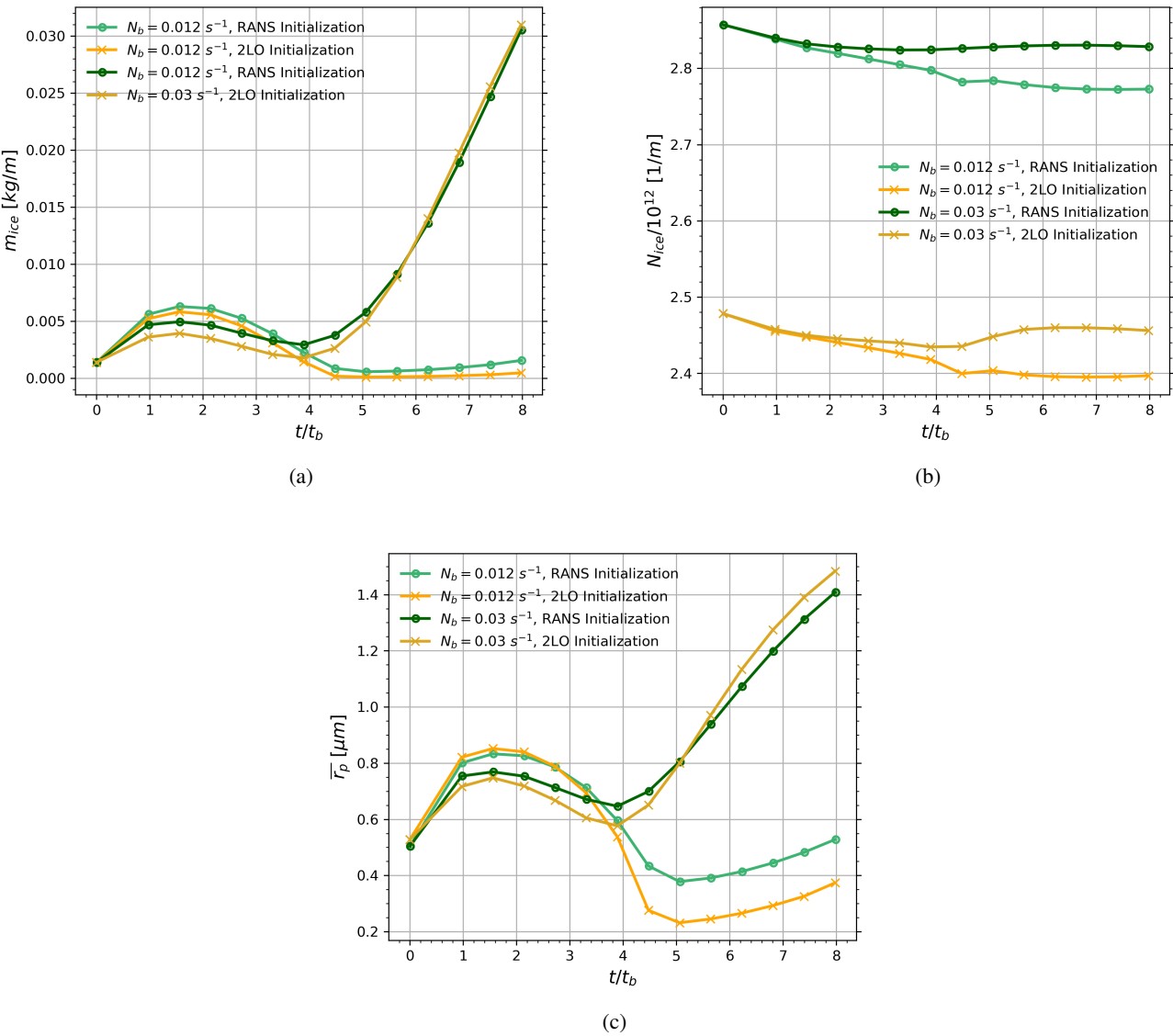

**Figure 15.** Evolution of contrail total quantities for the different stratification scenarios. Comparison between the RANS initialization strategy and the 2LO initialization strategy. (a) Total ice mass per meter of flightpath, (b) total ice crystal number per meter of flightpath and (c) mean ice crystal radius ($t_b = 25.7$ s).

The evolution of $N_{\mathrm{ice}}$ plotted in Fig.15b displays value in the same order of magnitude than previous studies in the literature (see Unterstrasser and Görsch (2014); Lewellen (2014)). We observe that the relative variations in $N_{\mathrm{ice}}$ for a given stratification scenario are much smaller than the variations in the $m_{\mathrm{ice}}$. This allows us to deduce that the increase of $m_{\mathrm{ice}}$ is mainly due to ice

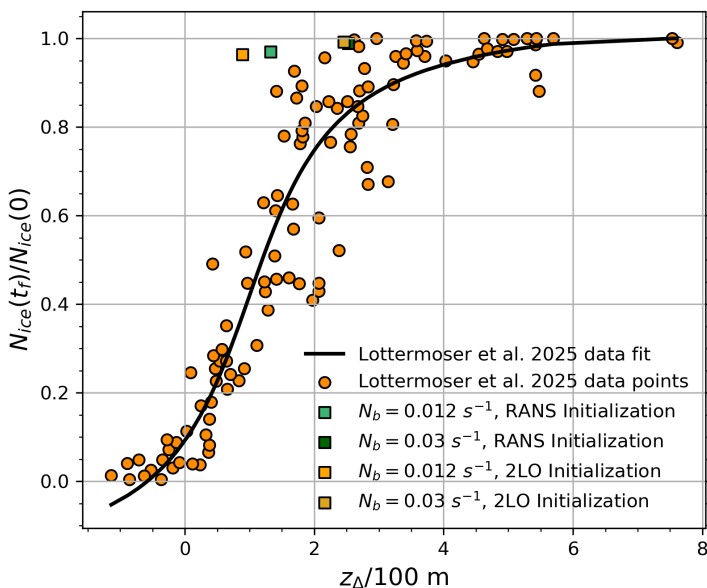

**Figure 16.** Survival fraction of ice crystals $N_{ice}(t_f)/N_{ice}(0)$ as a function of the parameter $z_\Delta$, with $t_f$ the final time of the LES simulation. Comparison with Lottermoser and Unterstrasser (2025) work.

crystals growth which can be observed and quantified in Fig.15c. However, it can still be observed that $N_{ice}$ slightly decreases for $N_b = 0.012 \text{ s}^{-1}$, indicating an overall loss of ice crystals in the contrail.

Comparison between the RANS initialization strategy and the 2LO initialization strategy reveals that the initial offset in $N_{ice}$
remains constant throughout contrail evolution. This suggests that if both initialization strategies were to begin with identical total ice crystal numbers per meter of flight path, the resulting $N_{ice}$ values would likely converge to similar magnitudes. This hypothesis will be examined in greater detail below, given that $N_{ice}$ represents one of the most critical parameters governing contrail radiative forcing in the diffusion regime, as demonstrated by the 2D simulations of Unterstrasser and Gierens (2010).

A comparison with Lottermoser and Unterstrasser (2025) work can be done by computing the $z_\Delta$ parameter and plotting it
in Fig.16. This parameter, which is defined in Lottermoser and Unterstrasser (2025) paper, is used to parametrize ice crystal loss. $z_\Delta$ is a weighted combination of three length scales: the final vertical displacement of the wake, the downward distance an air parcel must travel for its saturation pressure to equal its vapor pressure, and the vertical displacement corresponding to an adiabatic heating that maintains saturation in an initially saturated parcel when emitted water vapor is added.

Mathematically, $z_\Delta$ is a function of wake circulation, Brunt–Väisälä frequency $N_b$, cruise flight ambient temperature $T_\infty$,
cruise flight ambient relative humidity, initial ice crystal number per meter of flight path $N_{ice}(0)$ and contrail cross-section area $A_p$. Physically, a high $z_\Delta$ value indicates sufficient water vapor to maintain ice supersaturation within the plume. Conversely, a

low $z_\Delta$ value indicates that adiabatic heating during the vortex pair descent is strong enough to sublimate a significant fraction of the ice crystals.

Comparisons with the simulation data from Lottermoser and Unterstrasser (2025) indicate that, for the strong stratification case ($N_b = 0.03\,\mathrm{s}^{-1}$), both the RANS and 2LO initializations yield a value of $z_\Delta \approx 2.5$, which lies within the range of the scatter plot. In contrast, for the weaker stratification case ($N_b = 0.012\,\mathrm{s}^{-1}$), the predicted $z_\Delta$ values fall slightly outside the scatter range, with the 2LO initialization showing the largest deviation. Two factors may account for this discrepancy. First, in the present simulations using our microphysical model, the loss of ice crystals appears to be marginal despite a noticeable decrease in $m_{\mathrm{ice}}$. Second, the far-field LES simulations of Lottermoser and Unterstrasser (2025) extend over 400 seconds—nearly twice the duration of our runs. Extending the simulation time in this work could therefore lead to greater ice crystal loss and potentially bring the results closer to the scatter points.

All-in-all, more ice crystals survive in the $N_b = 0.03\,\mathrm{s}^{-1}$ scenario compared to $N_b = 0.012\,\mathrm{s}^{-1}$ as it was expected considering the higher level of ambient $RH_{\mathrm{ice}}$ in the strong stratification scenario.

### 4.3 Analytical initialization strategy with HTP vortices

In order to better understand the interaction between stratification and short wavelength vortex instabilities, another temporal LES simulation is performed for the medium stratification scenario $N_b = 0.012\,\mathrm{s}^{-1}$. This new simulation is similar to the 2LO initialization strategy. The only difference is that the vortex wake is now initialized with four vortices instead of two. To do so, a new Lamb-Oseen vortex pair is added and aims to model horizontal tailplane (HTP) vortices. The circulation, location and core radius of the HTP vortices are deduced from the RANS calculation and are respectively equal to $-77\,\mathrm{m}^2/\mathrm{s}$ and $2.5\,\mathrm{m}$. This new simulation will now be referred to as 4LO.

Fig.17 shows for $t/t_b = 3.89$ and $t/t_b = 7.78$ the soot number density isosurface colored by ice crystal mean radius for both the RANS initialization and 4LO initialization strategies. Qualitatively, the 4LO initialization produces a contrail much more similar to the RANS initialization than to the 2LO initialization. Indeed, the contrail exhibits the same turbulent aspect, both in the primary and secondary wake. This can be understood by looking at Fig.18, where a $Q$-criterion isosurface is plotted at $t/t_b = 3.89$ for both cases to visualize the two vortex tubes of the vortex pair. For the two cases, we observe the development of short-wave instabilities on the vortex tubes that are responsible for the wavy aspect of the vortex tubes. These instabilities are typical of quadripolar vortex systems in a counter-rotating configuration theoretically (see Fabre et al. (2002) theoretical study) and were not observed for the 2LO initialization as it can be seen in Fig.21 of Bouhafid et al. (2024). However, Widnall instability for vortex pairs, theorized in Tsai and Widnall (1976), may also occur as observed in Holzäpfel et al. (2001), despite not being observed in a similar case studied by Picot et al. (2014).

The absence of Widnall instability on the vortex pair for the 2LO initialization can be explained as follows. Widnall instability can occur in vortex pairs and is due to a resonance phenomenon between the transverse shear caused by one vortex on the other, and the existence of Kelvin waves with the same wavenumber $k$ and azimuthal wavenumbers $m$ and $m + 2$. For a pair of Lamb–Oseen vortices, Sipp and Jacquin (2003) demonstrated that a narrow band of Widnall instabilities exists only for Kelvin waves with $m = \pm 1$. The width of this instability band thus depends on the wavenumber $k_w$ of the Kelvin modes, the

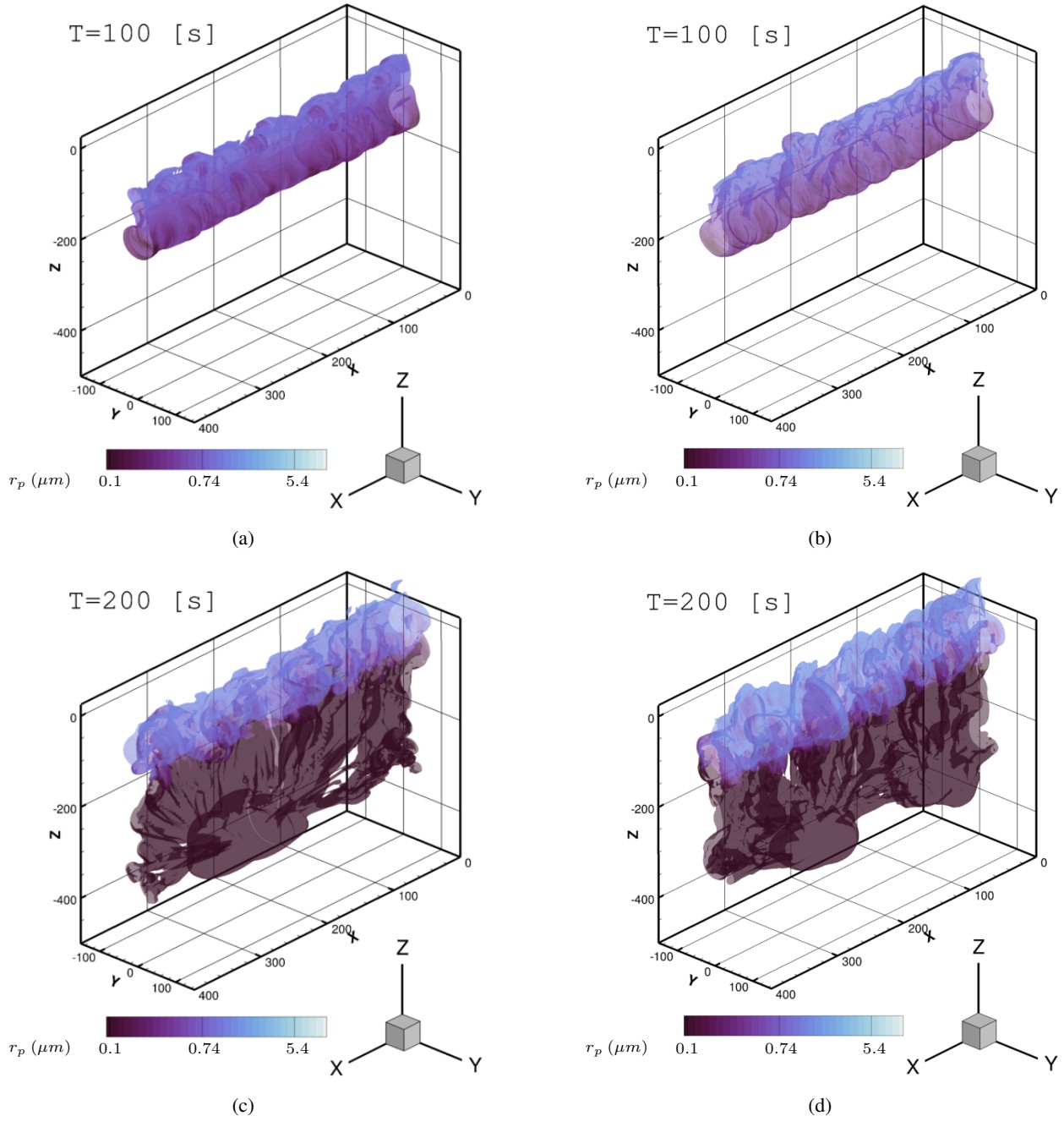

**Figure 17.** Soot number density isosurface $N_s = 10^7$ m$^{-3}$ colored by mean crystal radius $r_p$ in logarithmic scale. $N_b = 0.012$ s$^{-1}$. Comparison between RANS initialization strategy and 4LO initialization strategy. (a) $t/t_b = 3.89$ RANS initialization, (b) $t/t_b = 3.89$ 4LO initialization, (c) $t/t_b = 7.78$ RANS initialization and (d) $t/t_b = 7.78$ 4LO initialization. Distances are in meters ($t_b = 25.7$ s).

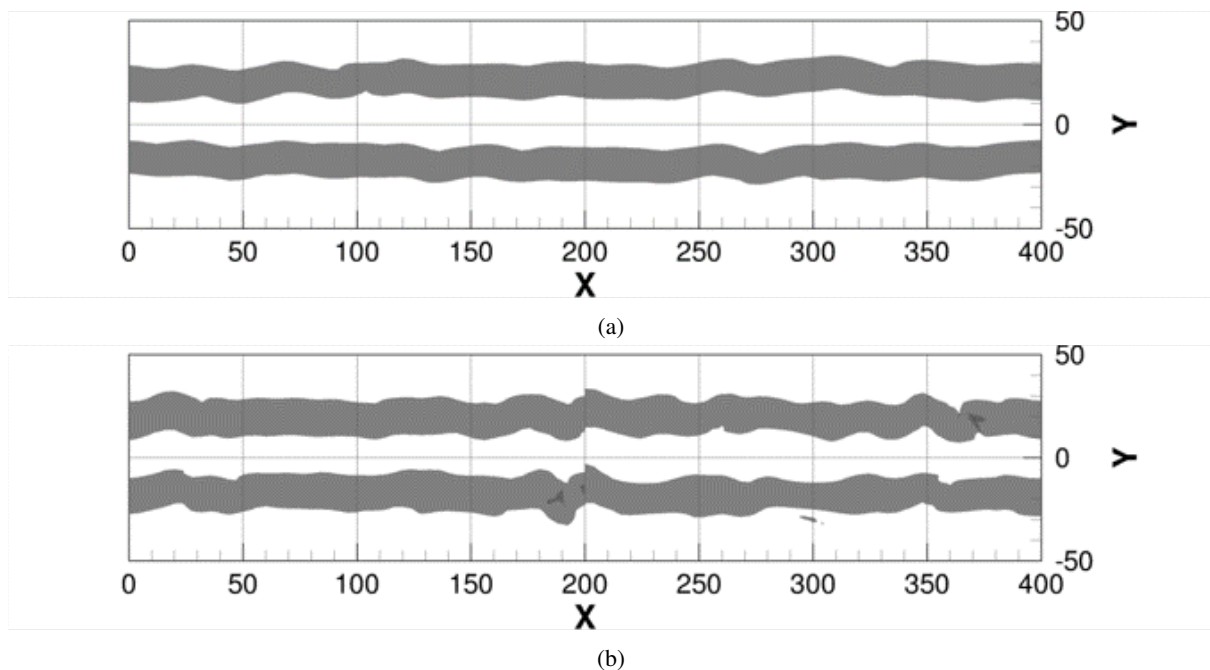

(a)

(b)

**Figure 18.** 2D view of $Q$-criterion isosurface $Q = 0.03\ \mathrm{s}^{-1}$ at $t/t_b = 3.89$ for (a) RANS initialization and (b) 4LO initialization. Distance are in meters. ($t_b = 25.7\ \mathrm{s}$).

ratio $r_c/b_0$ between the vortex core radius and the distance between the vortices, and two real constants. The instability bands for the first three modes are given by:

$$2.26 - 8.68 r_c^2/b_0^2 < k r_c < 2.26 + 8.68 r_c^2/b_0^2 \tag{20}$$

$$3.96 - 14.4 r_c^2/b_0^2 < k r_c < 3.96 + 14.4 r_c^2/b_0^2 \tag{21}$$

$$5.61 - 20.1 r_c^2/b_0^2 < k r_c < 5.61 + 20.1 r_c^2/b_0^2 \tag{22}$$

The numerical application for the case considered in this work gives $r_c^2/b_0^2 = 5.75 \times 10^{-3}$ at $t = 0$, leading to very narrow instability bands. Consequently, the absence of Widnall instabilities in the LES using a dipolar analytical initialization from two Lamb–Oseen vortices could be explained by the fact that the perturbation wavenumber lies outside these instability bands. The use of a more realistic synthetic turbulence technique in the wake if an analytical initialization strategy is used could potentially help circumvent that behavior.

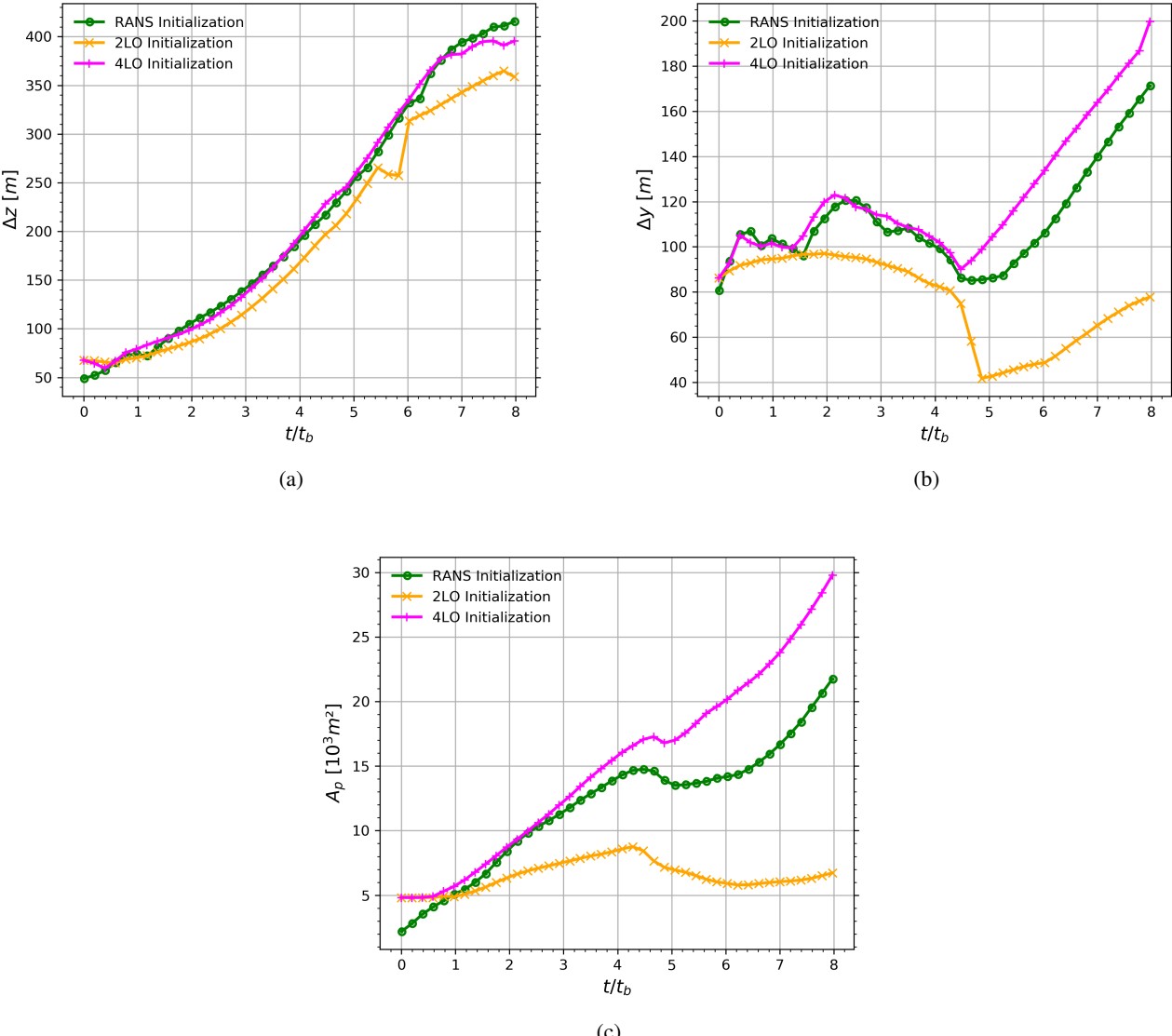

**Figure 19.** Contrail length scales for the different initialization strategies at $N_b = 0.012 \, \text{s}^{-1}$. (a) Contrail vertical extent, (b) contrail width and (c) area of contrail cross-section. ($t_b = 25.7 \, \text{s}$).

The temporal LES simulation performed by Holzäpfel et al. (2001) using a dipolar analytical initialization strategy displayed the Widnall instability on the vortex pair. The appearance of these instabilities can be explained by the synthetic turbulence approach used to model wake turbulence and atmospheric turbulence. First, wake turbulence was added directly in the vortices while atmospheric turbulence was added in the whole domain. Second, the corresponding synthetic wake turbulence fluctuations generated in the simulation by Holzäpfel et al. (2001) were distributed according to a normal distribution and are,

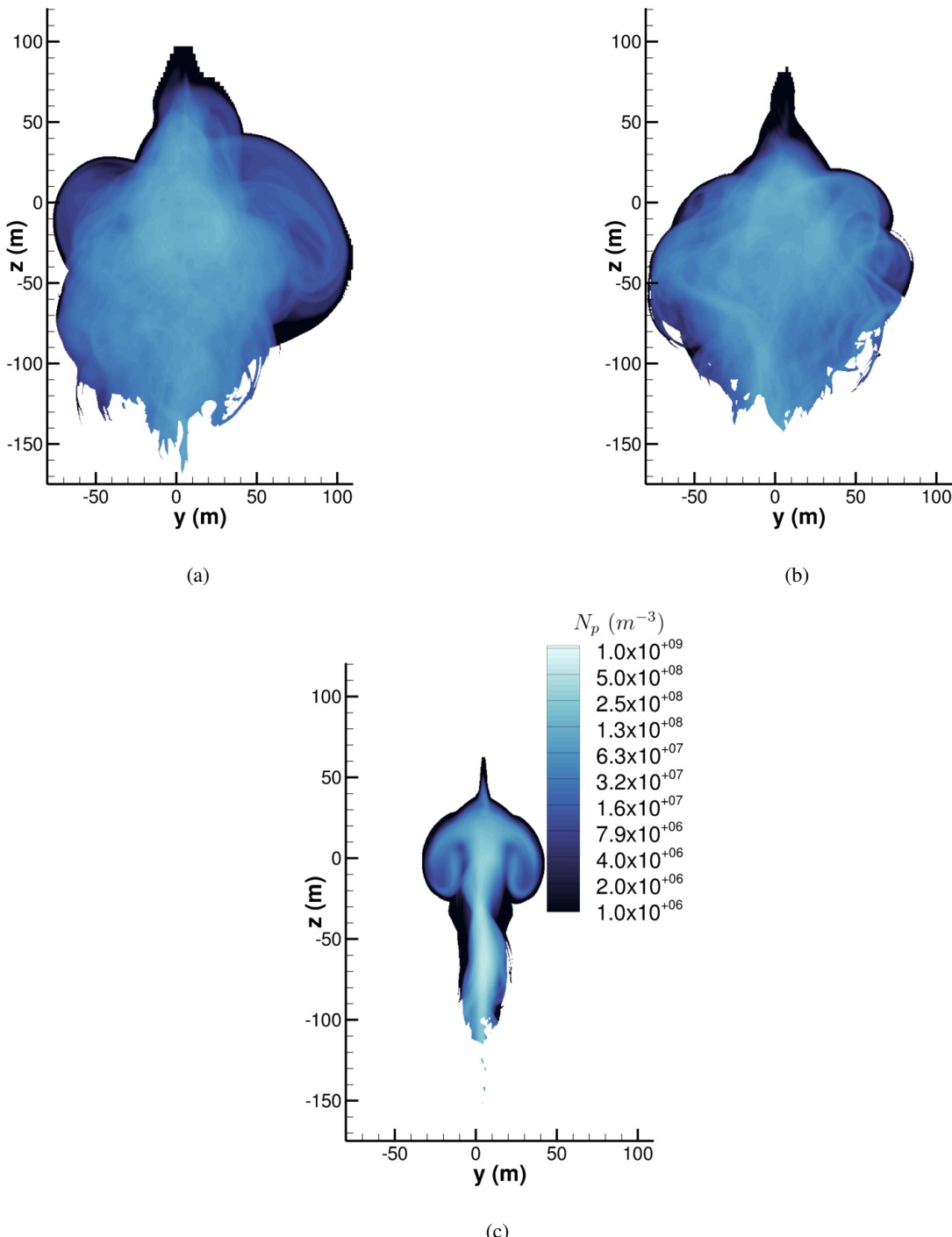

**Figure 20.** Spatially averaged ice crystal number concentration field $N_p$ in m$^{-3}$ at $t/t_b = 8$ ($t_b = 25.7$ s) and for $N_b = 0.012$ s$^{-1}$. Comparison between (a) *4LO*, (b) *RANS* and (c) *2LO initializations*. A logarithmic scale is used for the color levels.

therefore, not spatially correlated. As a result, their perturbation energy does not depend on their size and short-wave Widnall instability modes can potentially be excited with sufficient energy.

By contrast, the temporal LES simulation by Picot in Picot et al. (2014) also used a pair of Lamb–Oseen vortices to initialize the vortex regime but did not introduce synthetic wake turbulence inside the vortices. In addition, atmospheric turbulence was generated through an Ornstein-Uhlenbeck forcing technique, which resulted in correlated turbulence following a realistic turbulence spectrum in a stratified fluid. No short-wavelength instability was then observed on the vortices.

Not adding synthetic turbulence in the vortices for a far-field flow calculation can be justified by two main reasons. First,
the jet plume concentrates the vast majority of the turbulent kinetic energy. This is actually a consequence of the second point, which is that turbulence generated in a Lamb–Oseen vortex is rapidly destroyed in the jet regime because of rotation effects that were theoretically studied in Jacquin and Pantano (2002). Finally, in this work computations, the atmospheric turbulence is much closer to the one in Picot et al. (2014) work than the one from Holzäpfel et al. (2001) in terms of turbulent kinetic energy spectra and spatial correlations. This too may justify the absence of Windall instability in the 2LO case. Thus, it is believed
that it is not the Windall instability that is observed in the RANS and 4LO cases, but the shortwave instabilities resulting from the four-vortex nature of the wake (see Fabre et al. (2002)).

Fig.19 plots the relevant contrail dimensions as a function of time for the three different initialization strategies: RANS initialization, 2LO initialization and 4LO initialization. The corresponding spatially averaged cross-sections are plotted in Fig.20 for the ice crystal number concentration field. In terms of height and width, 4LO initialization gives a very similar
contrail in terms of size compared to the RANS initialization. The contrail has a larger cross-sectional area for the 4LO initialization but is still much closer to the RANS initialization cross-sectional area than the 2LO initialization.

The evolution of total microphysical quantities is plotted in Fig.21. The addition of the two HTP vortices in the analytical initialization strategy gives very similar results to the RANS initialization in terms of $m_{ice}$ until $t/t_b \approx 3.25$. At the end of the simulation, the 4LO initialization strategy gives the highest ice mass. More precisely, the contrail obtained with the 4LO
initialization is approximately six times heavier than the one obtained through RANS initialization. As the total ice crystal number per meter of flightpath $N_{ice}$ is very similar between the 2LO and 4LO initializations as observed in Fig.21b, the difference in ice mass results in larger ice crystals for the 4LO initialization. At the end of the LES simulation, ice crystals in the 4LO initialization are, on average, almost twice as large as those in the 2LO initialization, despite identical initial ice mass and total ice crystal number.

In the sensitivity study by Unterstrasser and Gierens (2010), doubling $m_{ice}$ while keeping the same initial number of ice crystals resulted in a $14\%$ increase in total extinction 5.5 hours after the formation of first ice crystals and with initial $N_{ice} = 6.8 \cdot 10^{12}$ m$^{-1}$. A tenfold increase in ice mass led to a $6\%$ decrease in total extinction for initial $N_{ice} = 3.4 \cdot 10^{13}$ m$^{-1}$. We can then reasonably expect the HTP vortices to have a non-negligible effect on the contrail-cirrus radiative properties in the diffusion regime.

In addition, we can also expect the difference in contrail size between the 2LO and 4LO initialization strategies to have a long-lasting impact on the contrail-cirrus in the diffusion regime. Indeed, assuming same initial value of $N_{ice}$ and $m_{ice}$, Unterstrasser and Görsch (2014) 2D simulations of the diffusion regime for different aircraft sizes measured a relative difference

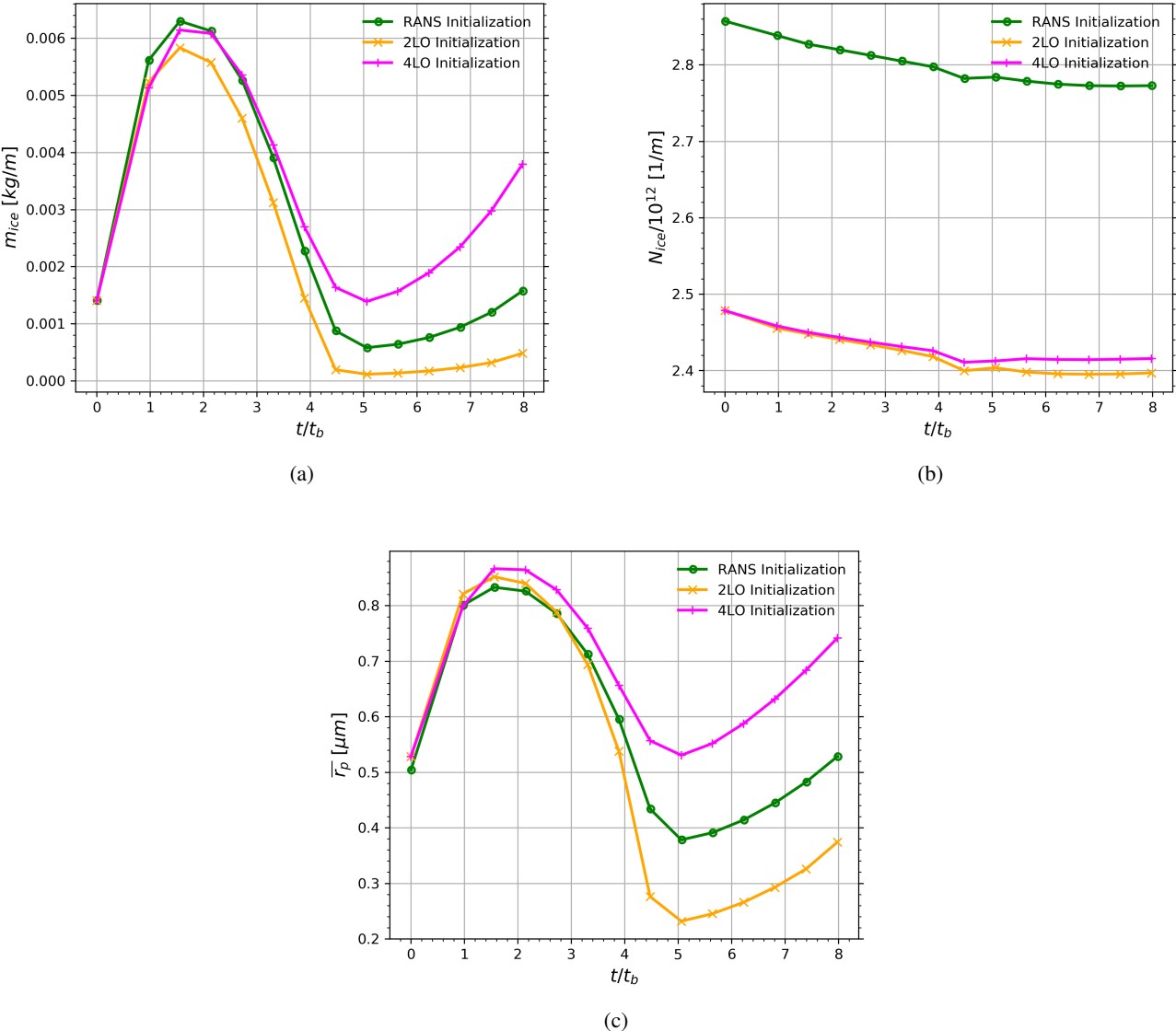

**Figure 21.** Evolution of contrail total quantities for the different LES initialization strategies at $N_b = 0.012\,\mathrm{s}^{-1}$.. (a) Total ice mass per meter of flightpath, (b) total ice crystal number per meter of flightpath and (c) mean ice crystal radius ($t_b = 25.7\,\mathrm{s}$).

of 7% (see Fig. 11 in Unterstrasser and Görsch (2014)) in terms of total extinction between a contrail produced by an Airbus A380 and a contrail produced by an Airbus A300 six hours after the formation of the first ice crystals. Setting the same initial

ice mass and ice crystal number allowed the study to isolate the effect of the contrail's initial size. Furthermore, the relative difference in total extinction was approximately 20% between an A300 and a CRJ200. However, it is important to note that in both Unterstrasser and Görsch (2014) and Unterstrasser and Gierens (2010) papers, initial value of $N_{\mathrm{ice}}$ constitutes the most

influential parameter to determine contrails radiative properties in the diffusion regime. In the study, assuming identical initial total ice mass, decreasing $N_{\text{ice}}$ by a factor 0.5 and 0.1 led to a total extinction decrease of $38\%$ and $78\%$, respectively.

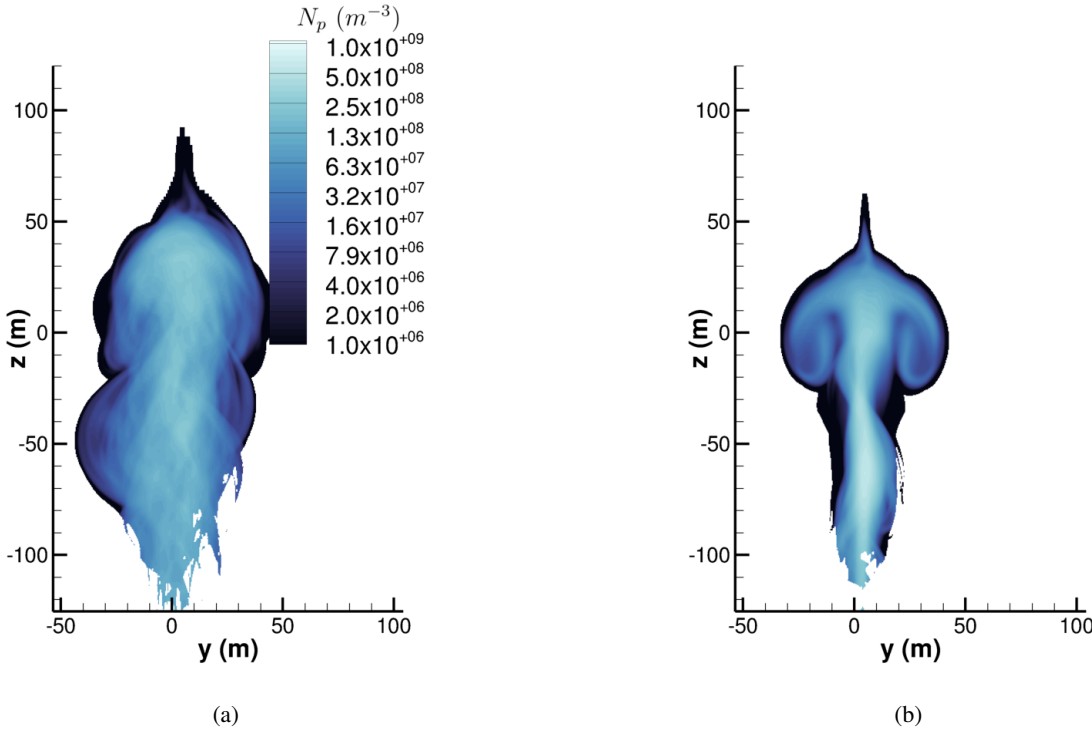

(a)                                                                 (b)

**Figure 22.** Spatially averaged ice crystal number concentration field $N_p$ in m$^{-3}$ at $t/t_b = 8$ ($t_b = 25.7$ s) and for $N_b = 0.012$ s$^{-1}$. Comparison between (a) *2LO initialization with RANS plume* and (b) *2LO initialization*. A logarithmic scale is used for the color levels.

### 4.4   Influence of plume initialization

As previously indicated, a final LES simulation is performed where the far-field calculation is initialized using two Lamb-Oseen vortices and the plume obtained from the RANS calculation instead of a circular plume as done classically (2LO initialization strategy). Such simulation allows us to investigate the effect of plume initialization for a contrail evolution. It will also enable a comparative analysis between the impact of plume initialization and the effect of vortex initialization, thereby elucidating their relative contributions to contrail development.

The ice crystal number concentration field on the slice obtained from the spatially averaged the contrail along flight direction is plotted in Fig.22 at $t/t_b = 8$ ($t = 200$ s) for the 2LO initialization and the 2LO initialization with RANS plume. Quantitative and qualitative differences can be observed. Indeed, the contrail is larger when the RANS plume is used instead of a simple circular plume. Moreover, a slightly smaller value for the ice crystal number concentration is obtained for the RANS plume. In other words, the contrail is more diluted when the plume is initialized through the RANS calculation of the near-field. However, it appears that the contrail vertical extent is very similar between the two cases. Initializing the plume through a circular jet or

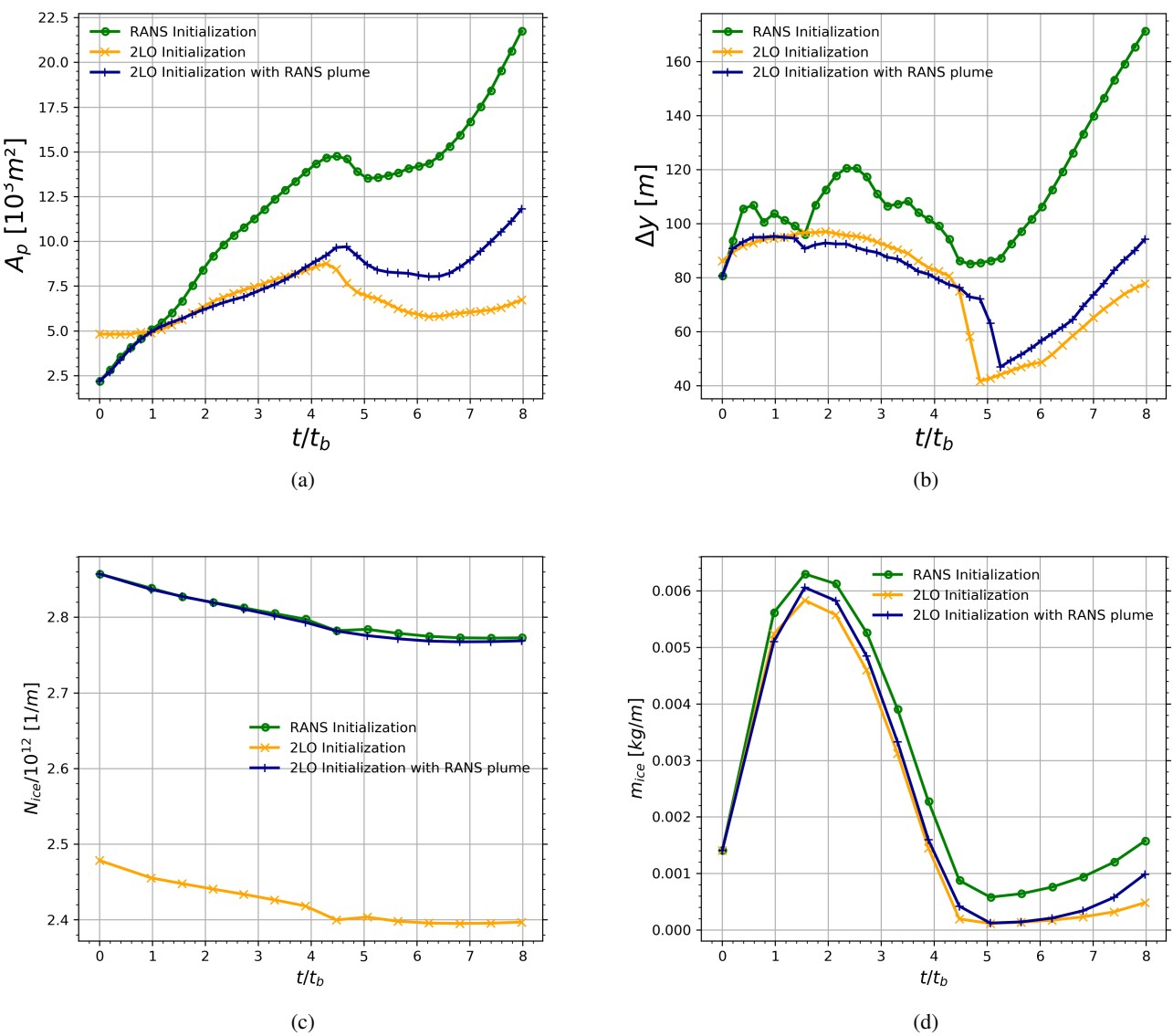

**Figure 23.** (a) Average contrail cross-sectional area, (b) average contrail width, (c) total ice mass per meter of flightpath and (d) total ice crystal number per meter of flightpath. Comparison between 2LO initialization, 2LO with RANS plume initialization and full RANS initialization strategies.

the deflected jet obtained from the RANS calculation has little effect on how far the ice crystals descent. More precisely, the vortex dynamics and the stratification exert a stronger influence than the choice of initializing the plume with a circular jet or with a more complex jet obtained from a precursor RANS simulation.

More quantitative comparisons can be made by plotting the evolution of the contrail spatial dimensions in the spatially averaged contrail, the contrail total ice mass per meter of flightpath and the contrail number of ice crystals per meter of flightpath. These quantities are plotted in Fig.23 for the 2LO initialization, 2LO initialization with RANS plume and the full RANS initialization which is our reference case. We observe that for the vortex regime, that is $t/t_b < 4$, the contrail cross-sectional area and the contrail width are very similar between the 2LO initialization and the 2LO initialization with RANS

plume despite the initial discrepancies. Some differences arise after the vortex regime. The contrail obtained with the 2LO initialization with the RANS plume is larger than the one obtained with a circular plume. However, the cross-sectional area and width evolution show very similar trends between the two initializations. Moreover, the differences observed are significantly smaller between the 2LO and the 2LO with RANS plume than between the 2LO and the full RANS initialization strategies, indicating a greater impact of the vortex initialization.

Concerning the evolution of $N_{ice}$, the comparison between the full RANS initialization and the 2LO initialization with RANS plume shows a negligible effect of the vortex initialization on $N_{ice}$. As for the evolution of $m_{ice}$, the effect of plume initialization using two Lamb-Oseen vortices to model the wake, which corresponds to the classical 2LO initialization, shows little effect for the vast majority of the simulated time.

  The temporal evolution of contrail cross-sectional area and width reveals that the 2LO initialization incorporating the RANS

plume consistently produces larger contrail dimensions compared to the classical 2LO initialization. A comparison of cross-sectional areas between the full RANS initialization and the 2LO initialization indicates that the influence of vortex initialization on contrail dimensions becomes apparent at $t/t_b = 1$. During the interval $t/t_b = 1$ to $t/t_b = 4.2$, the contrail spatial dimensions remain approximately equivalent between the 2LO initialization with RANS plume and the classical 2LO initialization. At later times, following vortex pair breakup induced by Crow instability, the contrail generated using the 2LO

initialization with RANS plume exhibits larger dimensions than that obtained with the classical 2LO initialization. Nevertheless, the discrepancy is considerably more pronounced when comparing the 2LO initialization with RANS plume to the full RANS initialization.

  In summary, vortex initialization exerts a more significant influence than plume initialization on total quantities and contrail spatial dimensions. However, it is important to note that, in light of the results obtained, the effects of plume initialization

cannot be neglected either.

## 5 Conclusions

This work extended the methodology of Bouhafid et al. (2024) to simulate the formation and early evolution of contrails for a realistic aircraft—from the formation of initial ice crystals to several minutes thereafter—without relying on analytical initialization classically used in the literature. Instead, we directly solved the near-field aerodynamic wake and contrail formation

using a RANS approach with a Reynolds-Stress turbulence model and then transitioned to far-field evolution with a temporal LES augmented by synthetic turbulence.

Applied to the B777-like CRM geometry under two stratification scenarios, our method revealed significant differences compared to classical analytical initialization using Lamb-Oseen vortices. In particular, RANS initialization produced a more turbulent contrail with greater size, larger ice crystals, and larger extinction. This behavior is attributed to short-wave instabilities inherent to a quadripolar vortex system in a counter-rotating configuration. A subsequent LES using four Lamb-Oseen vortices (representing two main wing-tip vortices and two HTP vortices) yielded contrail characteristics qualitatively similar to those from the RANS initialization. Moreover, a LES simulation was performed using two Lamb–Oseen vortices together with the plume obtained from the RANS calculation, and it was found that vortex initialization exerted a stronger influence on the early contrail evolution than plume initialization.

It is important to note that these results are valid only for recently formed contrails under moderately ice-supersaturated conditions. Future work should assess whether these differences persist during the diffusion regime using Paoli et al. (2017) methodology. The differences observed in the early part of a contrail life cycle may potentially decrease, or even vanish, in the diffusion regime as atmospheric processes become predominant. Moreover, for calculations of the diffusion regime, it will be essential to incorporate microphysical models that account for more accurate soot activation, freezing, ice particle inertia, background aerosol effects and most importantly the polydisperse nature of ice crystals size. Additionally, further investigation into the characteristics of HTP vortices and the impact of engine positioning (see Ramsay et al. (2024); Saulgeot et al. (2023)) on contrail development could inform more effective contrail mitigation strategies through optimized aircraft design.

*Author contributions.* **YB**: Writing – review & editing, Writing – original draft, Visualization, Validation, Methodology, Investigation, Formal analysis, Data curation, Conceptualization. **NB**: Writing – review & editing, Validation, Methodology, Supervision, Formal analysis, Conceptualization.

*Competing interests.* The authors declare that they have no known competing financial interests or personal relationships that could have appeared to influence the work reported in this paper.

*Acknowledgements.* This research has been supported by the French Ministère de la Transition Ecologique et Solidaire under the CLIMAVI-ATION programme (grant No. DGAC N2021-39), with support from France's Plan National de Relance et de Résilience (PNRR) and the European Union's NextGenerationEU.

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
