# Peer review of "Simulation of a contrail formation and early life cycle for a realistic airliner geometry"

_EGUsphere, 2025_

## Referee Comment (RC2)

Review of

**"Simulation of a contrail formation and early life cycle for a realistic airliner geometry"**

By Bouhafid and Bonne

**Summary**

This study present simulations of early contrails with an advanced CFD method. The paper is well-formulated and the results are clearly presented. Basically a few simulations with an high-fidelity dynamical approach are analysed and the impact on the type of wake vortex and exhaust plume initialization and on stratification are discussed.

**General comments**

The comparison between the RANS-based and analytical wake vortex initialization is in general well-described. I appreciate the efforts with the 4LO initialization that tries to mimic the RANS initialization by simpler means (future applications might use the 4LO initialization and do not require a-priori RANS simulations of the initial vortex roll-up phase).

We identified several major issues (mostly in the study design and analysis of the simulation data) that should be addressed in a revised version

1.    The analysis of the extinction and optical depth of young contrail may be misleading and misinterpreted by readers.
      a.        The contribution of the first few minutes of the full contrail lifecycle to the time-integrated radiative forcing (or extinction) is usually not substantial. A relatively larger extinction in the beginning does not imply a larger radiative impact at later times and in total.
      b.        Simulations in Unterstrasser & Gierens (2010b) and Lewellen (2014) show that the total number of ice crystals is the most crucial quantity of young contrails that determines the further fate. The early ice crystal mass (and also optical depth and integrated extinction) does not significantly affect the long-term behaviour of the contrail-cirrus transition. Hence, an evaluation of total ice crystal number would be more insightful.
      c.        Moreover, I strongly suggest to not use the term "radiative forcing of young contrails".
      First of all, radiative forcing is defined as a radiative imbalance typically evaluated at the top of the atmosphere and this is not what is evaluated in your study. You should make clearer, how to interpret the extinction quantity that you analysed.
      d.        In line 515, the conclusion states, e.g., "RANS initialization produced a more turbulent contrail with ... increased radiative impact". I think this formulation is too strong. Similar formulations appear in other locations as well.
2.    How robust are the evaluations of $\Delta z$ and $\Delta y$? In Eulerian models, contrails typically do not feature very strong gradients at the boundaries and fade out. Hence, the values you determined may depend on thresholds with which you define a contrail. I believe it would help readers to also show vertical and transvers profiles of ice crystal number

and mass. This would allow for a more quantitative comparison compared to Figs. 7, 8 and 12 and also makes clearer how robust the evaluations of $\Delta z$ and $\Delta y$ are.

3. How is the boundary of the contrail defined? Why do you choose to apply a weighting by number? Why at all and why not by ice mass e.g.? No spatial distributions of ice crystal number concentration are displayed. Nor the time evolution of total ice crystal number is shown. Hence, it is not transparent what effects the weighting in the averaging procedure does introduce.

4. Your analysis focuses on intensive mean quantities, which depend on the contrail-cross-section of the contrail. It would be interesting to also see integrated quantities like the total ice crystal number and mass (which do not depend on the definition of the contrail boundary).

5. Your interpretation of the simulation results focuses on the differences in the dynamical setup (RANS versus 2LO and 4LO). It is not much discussed that in addition the exhaust plumes are initialized in a different way. Currently you simply assume that observed differences in the simulation results are due to the different wake vortex initialization, but this is not really proven. It may help to perform another type simulation where the RANS with the idealised exhaust plume or the LO wake vortices are combined with the RANS exhaust plume. This way you could answer, which of the two aspects is more crucial.

6. Why do your simulations run up to t= 200s? Previous contrail vortex phase simulations considered a longer time period (5 minutes or longer). Have the vortices decayed after that time? Both $N_b$ values represent strongly to extremely stable conditions. Air masses that move downwards will rise when the vortices get weaker. Is this process already completed at t=200s? It would be interesting to see whether vertical profiles change more slowly after the vortices break up and buoyancy-driven air motions cease. Is this already the case after 200s?
The background of the question is that Fig.8 shows a strong difference after 200s between 'RANS' and '2LO'. It is not clear if this discrepancy is just a transient phenomenon and whether the difference between the two simulations is long-lasting (as the vertical distribution does not change much beyond t=200s).

7. Line 155: What do you mean with heterogeneous nucleation? The scientific consensus is that contrail formation occurs on condensation nuclei via heterogeneous droplet nucleation, with subsequent homogeneous freezing. Referring to heterogeneous nucleation may imply that soot particles act directly as ice nuclei (IN). In Equation (9), it appears that soot particles are assumed to become ice crystals immediately upon activation. Why is the liquid phase and the freezing process not explicitly represented in your model?
Another manuscript in review for ACP by Ponsonby et al. ([https://doi.org/10.5194/egusphere-2025-1717](https://doi.org/10.5194/egusphere-2025-1717)) states: "To that end, several LES models prescribe water saturation as the critical condition for contrail ice formation (Paoli et al., 2013; Picot et al., 2015) or heterogeneous ice nucleation as the primary formation pathway (Khou et al., 2017, 2015), both of which have been rejected by in-situ observations (Kärcher et al., 2015). More representative microphysical treatment can be achieved using 0D box- and parcel model simulations (Bier et al., 2022; Rojo et al., 2015; Yu et al., 2024). Here, the dilution of a parcel of exhaust air is simulated and microphysical phase transitions such as particle activation and homogeneous ice nucleation are tracked. While these models are unable to incorporate feedback between different plume parcels, which may otherwise lead to a diversity of particle history

(Lewellen, 2020), they are configured for sophisticated treatment of complex ice microphysics, which is critical for describing contrail properties (Yu et al., 2024)."

**Specific comments**

- o  Line 2: The accumulated CO2 emissions by aviation
- o  Line 12: Make clear that this statement holds only for young contrails. Emphasize that further research for long-lived contrails is needed.
- o  Line 16: replace nucleation by condensation
- o  Line 44: The ice crystals do not heat up adiabatically. The surrounding air does so with implication on the relative humidity and ice crystal growth.
- o  Line 50: Do those vortex rings always form?
- o  Line 52: I would prefer to reformulate to something like "Contrail evolution is driven *or governed* by physical *processes* (not by conditions) which are affected by *(conditions like)* wind shear, stratification etc.
- o  Line 55: Lewellen (2014) is also a great source of contrail-cirrus analyses.
- o  Line 117: to form ice crystals, condensation is not enough, you should explicitly mention also the freezing process.
- o  Line 126-127: already stated in line 115
- o  Inclusion of Eqs. (7) -(9) into the text would facilitate reading.
- o  Eqs. 2, 3 and 6 do not convey a lot of information. With which rates do those conversions occur? Would it be more informative to write the equations for the mass fractions of all or selected quantities?
- o  Section 3.2 does not explicitly mention how water vapour is initialized. The sentence in line 281 may imply that absolute water vapour mixing ratio is held constant with altitude. Most other studies of the contrail vortex phase kept the relative humidity constant. Could you plot RHi(z) to see how much this quantity changes with altitude?
- o  Fig. 9: the contrail height and width evolution of the RANS case with $N_b = 0.03$ s$_{-1}$ looks a bit strange in the sense, that at t=4.5 the height suddenly stops to increase (which might be linked to vortex break up) and width increases. What process leads to such a large change in the width increase?
- o  Section 4.1. It would be interesting to also see the time evolution of total ice crystal number I and mass M and possibly also of the ratio Nice/Ns. $M_{ice}$ and $N_{ice}$ are more straightforward to analyse and interpret than the derived mean radius ~ $(M_{ice}/N_{ice})^{1/3}$.
  Computing the mean radius via $M_{ice}$ and $N_{ice}$ is probably better than evaluating $r_p$ in each grid cell and do a number-weighted average. How much do the computed values differ between the two formulas?
  The formula in line 325 might be interpreted in a way that $r_p$ depends only on IWC. I would prefer to include $N_{ice}$ in the formula.
- o  Lin330: If all soot particles are activated and more ice crystals are present, then they should be smaller not larger if ice mass is similar.
- o  Line 361: It is not clear, if the scaling of time is applied in the model or only in the presentation of the results. What's the advantage of using a normalized time coordinate in this study? All setups have the same $t_0$ and there is no benefit of normalization to make results better comparable. If you keep the normalized values, it would help the readers to add the $t_0$ value to the figure captions.

- o Fig.7. shows soot number density which is not an actual contrail property. Wouldn't it be more logical to show contrail ice crystal number or mass concentration?
- o Line 374: this sounds like a general statement about contrail formation in strongly stable conditions. But I guess it only relates to your choice of water vapour initialization. Do you mean the actual formation process or the time evolution over 200s?
- o Fig.11: Trends are very similar. Do you expect a long-lasting impact?
- o Eq. 19: I believe in the contrail community, the quantity defined in Eq. 19 was first introduced in Unterstrasser & Gierens (2010a) and named total extinction.
- o Line 439: Yes, that is the important point. I would appreciate to see this statement also in the abstract.
- o Line 504: The statement is too strong for reasons stated above.

**Technical corrections**

- o Figure 1: replace $M_{infty}$ by $U_{infty}$
- o Line 21: estimation of effective radiative ERF of contrails and other forcing agents.
- o Line 24: Is controversial the correct word here? What fact is controversial?
- o Lines 103 and 195: "negelected before" does not sound like proper English
- o Line 114 "water vapour H2O" = "gas-phase H20"? Similary, ice-phase $H20_s$
- o Line 155: Contrail (without 's') ice
- o Section 3.2 should state the time period that is simulated.
- o Line 301: weighted
- o Line 330: soot PARTICLES
- o Figure 5 and Figure 10: soot number density is named $N_s$ and $N_p$. Please be consistent.
- o Figure 7: The $r_p$ values on the colour bar should have nice values. Is a linear or logarithmic scale used?
- o Line 378 Widnall
- o Line 382: length scales
- o Line 395: descend
- o Line 395: 'contrail surface reduction': do you mean a decrease of the contrail cross-sectional area?

**References**

Lewellen, D. C.: Persistent contrails and contrail cirrus. Part 2: Full Lifetime Behavior, J. Atmos. Sci., 71, 4420–4438, doi:10.1175/JAS-D-13-0317.1, 2014.

Unterstrasser, S. and Gierens, K.: Numerical simulations of contrail-to-cirrus transition – Part 1: An extensive parametric study, Atmos. Chem. Phys., 10, 2017–2036, doi:10.5194/acp-10- 2017-2010, 2010a.

Unterstrasser, S. and Gierens, K.: Numerical simulations of contrail-to-cirrus transition – Part 2: Impact of initial ice crystal number, radiation, stratification, secondary nucleation and layer depth, Atmos. Chem. Phys., 10, 2037–2051, doi: 10.5194/acp-10-2037-2010, 2010b.

---

## Author Comment (AC1)

We wish first to thank you very much for your important and relevant comments. You will find below an answer to all the comments and the corrections we propose in the paper.

Note: Corrections from your comments are highlighted in blue color in the paper.

**GENERAL COMMENTS :**

**Reviewer comment:** *The analysis of the extinction and optical depth of young contrail may be misleading and misinterpreted by readers.*
*a.*
*The contribution of the first few minutes of the full contrail lifecycle to the time-integrated radiative forcing (or extinction) is usually not substantial. A relatively*
*larger extinction in the beginning does not imply a larger radiative impact at later*
*times and in total.*
**Authors answer:** Thank you for this valuable comment. We agree that the early contrail radiative forcing is not relevant for the climatic impact of a contrail. And that an initial larger extinction does not imply a larger radiative impact several hours later. In order to avoid misinterpretation the following warning has been added **line 487**:

Indeed, the differences in extinction observed for the first few minutes may potentially decrease, or even vanish, over longer timescales owing to the effect of atmospheric turbulence and wind shear on the ice crystals spatial distribution. A larger extinction in the beginning does not necessarily imply a larger radiative impact at later times and over the full lifetime of the contrail.

**Reviewer comment:** *Simulations in Unterstrasser & Gierens (2010b) and Lewellen (2014) show that the total number of ice crystals is the most crucial quantity of young contrails that determines the further fate. The early ice crystal mass (and also optical depth and integrated extinction) does not significantly affect the long-term behaviour of the contrail-cirrus transition. Hence, an evaluation of total ice crystal number would be more insightful.*

**Authors answer:** We give the evolution of averaged ice crystal number (Np) in the contrail as a function of time (Fig.7, Fig. 15, Fig.20 and Fig.24). We believe this gives an insight on the number of ice crystals in the domain.

**Reviewer comment:** *Moreover, I strongly suggest to not use the term "radiative forcing of young contrails". First of all, radiative forcing is defined as a radiative imbalance typically evaluated at the top of the atmosphere and this is not what is evaluated in your study. You should make clearer, how to interpret the extinction quantity that you analysed.*
**Authors answer:** "Young contrail" has been replaced by "Recently formed contrail" and "Radiative forcing" (RF) by "extinction" except for the reference to Ferreira et al. work where an RF parametrization has been used to estimate RF for a recently formed contrail. However, it is true that we cannot extrapolate the results of our simulations to estimate RF a few hours after the end of the dissipation phase, that is in the diffusion phase.

**Reviewer comment:** *In line 515, the conclusion states, e.g., "RANS initialization produced a more turbulent contrail with ... increased radiative impact". I think this formulation is too strong. Similar formulations appear in other locations as well.*
**Authors answer:** Radiative impact has been replaced by extinction and/or radiative properties. It is true that we cannot deduce radiative impact of recently formed contrails solely from the extinction as extinction takes into account both light scattering and light absorption by the ice crystals.

**Reviewer comment:** *How robust are the evaluations of Δz and Δy? In Eulerian models, contrails typically do not feature very strong gradients at the boundaries and fade out. Hence, the values you determined may depend on thresholds with which you define a contrail. I believe it would help readers to also show vertical and transvers profiles of ice crystal number and mass. This would allow for a more quantitative comparison compared to Figs. 7, 8 and 12 and also makes clearer how robust the evaluations of Δz and Δy are.*
**Authors answer:** Yes, it depends on the treshhold but the goal here was to compare the different initialization strategies results using the same treshhold. Spatially averaged ice crystal numbers field 2D contours have

been added to the paper (Fig.12, Fig.13, Fig.21, Fig.23) to have a better understanding of contrail size.

**Reviewer comment:** *How is the boundary of the contrail defined? Why do you choose to apply a weighting by number? Why at all and why not by ice mass e.g.? No spatial distributions of ice crystal number concentration are displayed. Nor the time evolution of total ice crystal number is shown. Hence, it is not transparent what effects the weighting in the averaging procedure does introduce.*

**Authors answer:** The contrail is defined by the mesh cells where the ice crystal radius rp is strictly greater than the radius of a dry soot particle rs (27 nm). This is now stated in the paper. Thus, only particles with an ice cap are considered. Those particles are by definition ice crystals. We applied a weighting by number of ice crystals because it adequately represents the influence of each ice crystals on the contrail mean quantities. For X a microphysical quantity, each value X contributes to the average proportionally to how many particles have that value. This is exactly the same as weighting by the number of ice crystals. Such weighting is commonly used in statistical physics. If we weighted by ice mass, we could have situations where a cell have a high ice mass but not that many ice crystals, which would bias the computed mean.

**Reviewer comment:** *Your analysis focuses on intensive mean quantities, which depend on the contrail-cross-section of the contrail. It would be interesting to also see integrated quantities like the total ice crystal number and mass (which do not depend on the definition of the contrail boundary).*

**Authors answer:** We believe that the definition of the contrail boundary by rp>rs is valid enough to define the contrail boundary with no ambiguity. As mentioned in the previous answer, in our model if a particle radius is less or equal to its core soot radius, it is not an ice crystal. Therefore averaging on every cell where rp>rs consider all of the ice crystals in the computational domain.

**Reviewer comment:** *Your interpretation of the simulation results focuses on the differences in the dynamical setup (RANS versus 2LO and 4LO). It is not much discussed that in addition the exhaust plumes are initialized in a different way. Currently you simply assume that observed differences in the simulation results are due to the different wake*

*vortex initialization, but this is not really proven. It may help to perform another type simulation where the RANS with the idealised exhaust plume or the LO wake vortices are combined with the RANS exhaust plume. This way you could answer, which of the two aspects is more crucial.*

**Authors answer:** We completely agree with this comment. Another simulation with the two Lamb-Oseen vortices but this time with the RANS exhaust plume was performed. The results are shown and discussed in the new section 4.4. It is found that vortex initialization influences the contrail evolution more than plume initialization, at least for the conditions considered in this work.

**Reviewer comment:** *Why do your simulations run up to t= 200s? Previous contrail vortex phase simulations considered a longer time period (5 minutes or longer). Have the vortices decayed after that time? Both Nb values represent strongly to extremely stable conditions. Air masses that move downwards will rise when the vortices get weaker. Is this process already completed at t=200s? It would be interesting to see whether vertical profiles change more slowly after the vortices break up and buoyancy-driven air motions cease. Is this already the case after 200s? The background of the question is that Fig.8 shows a strong difference after 200s between 'RANS' and '2LO'. It is not clear if this discrepancy is just a transient phenomenon and whether the difference between the two simulations is long-lasting (as the vertical distribution does not change much beyond t=200s).*

**Authors answer:** The simulations run up to t=200 s because this corresponds approximately to eight characteristic timescales tb for the vortex pair descent. Crow instability (see Crow [1] and Sarpkaya [2] work) destroys the vortex pair between 1 and 8 tb, the exact value depending on the atmospheric turbulence intensity and atmospheric stratification. For our simulations, the vortex pair destruction happens around t=4.5tb for the medium stratification scenario and at t~3tb for the strong stratification scenarios. After the vortex pair destruction, the influence of the aircraft aerodynamic wake begins to decrease. At t=200 s, most of the secondary wake has reached flight altitude and beyond. This statement can be supported by the fact that the contrail height increase is slowing down in the vortex phase, even reaching a plateau for strong stratification case. Therefore, 200 s of simulations is enough to observe and comprehend the impact of the initialization strategy on the LES. Whether the strong difference between "RANS" and "2LO" is a transient phenomenon is an

open question. But, considering the shapes of the curves in Fig14, the observed differences are very unlikely to fade out at 300s if it is a transient phenomenon. If it is transient, we believe that we will observe the merging of the curves in the diffusion regime, which is beyond the scope of this study. We believe that the contrail size difference will remain or initially increase with the addition of wind shear but that both contrails might eventually be of comparable size after a few hours, meaning that it could have an influence on its climatic impact.

**Reviewer comment:** *Line 155: What do you mean with heterogeneous nucleation? The scientific consensus is that contrail formation occurs on condensation nuclei via heterogeneous droplet nucleation, with subsequent homogeneous freezing. Referring to heterogeneous nucleation may imply that soot particles act directly as ice nuclei (IN). In Equation (9), it appears that soot particles are assumed to become ice crystals immediately upon activation. Why is the liquid phase and the freezing process not explicitly represented in your model? Another manuscript in review for ACP by Ponsonby et al. (https://doi.org/10.5194/egusphere-2025-1717) states: "To that end, several LES models prescribe water saturation as the critical condition for contrail ice formation (Paoli et al., 2013; Picot et al., 2015) or heterogeneous ice nucleation as the primary formation pathway (Khou et al., 2017, 2015), both of which have been rejected by in- situ observations (Kärcher et al., 2015). More representative microphysical treatment can be achieved using 0D box- and parcel model simulations (Bier et al., 2022; Rojo et al., 2015; Yu et al., 2024). Here, the dilution of a parcel of exhaust air is simulated and microphysical phase transitions such as particle activation and homogeneous ice nucleation are tracked. While these models are unable to incorporate feedback between different plume parcels, which may otherwise lead to a diversity of particle history(Lewellen, 2020), they are configured for sophisticated treatment of complex ice microphysics, which is critical for describing contrail properties (Yu et al., 2024)."*
**Authors answer:** By heterogeneous nucleation we mean condensation of water vapor around a nucleation site (soot particles in the case of this study). We agree that the transformation of the liquid water on soot to ice is achieved with homogeneous freezing. We disagree with the mentioned paper on that point, that is Khou et al. [3] model not explicitly solving the

freezing as you mentioned it. This hypothesis has been made mostly because of the monodisperse hypothesis, which guarantee that, considering the low ambient temperature, every particles will become ice crystals. If we admit that the microphysical scheme needs some refinement for the ice crystal refinement, once ice crystals are formed, we have the usual condensation/sublimation equation. Therefore, the vortex phase results still yields.

**SPECIFIC COMMENTS :**

Reviewer comments are in *italic*.

- *Line 2: The accumulated CO2 emissions by aviation*
→ It has been changed accordingly

- Line 12 : *Make clear that this statement holds only for young contrails. Emphasize that further research for long-lived contrails is needed:*
→ modification in **line 15** where it is emphasized that those results only hold in the first few minutes after ice crystals formation.

- Line 16 : *replace nucleation by condensation:*
→ done

- Line 44 :*The ice crystals do not heat up adiabatically. The surrounding air does so:*
→ adiabatic heating replaced by heating (**line 49**)

- *Line 50: Do those vortex rings always form?*
→ If Crow instability is triggered and is responsible for the vortex pair destruction, vortex rings will form. For strong stratification, other instabilities might destroy the vortex pair before Crow instability. If that occurs, no rings will be formed. This has been clarified in **line 56**.

- Line 52 : *I would prefer to reformulate to something like "Contrail evolution is driven or governed by physical processes (not by conditions) which are affected by (conditions like) wind shear, stratification etc.*

→ Modifications implemented accordingly.

- Line 55 : *Lewellen (2014) is also a great source of contrail-cirrus analyses.:*
→ Yes, it is now cited (**line 62**)

- Line 117 : *to form ice crystals, condensation is not enough, you should explicitly mention also the freezing process. :*
→ Yes, precision added in **line 127**.

- Line 126 : *already stated in line 115:*
→ Yes, the text has been modified accordingly (**line 135** crossed out).

- *Inclusion of Eqs. (7) -(9) into the text would facilitate reading :*
→ We have tried to do so but we found the text easier to read the way it was so we did not change it in the end.

- *Eqs. 2, 3 and 6 do not convey a lot of information. With which rates do those conversions occur? Would it be more informative to write the equations for the mass fractions of all or selected quantities? :*
→ The mass production rates are now given in Eq.2 and Eq.4.

- *Section 3.2 does not explicitly mention how water vapour is initialized. The sentence in line 281 may imply that absolute water vapour mixing ratio is held constant with altitude. Most other studies of the contrail vortex phase kept the relative humidity constant. Could you plot RHi(z) to see how much this quantity changes with altitude? :*
→ Ambient water vapour is initially constant in the computational domain. We decided to keep ambient water vapour constant instead of RH after informal discussions with climate scientist colleagues. ISSR measurements are needed to accurately define RH and water vapour profiles in the tropopause. RH profiles are now given in in Fig. 5 for the two stratification scenarios. With this hyphesis very high values of RH are reached at the bottom of domain in the strong stratification case. However, no ice crystals descend at such altitude. This point has been developed clarified in **line 301**.

*- Fig. 9: the contrail height and width evolution of the RANS case with Nb =0.03 s^-1 looks a bit strange in the sense, that at t=4.5 the height suddenly stops to increase (which might be linked to vortex break up) and width increases. What process leads to such a large change in the width increase?*

→ Height stopping to increase is due to vortex break up which happens sooner for Nb=0,03 s^-1. The width increase is most likely due to the strongly turbulent nature of the secondary wake that will mix the ice crystals with the ambient air way more efficiently. This has been clarified in **line 440**.

*- Section 4.1. It would be interesting to also see the time evolution of total ice crystal number I and mass M and possibly also of the ratio Nice/Ns. M ice and N ice are more straightforward to analyse and interpret than the derived mean radius ~(M ice /N ice ) 1/3 . Computing the mean radius via M ice and N ice is probably better than evaluating rp in each grid cell and do a number-weighted average. How much do the computed values differ between the two formulas? The formula in line 325 might be interpreted in a way that rp depends only on IWC. I would prefer to include N ice in the formula*

→ We have added the evolution of total ice mass and total ice crystal number in Fig .9. Concerning the mean ice crystal radius, we are not sure to understand your comment. We believe that knowing the ice crystal radius in each cell of the contrail and doing a weighted average gives a good overview of the ice crystals size in the contrail. rp formula as a function of IWC and N_ice is given in Eq.7.

*-Line 330: If all soot particles are activated and more ice crystals are present, then they should be smaller not larger if ice mass is similar.*

→ This is true but this effect is not taken into account in our Eulerian model. More precisely, our model considers that soot surface activation is done only with sulfur compounds. Ice production term in Eq.4 is directly proportional to activation fraction. Consequently, the higher the activation fraction, the higher the ice production and the higher the ice crystal radius. This represents a limitation of our model, as it should also consider soot activation caused by the ice cap formed on soot particles, and not only activation due to sulfur particles. This point has now been clarified in **line 355.**

*- Line 361: It is not clear, if the scaling of time is applied in the model or only in the presentation of the results. What's the advantage of using a normalized time coordinate in this study? All setups have the same t0 and there is no benefit of normalization to make results better comparable. If you keep the normalized values, it would help the readers to add the t0 value to the figure captions.*

→ The scaling of time is only applied in the results. Normalizing by Crow characteristic time is justified by the fact that vortex pair lifespan (and thus vortex influence on contrails) depends mainly on Crow characteristic time. A medium-range aircraft and long-range aircraft will have a vortex pair lifespan typically between 4 and 8 Crow characteristic times, even though in seconds the lifespan will be very different. However, this distinction is mainly relevant in an aerodynamic context and not necessarily when focusing on ice microphysics. The value of t0 in seconds has now been added to the figure captions."

*- Fig.7. shows soot number density which is not an actual contrail property. Wouldn't it be more logical to show contrail ice crystal number or mass concentration?*

→ Using soot number density is justified as it indicates which parts of the contrail have sublimated and which have reached flight altitude, where ice can form. In the medium stratification scenario, it shows that most soot particles remain without ice at the end of the simulation.

*- Line 374: this sounds like a general statement about contrail formation in strongly stable conditions. But I guess it only relates to your choice of water vapour initialization. Do you mean the actual formation process or the time evolution over 200s?*

→ Yes, the results are only valid for the water vapor mass fraction at flight altitude chosen for this study. We mean time evolution over 200 s because RH is the same at flight altitude for medium and strong stratification scenarios and thus the formation conditions are the same. "Formation" has been replaced by "persistence" in **line 411**.

*- Fig.11: Trends are very similar. Do you expect a long-lasting impact?*

→ That's a good question and we don't really have any clear answer. We can imagine that because of the initial difference in contrail width, wind shear and atmospheric turbulence might maintain the differences.

*- Eq. 19: I believe in the contrail community, the quantity defined in Eq. 19 was first introduced in Unterstrasser & Gierens (2010a) and named total extinction.*
→ It has been cited accordingly.

*-Line 439: Yes, that is the important point. I would appreciate to see this statement also in the abstract.*
→ This has been added in the abstract in **line 16**.

*- Line 504: The statement is too strong for reasons stated above.*
→ The LES simulation with the two Lamb-Oseen vortices and the RANS plume still shows that vortex initialization has a greater impact than plume initialization. For the flight regime studied, initializing the LES with four vortices is more critical that initializing the LES with the more complex RANS plume. However, it is important to note the strength of the short-wave instability will depend on the circulation ratio between the circulation of the main wing-tip vortex and the circulation of the horizontal tailplane vortex (ratio that will depend on the aircraft trim), and the ratio of the distances between the vortices center and the symmetry plane [4]. The importance of plume initialization might be greater for other flight conditions.

**TECHNICAL CORRECTIONS :**

*- Figure 1: replace M infty by U infty*
→ We disagree, aircraft speed in cruise flight is commonly given in Mach number.

*- Line 21: estimation of effective radiative ERF of contrails and other forcing agents.*
→ Correction added.

*- Line 24: Is controversial the correct word here? What fact is controversial?*
→ We replaced controversial with « currently debated » (see **line 29)**.

*- Lines 103 and 195: "negelected before" does not sound like proper English*
→ « neglected before » replaced by « neglected in comparison to»

*- Line 114 "water vapour H2O" = "gas-phase H20"? Similary, ice-phase H20 s*

→ This has been clarified in **line 122**.

*- Line 155: Contrail (without 's') ice*

→ Correction added

*- Section 3.2 should state the time period that is simulated.*

→ Time period added in **line 272**.

*- Line 301: weighted*

→ Correction added

*- Line 330: soot PARTICLES*

→ Correction added

*- Figure 5 and Figure 10: soot number density is named Ns and Np . Please be consistent.*

→ You are right. Ns is soot number density and Np is ice crystal number and they are not exactly the same quantity. Np is equal to Ns but only in the mesh cells which contains ice, that is when rp is strictly greater than rs. This has been clarified in the paper in **line 420**. Actually, it is the ice crystal number that is plotted in Fig.7.a and not the soot number density since we only consider the cells very particle radius is strictly greater than soot dry particle radius. This has been corrected in the paper.

*- Figure 7: The rp values on the colour bar should have nice values. Is a linear or logarithmic scale used?*

→ The values are now nicer and it has been clarified in the figure caption that a logarithmic scale is used.

*- Line 378 Widnall*

→ Correction added

*- Line 382: length scales*

→ Correction added

*- Line 395: descend*

→ Correction added

*- Line 395: 'contrail surface reduction': do you mean a decrease of the contrail cross-sectional area?*
→ Yes. It has been clarified in the text in **line 438**.

**References :**

[1] Crow, S. C., & Bate Jr, E. R. (1976). Lifespan of trailing vortices in a turbulent atmosphere. *Journal of Aircraft*, *13*(7), 476-482.
[2] Sarpkaya, T. (1998). Decay of wake vortices of large aircraft. *AIAA journal*, *36*(9), 1671-1679.
[3] Khou, J. C., Ghedhaifi, W., Vancassel, X., & Garnier, F. (2015). Spatial simulation of contrail formation in near-field of commercial aircraft. *Journal of Aircraft*, *52*(6), 1927-1938.
[4] Fabre, D., Jacquin, L., & Loof, A. (2002). Optimal perturbations in a four-vortex aircraft wake in counter-rotating configuration. *Journal of Fluid Mechanics*, *451*, 319-328.

---

## Author Comment (AC2)

Thank you for your valuable comments ! You will find below an answer to all the comments and the corrections we propose in the paper.

Note: Corrections from your comments are highlighted in red color in the paper.

**Reviewer comment :** Line 37 - an addition of a value to define soot-rich conditions? This would also make clear what constitutes soot-poor in the next sentence.
**Authors answer :** A definition of soot-rich conditions has been added in **line 42**. The soot emission index for our case has been computed and given in **line 245**.

**Reviewer comment :** The model references that the engine is representative of an UHBR turbofan, however the water vapour emissions used (and hence in effect, estimated fuel flow) are from an RB211. This is an older, much lower BPR engine. It does not take away from the results and modelling, but perhaps more representative boundary condition to an UHBR could be used within future work.
**Authors answer :** This is correct. This has been clarified in **line 248**

**Reviewer comment :** Figure 2 could be reduced in size, and/or added alongside it, images of the mesh (and result of mesh adaption process) downstream could be shown to understand how refined RANS grids must be.
**Authors answer :** Images of the mesh downstream are available in our paper [1] concerning the aerodynamic analysis of the CRM wake for the exact same conditions (Fig 11 in [1]). Size of Fig.2 has been reduced.

**Reviewer comment :** The water vapour and soot particle emission number could also be added in Table 1 for engine boundary conditions. This would allow for results to replicated and compared more accurately within the community. This would be beneficial whilst experimental measurements remain scarce.

**Authors answer :** Thank you for this very important comment. A new table (table 2) has been added and gives the values of mass fractions used in the calculation.

**Reviewer comment** : Line 329 - would assuming that all soot particles were activated not lead to smaller ice crystals, as more particles compete for the same water vapour?

**Authors answer :** This is true but this effect is not taken into account in our Eulerian model. More precisely, our model considers that soot surface activation is done only with sulfur compounds. Ice production term in Eq.7 is directly proportional to activation fraction. Consequently, the higher the activation fraction, the higher the ice production and the higher the ice crystal radius. This represents a

limitation of our model, as it should also consider soot activation caused by the ice cap formed on soot particles, and not only activation due to sulfur particles. This point has now been clarified in **line 355**.

**Reviewer comment** :In abstract/introduction, CO2 and NOx do not use subscripts. In 2.2 Microphysical Model, the chemical formulas do use subscripts. To be consistent, should CO2 and NOx use subscripts too? (i.e. $CO_2$, $NO_x$)

**Authors answer :** This has been corrected

**Reviewer comment :** Line 24 - can use ERF instead of 'effective radiative forcing' as acronym just introduced.
**Authors answer :** This has been corrected

**Reviewer comment :**Line 76 - Use of 'us' – 'This allowed the full… …to be taken into account' would be less ambiguous.

**Authors answer :** This has been corrected

**Reviewer comment :** Line 224 - Are these values defined within a table? (i.e. Tab. ??)

**Authors answer :** Values are now defined in Table 2.

**Reviewer comment :**Line 232 - Need to update table reference (Tab. ??)

**Authors answer :** Values are now defined in Table 2.

References :
[1] Bouhafid, Y., Bonne, N., & Jacquin, L. (2024). Combined Reynolds-averaged Navier-Stokes/Large-Eddy Simulations for an aircraft wake until dissipation regime. *Aerospace Science and Technology*, *154*, 109512.

---

## Referee Report (RR1)

**Review of revised version of Bouhafid & Bonne**

The replies of the authors to my first-round comments are not satisfactory, in particular most major issues have not been addressed properly. Resolved issues are not listed here, only open issues are repeated. The style of this document is as follows: In red, you can see the new comments of this second-round review. The text in black is extracted from the author's reply (it lists the reviewer comment of the first round, the author's answer, and manuscript changes use italics).

The authors should put more effort in addressing the reviewer's comment, otherwise it becomes tedious for the reviewers.

Most figures show quantities that are not really relevant for the later climate impact of contrail-cirrus. Some of the quantities can be shown to explain early contrail processes, but your descriptions tend to say that every effect you see can have a long-lasting impact on the mature contrail.

I understand and acknowledge that it is out of scope to present contrail-cirrus simulations in this study. Yet, the selection of plots and displayed quantities and the interpretation of the results should reflect what is known about the connection between early contrail properties and the radiative impact of the contrail-cirrus over its full life cycle.

GENERAL COMMENTS :

**Reviewer comment:** The analysis of the extinction and optical depth of young contrail may be misleading and misinterpreted by readers. The contribution of the first few minutes of the full contrail lifecycle to the time-integrated radiative forcing (or extinction) is usually not substantial. A relatively larger extinction in the beginning does not imply a larger radiative impact at later times and in total.

**Authors answer:** Thank you for this valuable comment. We agree that the early contrail radiative forcing is not relevant for the climatic impact of a contrail. And that an initial larger extinction does not imply a larger radiative impact several hours later. In order to avoid misinterpretation the following warning has been added line 487:

*Indeed, the differences in extinction observed for the first few minutes may potentially decrease, or even vanish, over longer timescales owing to the effect of atmospheric turbulence and wind shear on the ice crystals spatial distribution. A larger extinction in the beginning does not necessarily imply a larger radiative impact at later times and over the full lifetime of the contrail.*

Thanks for adding these sentences. Nevertheless, I do not think that adding only these few lines is sufficient as you still show all the plots with quantities (for which you write they may not matter).

Previous studies (Unterstrasser & Gierens, 2010b; Lewellen, 2014) clearly demonstrated that the number of early ice crystals has a significant long-lasting impact. Early differences in mass (and as a consequence, also changes of the total extinction that are due to ice mass changes) are not really relevant. Your selection of plots does not reflect this at all. This is why GCM models with a contrail parametrization aim at a refined initialization with advanced estimates of the initial ice crystal number (Bier & Burkhardt, 2019, 2022). Moreover, measurement campaigns aim at deriving apparent ice emission indices to evaluate contrail ice number formation (Märkl et al, 2024; Bräuer et al, 2021).

References doi: 10.1029/2018JD029155, 10.1029/2022JD036677, 10.5194/acp-24-3813-2024, 10.1029/2020GL092166

**Reviewer comment:** Simulations in Unterstrasser & Gierens (2010b) and Lewellen (2014) show that the total number of ice crystals is the most crucial quantity of young contrails that determines the further fate. The early ice crystal mass (and also optical depth and integrated extinction) does not significantly affect the long-term behaviour of the contrail-cirrus transition. Hence, an evaluation of total ice crystal number would be more insightful.

**Authors answer:** We give the evolution of averaged ice crystal number (Np) in the contrail as a function of time (Fig.7, Fig. 15, Fig.20 and Fig.24). We believe this gives an insight on the number of ice crystals in the domain.

The ice crystal number have not been put into context. As mentioned, this quantity is crucial and should compare the survival fraction with previous studies. There are many contrail formation and contrail vortex phase studies to compare with. This would also help to motivate the added value of your approach compared to existing ones.

**Reviewer comment:** Moreover, I strongly suggest to not use the term "radiative forcing of young contrails". First of all, radiative forcing is defined as a radiative imbalance typically evaluated at the top of the atmosphere and this is not what is evaluated in your study. You should make clearer, how to interpret the extinction quantity that you analysed.

**Authors answer**: "Young contrail" has been replaced by "Recently formed contrail" and "Radiative forcing" (RF) by "extinction" except for the reference to Ferreira et al. work where an RF parametrization has been used to estimate RF for a recently formed contrail. However, it is true that we cannot extrapolate the results of our simulations to estimate RF a few hours after the end of the dissipation phase, that is in the diffusion phase.

Indeed, Ferreira et al computed the RF of a 10s-old contrails. But this RF estimate is irrelevant for several reasons and I recommend to not cite it in the way you do it:

- The RF parametrization was never meant to be used for a 10s old contrail as it was done by the cited study.
- The contribution of 10s old contrails to the lifetime-integrated radiative effect is negligible.
- Moreover, changes in the RF at t=10s gives no indication about RF changes at later times nor about the lifetime-integrated radiative effect.

**Reviewer comment:** How robust are the evaluations of Δz and Δy? In Eulerian models, contrails typically do not feature very strong gradients at the boundaries and fade out. Hence, the values you determined may depend on thresholds with which you define a contrail. I believe it would help readers to also show vertical and transvers profiles of ice crystal number and mass. This would allow for a more quantitative comparison compared to Figs. 7, 8 and 12 and also makes clearer how robust the evaluations of Δz and Δy are.

**Authors answer:** Yes, it depends on the treshhold but the goal here was to compare the different initialization strategies results using the same treshhold. Spatially averaged ice crystal numbers field 2D contours have been added to the paper (Fig.12, Fig.13, Fig.21, Fig.23) to have a better understanding of contrail size.

I understand your goal, but your findings may depend on which threshold value you choose. You write in your reply that the quantities of interest depend on the chosen threshold values. Hence, it is important to demonstrate the robustness of your definitions. You should convince the reader that your conclusions do not depend on the choice of threshold.
Many thanks for adding the additional figures. That helps a lot. However, the figures contain a lot of

white areas and you should zoom into the relevant areas. Moreover, clarity could be enhanced by adjusting the colorbar to span from 1e6 to 1e9 and using nicer values on the tick labels! Using one colorbar for all subpanels would be sufficient. Moreover, the figures should be combined into a single one that can then be referenced throughout the text. The identical plot (with caption '2LO initialization' for Nb = 0.012 s-1) appears three times in the manuscript. It is very unusual to show the same plot multiple times in the manuscript.

**Reviewer comment:** How is the boundary of the contrail defined? Why do you choose to apply a weighting by number? Why at all and why not by ice mass e.g.? No spatial distributions of ice crystal number concentration are displayed. Nor the time evolution of total ice crystal number is shown. Hence, it is not transparent what effects the weighting in the averaging procedure does introduce.

**Authors answer:** The contrail is defined by the mesh cells where the ice crystal radius rp is strictly greater than the radius of a dry soot particle rs (27 nm). This is now stated in the paper. Thus, only particles with an ice cap are considered. Those particles are by definition ice crystals. We applied a weighting by number of ice crystals because it adequately represents the influence of each ice crystals on the contrail mean quantities. For X a microphysical quantity, each value X contributes to the average proportionally to how many particles have that value. This is exactly the same as weighting by the number of ice crystals. Such weighting is commonly used in statistical physics. If we weighted by ice mass, we could have situations where a cell have a high ice mass but not that many ice crystals, which would bias the computed mean.

I understand how the weighting is done, however it is still not reasonable for me. You state that the weighting is applied for all quantities depicted in Fig. 7. For me such a weighting does only make sense for quantities that describe the properties of a single particle! Hence, only the activated surface fraction and the mean crystal radius are reasonably defined (panel c-d). Why should you apply a weighting by number when you want to obtain the mean number concentration (panel a). The same is true for the IWC (panel b). It is not a property of a single particle. IWC is defined as a mass per volume. Hence, only a weighting with the volume of your grid cells makes sense.
What is a number-weighted relative humidity (panel e)? What's the physical interpretation of weighting RH with the ice crystal number?

A more general comment:

Your goal is to analyse the radiative effect of a contrail. However, in most figures, you show mean quantities, which are intensive quantities. If you are interested in the total effect, then total quantities should be analysed (i.e. extensive quantities), and not total quantities divided by the contrail volume.

Moreover, mean quantities are hard to interpret as they show the combined effect of a change in the total quantity and in contrail dimensions. And to make matters worse, the latter depends on the choice of a threshold.

As a side comment: For contrails, however, it makes no sense to integrate over all three space dimensions. Clearly, the total quantities scales with the length of the contrail in flight direction. Hence, intensive quantities are typically integrated over the contrail cross-sectional area.

In the RANS domain, you divide the contrail into thin slices of width dx, sum up the quantity of interest in each slice, and then divide by dx. This yields the integrated quantity as a function of downstream distance x with units of m^-1. In the temporal LES, you can integrate over all three dimensions and divide by the length of your domain/contrail.

**Reviewer comment:** Your analysis focuses on intensive mean quantities, which depend on the contrail-cross-section of the contrail. It would be interesting to also see integrated quantities like the total ice crystal number and mass (which do not depend on the definition of the contrail boundary).

**Authors answer:** We believe that the definition of the contrail boundary by rp>rs is valid enough to define the contrail boundary with no ambiguity. As mentioned in the previous answer, in our model if a particle radius is less or equal to its core soot radius, it is not an ice crystal. Therefore averaging on every cell where rp>rs consider all of the ice crystals in the computational domain.

I think, you misunderstood the comment. The comment aimed at raising awareness, that the total effect of a young contrail is better described by extensive quantities, that means quantities that are integrals over the contrail-cross section.

The time series plots in Figs. 7, 14, 16,19, 21, 22,24 all show only intensive quantities. The newly included Fig. 9 is the only figure that shows total quantities. Unfortunately, they are apparently ill-defined as stated further below. Hence, none of the current plots shows total values. Hence, it is nearly impossible to interpret your simulation data in terms on implications on the contrail effect.

SPECIFIC COMMENTS :

The original reviewer comments are in bold.

**- Line 44:The ice crystals do not heat up adiabatically. The surrounding air does so:**

→ adiabatic heating replaced by heating (line 49)

You misunderstood what I meant. The air heats up and leads to an increase of the saturation pressure. Ice crystals may relax to the air parcel's temperature. But the important aspect is the adiabatic heating of the air. (Your original formulation stated that ice crystal heat up adiabatically, but this works only for gases!)

**- Line 52 : I would prefer to reformulate to something like "Contrail evolution is driven or governed by physical processes (not by conditions) which are affected by (conditions like) wind shear, stratification etc.**

→ Modifications implemented accordingly.
Could you list the physical processes instead of just saying 'certain number of physical processes?

**- Eqs. 2, 3 and 6 do not convey a lot of information. With which rates do those conversions occur? Would it be more informative to write the equations for the mass fractions of all or selected quantities? :**

→ The mass production rates are now given in Eq.2 and Eq.4.

"The transition from "free" to "adsorbed" states occurs through the reactions SO3 → SO3,*ads* and H2SO4 → H2SO4,*ads*". Sentences like this do not convey much information. It remains open with which rate the conversions occur.

What I miss, is a list of all prognostic equations; only the one for soot is given in Eq.1. It would help to see the analogous equations for all other species. They are more complicated, as they contain source and sink terms. Hence, it would be good to write them down.

**- Section 3.2 does not explicitly mention how water vapour is initialized. The sentence in line 281 may imply that absolute water vapour mixing ratio is held constant with altitude. Most other studies of the contrail vortex phase kept the relative humidity constant. Could you plot RHi(z) to see how much this quantity changes with altitude? :**

→ Ambient water vapour is initially constant in the computational domain. We decided to keep ambient water vapour constant instead of RH after informal discussions with climate scientist colleagues. ISSR measurements are needed to accurately define RH and water vapour profiles in the tropopause. RH profiles are now given in in Fig. 5 for the two stratification scenarios. With this hypthesis very high values of RH are reached at the bottom of domain in the strong stratification case. However, no ice crystals descend at such altitude. This point has been developed clarified in line 301.

Fig. 5 needs a lot of space. The ratio of information content over space is quite low. Two lines with vertical profiles would be sufficient. Alternatively, you can show the RHi fields at the end of the simulation. Then, the contrail vertical extent would be directly visible.

How is the flight altitude chosen? It appears to be at some value >0?  In Fig. 6 cruise altitude seems to be at z=0. Please clarify.

I do not think that ISSR measurements are needed for your application to accurately define RH and water vapour profiles. The spatial variability in nature is very high. Hence, there is no unique "precise profile". Your profiles should be plausible and not the most extreme examples of what could occur in reality. In this sense, your profile for the strong stratification case is not really appropriately chosen. Supersaturation values below z=-250m are just too high to occur in nature. It is not very comforting to see that one of two meteorological scenarios does not really make sense. I cannot rate how much your results are affected by using such high peak RH values.

**- Fig. 9: the contrail height and width evolution of the RANS case with Nb =0.03 s^-1 looks a bit strange in the sense, that at t=4.5 the height suddenly stops to increase (which might be linked to vortex break up) and width increases. What process leads to such a large change in the width increase?**

→ Height stopping to increase is due to vortex break up which happens sooner for Nb=0,03 s^-1. The width increase is most likely due to the strongly turbulent nature of the secondary wake that will mix the ice crystals with the ambient air way more efficiently. This has been clarified in line 440.

Line 440: "contrail height stops increasing". Contrail height is typically used to describe at which altitude a contrail is located. Better say, that "the contrail vertical extent does not increase anymore".

**- Section 4.1. It would be interesting to also see the time evolution of total ice crystal number Nice and mass Mice and possibly also of the ratio Nice/Ns. Mice and Nice are more straightforward to analyse and interpret than the derived mean radius ~(M ice /N ice ) 1/3 . Computing the mean radius via Mice and  ice is probably better than evaluating rp in each grid cell and do a number-weighted average. How much do the computed values differ between the two formulas? The formula in line 325 might be interpreted in a way that rp depends only on IWC. I would prefer to include  Nice in the formula.**

→ We have added the evolution of total ice mass and total ice crystal number in Fig .9. Concerning the mean ice crystal radius, we are not sure to understand your comment. We believe that knowing the ice crystal radius in each cell of the contrail and doing a weighted average gives a good overview of the ice crystals size in the contrail. rp formula as a function of IWC and N_ice is given in Eq.7.

Point 1

Fig.9: Why does the ice crystal number continuously increase and the ice mass not? I think you misunderstood what I meant with $N_{ice}$. $N_{ice}$ should be obtained by integrating over the cross-sectional area of the contrail at each downstream distance. $N_{ice}$ then gives the ice crystal number per meter of flightpath in units m^-1. $N_{ice}(x)$ gives then the ice crystal number for different downstream distances/contrail ages! I expect that $N_{ice}$ first increases during ice crystal formation, then it may reach a plateau and further downstream it might likely decrease due to sublimation processes. It seems that your $N_{ice}(x)$ are integrals not only over the cross-section but also from zero up to x along flight direction. I do not see what this quantity should tell us.

Point 2:

As mentioned, the weighting does not make sense for several quantities you show. As written, the number weighting makes in theory sense for computing a mean radius. However, this is more complicated than it should. Once you evaluate the total ice mass and number $M_{ice}$ and $N_{ice}$, the mean radius can be derived via $(M_{ice}/N_{ice})^{1/3}$, which is more straightforward than your approach.

**-Line 330: If all soot particles are activated and more ice crystals are present, then they should be smaller not larger if ice mass is similar.**

→ This is true but this effect is not taken into account in our Eulerian model. More precisely, our model considers that soot surface activation is done only with sulfur compounds. Ice production term in Eq.4 is directly proportional to activation fraction. Consequently, the higher the activation fraction, the higher the ice production and the higher the ice crystal radius. This represents a limitation of our model, as it should also consider soot activation caused by the ice cap formed on soot particles, and not only activation due to sulfur particles. This point has now been clarified in line 355.
Unfortunately, I do not understand your argumentation. I think I understood how the activation works. But at a later stage, when ice crystal formation is completed, I do not see why this should lead to larger ice crystals. Assuming the same total amount of water vapor is depleted onto the ice crystals, having more ice crystals should, on average, result in smaller crystals compared to when the same water mass is distributed over fewer crystals. Could you try to explain your reasoning in a different way?

TECHNICAL CORRECTIONS:

- Line 21: "estimation of effective radiative ERF of contrails and other forcing agents."

→ Correction added.
Your corrected sentence is not well-formulated.

"This work enabled the estimation of contrails' Effective Radiative Forcing (ERF) and other forcing agents such as CO2, NO*x*, aerosols, and water vapor.".

What does "other forcing agents such as CO2, NO*x*, aerosols, and water vapor." refer to?

"Estimation of other forcing agents?" (no!)

"Estimation of contrail's ERF of other forcing agents?" (which makes no sense!)

Moreover, Lee did not enable the estimation, they only reviewed, summarized and re-scaled existing studies.

My proposition: "This work provided an estimate of the Effective Radiative Forcing (ERF) of contrails and other forcing agents such as $CO_2$, $NO_x$, aerosols, and water vapor."

**Further comments on newly added text parts:**

Line 168: '…then act as condensation nuclei' makes no sense. When ice crystals are already formed, they cannot act again as nuclei!

Fig.6 needs a lot of space without containing much information. You could cut the blue domain.

Line 350: "changes" instead of "evolves"

422: ice crystal number concentration

Line 580: full RANS initialization

**Summary:**

The revised version and also the author reply revealed many shortcomings of the current study design and data analysis. In my opinion, much more changes are needed than what the authors offered in the first revision round. In particular, the selection of analysed and displayed quantities must be strongly adapted to focus on aspects that really matter for the long-term contrail evolution. And more connections to existing contrail formation and contrail vortex phase studies should be made.

---

## Author Response (AR2)

Thank you for your feedback. It seems that some points from our previous response require clarification to clear up a few misunderstandings. In the attached document, you will find our answers to your latest comments and requests.

First, we would like to stress that the primary objective of this paper is to demonstrate the significant influence of near-field aerodynamics on contrail development in the far field—that is, a few minutes after the initial formation of ice crystals. Our results clearly show that horizontal tailplane (HTP) vortices play a key role in shaping the microphysical properties of contrails. We believe this is an important finding, as much of the existing literature tends to idealize the aerodynamic wake and neglect HTP vortices when initializing contrail far-field simulations. We acknowledge, however, that it remains an open question whether the influence of near-field aerodynamics persists throughout the diffusion regime and whether it significantly affects the radiative forcing of contrails.

The modifications implemented in the new version of the manuscript are in color **blue**.

The style of this document is as follows: In red, you can see the answers to your comments. The text in color black contains the reviewer first comment, the authors first answer and the reviewer second comment.

**GENERAL COMMENTS:**

**Reviewer comment round 1**: The analysis of the extinction and optical depth of young contrail may be misleading and misinterpreted by readers. The contribution of the first few minutes of the full contrail lifecycle to the time-integrated radiative forcing (or extinction) is usually not substantial. A relatively larger extinction in the beginning does not imply a larger radiative impact at later times and in total.

**Authors answer round 1**: Thank you for this valuable comment. We agree that the early contrail radiative forcing is not relevant for the climatic impact of a contrail. And that an initial larger extinction does not imply a larger radiative impact several hours later. In order to avoid misinterpretation, the following warning has been added line 487: Indeed, the differences in extinction observed for the first few minutes may potentially decrease, or even vanish, over longer timescales owing to the effect of atmospheric turbulence and wind shear on the ice crystals spatial distribution. A larger extinction in the beginning does not necessarily imply a larger radiative impact at later times and over the full lifetime of the contrail.

**Reviewer comment round 2**: *Thanks for adding these sentences. Nevertheless, I do not think that adding only these few lines is sufficient as you still show all the plots with quantities (for which you write they may not matter).*

*Previous studies (Unterstrasser & Gierens, 2010b; Lewellen, 2014) clearly demonstrated that the number of early ice crystals has a significant long-lasting impact. Early differences in mass (and as a consequence, also changes of the total extinction that are due to ice mass changes) are not really relevant. Your selection of plots does not reflect this at all. This is why GCM models with a contrail parametrization aim at a refined initialization with advanced estimates of the initial ice crystal number (Bier & Burkhardt, 2019, 2022). Moreover, measurement campaigns aim at deriving apparent ice emission indices to evaluate contrail ice number formation (Märkl et al, 2024; Bräuer et al, 2021).*

*References doi: 10.1029/2018JD029155, 10.1029/2022JD036677, 10.5194/acp-24-3813-2024, 10.1029/2020GL092166*

**Authors answer round 2:** The plots of the optical depth have been removed from the paper. We agree with you concerning the very strong influence of ice crystal number on contrails total extinction. As will be discussed below, we have added in our paper the evolution of total ice crystal number per meter of flightpath. However, we would like to add that for some initial ice crystal number values ice mass is a relevant parameter too. For example, in Unterstrasser and Gierens (https://acp.copernicus.org/articles/10/2037/2010/acp-10-2037-2010.pdf), looking at Fig. 1 for total extinction shows that increasing total ice mass by a factor 2 leads to a 14% increase in total extinction at t=20000 s (compare green solid line with green dot line).  On the other hand, increasing total ice mass by a factor 10 leads this time to a 6% decrease in total extinction (compare brown solid line with brown dot line).

**Reviewer comment round 1**: Simulations in Unterstrasser & Gierens (2010b) and Lewellen (2014) show that the total number of ice crystals is the most crucial quantity of young contrails that determines the further fate. The early ice crystal mass (and also optical depth and integrated extinction) does not significantly affect the long-term behaviour of the contrail-cirrus transition. Hence, an evaluation of total ice crystal number would be more insightful.

**Authors answer round 1**:  We give the evolution of averaged ice crystal number (Np) in the contrail as a function of time (Fig.7, Fig. 15, Fig.20 and Fig.24). We believe this gives an insight on the number of ice crystals in the domain.

*Reviewer comment round 2: The ice crystal number have not been put into context. As mentioned, this quantity is crucial and should compare the survival fraction with previous studies. There are many contrail formation and contrail vortex phase studies to compare with. This would also help to motivate the added value of your approach compared to existing ones.*

**Authors answer round 2:** As mentioned in our previous response, we have now highlighted the importance of the number of early ice crystals for contrail evolution relatively to other parameters (see line 522). Moreover, a direct comparison has been made with Lottermoser and Unterstrasser (https://doi.org/10.5194/acp-25-7903-2025, 2025) where we computed $z_\Delta$ parameter and compared our values with the ones obtained in Lottermoser and Unterstrasser work (see Fig.16). Comparisons with other works from the literature have been made through the paper concerning the values we get for total ice crystal number.

**Reviewer comment round 1:** Moreover, I strongly suggest to not use the term "radiative forcing of young contrails". First of all, radiative forcing is defined as a radiative imbalance typically evaluated at the top of the atmosphere and this is not what is evaluated in your study. You should make clearer, how to interpret the extinction quantity that you analysed.

**Authors answer round 1:** "Young contrail" has been replaced by "Recently formed contrail" and "Radiative forcing" (RF) by "extinction" except for the reference to Ferreira et al. work where an RF parametrization has been used to estimate RF for a recently formed contrail. However, it is true that

we cannot extrapolate the results of our simulations to estimate RF a few hours after the end of the dissipation phase, that is in the diffusion phase.

***Reviewer comment round 2:*** *Indeed, Ferreira et al computed the RF of a 10s-old contrails. But this RF estimate is irrelevant for several reasons and I recommend to not cite it in the way you do it:*

• *The RF parametrization was never meant to be used for a 10s old contrail as it was done by the cited study.*

• *The contribution of 10s old contrails to the lifetime-integrated radiative effect is negligible.*

• *Moreover, changes in the RF at t=10s gives no indication about RF changes at later times nor about the lifetime-integrated radiative effect.*

**Authors answer round 2:** Ferreira citation has been removed from the paper.

**Reviewer comment round 1**: How robust are the evaluations of Δz and Δy? In Eulerian models, contrails typically do not feature very strong gradients at the boundaries and fade out. Hence, the values you determined may depend on thresholds with which you define a contrail. I believe it would help readers to also show vertical and transvers profiles of ice crystal number and mass. This would allow for a more quantitative comparison compared to Figs. 7, 8 and 12 and also makes clearer how robust the evaluations of Δz and Δy are.

**Authors answer round 1**: Yes, it depends on the treshhold but the goal here was to compare the different initialization strategies results using the same treshhold. Spatially averaged ice crystal numbers field 2D contours have been added to the paper (Fig.12, Fig.13, Fig.21, Fig.23) to have a better understanding of contrail size

***Reviewer comment round 2***: *I understand your goal, but your findings may depend on which threshold value you choose. You write in your reply that the quantities of interest depend on the chosen threshold values. Hence, it is important to demonstrate the robustness of your definitions. You should convince the reader that your conclusions do not depend on the choice of threshold. Many thanks for adding the additional figures. That helps a lot. However, the figures contain a lot of white areas and you should zoom into the relevant areas. Moreover, clarity could be enhanced by adjusting the colorbar to span from 1e6 to 1e9 and using nicer values on the tick labels! Using one colorbar for all subpanels would be sufficient. Moreover, the figures should be combined into a single one that can then be referenced throughout the text. The identical plot (with caption '2LO initialization' for Nb = 0.012 s-1) appears three times in the manuscript. It is very unusual to show the same plot multiple times in the manuscript.*

- There has been a misunderstanding from us concerning your comment, misunderstanding which impacted our subsequent answer. By "treshhold", we understood the treshhold concerning the definition of the contrail length and width, not the definition of the contrail itself, that is the cells in the domain where rp>rs.

  We strongly believe that defining the contrail by rp>rs using our specific Eulerian microphysics model is a robust definition of the contrail in the context of our Eulerian

microphysics model. We agree that this is a point that needs clarification. The mathematical definition of rp is:

$$r_p^3 = \frac{3}{4\pi} \frac{\rho Y_{H_2O_s}}{\rho_{sol} N_s} + r_s^3$$

$\rho, N_s, \rho_{sol}, Y_{H_2O_s}, r_s$ are respectively fluid density, soot density number, ice density, ice mass fraction and dry soot particle radius. When rp>rs in a mesh cell, it necessarily means that $Y_{H_2O_s} > 0$ and that the cell contains ice crystals. Thus, that cell is part of the contrail. When rp=rs, $Y_{H_2O_s} = 0$ and the particles contained in the cell are all dry soot particles.
-   We have zoomed on the mentioned figures to have less white. Colorbar on the figures has been fixed to span fro 1e6 to 1e9 with nicer values. Only one colorbar is used on each panel now. The titles of the plots/figures have been modified to differentiate them in the paper.

**Reviewer comment round 1**: How is the boundary of the contrail defined? Why do you choose to apply a weighting by number? Why at all and why not by ice mass e.g.? No spatial distributions of ice crystal number concentration are displayed. Nor the time evolution of total ice crystal number is shown. Hence, it is not transparent what effects the weighting in the averaging procedure does introduce.

**Authors answer round 1**: The contrail is defined by the mesh cells where the ice crystal radius rp is strictly greater than the radius of a dry soot particle rs (27 nm). This is now stated in the paper. Thus, only particles with an ice cap are considered. Those particles are by definition ice crystals. We applied a weighting by number of ice crystals because it adequately represents the influence of each ice crystals on the contrail mean quantities. For X a microphysical quantity, each value X contributes to the average proportionally to how many particles have that value. This is exactly the same as weighting by the number of ice crystals. Such weighting is commonly used in statistical physics. If we weighted by ice mass, we could have situations where a cell have a high ice mass but not that many ice crystals, which would bias the computed mean.

*Reviewer comment round 2*: *I understand how the weighting is done, however it is still not reasonable for me. You state that the weighting is applied for all quantities depicted in Fig. 7. For me such a weighting does only make sense for quantities that describe the properties of a single particle! Hence, only the activated surface fraction and the mean crystal radius are reasonably defined (panel c-d). Why should you apply a weighting by number when you want to obtain the mean number concentration (panel a). The same is true for the IWC (panel b). It is not a property of a single particle. IWC is defined as a mass per volume. Hence, only a weighting with the volume of your grid cells makes sense. What is a number-weighted relative humidity (panel e)? What's the physical interpretation of weighting RH with the ice crystal number?*

**Authors answer round 2**: We agree with your comment. IWC, RH and soot number density are now weighted by volume instead of ice crystal number. Corrections have been implemented in Fig.7 for the near-field RANS simulation. Concerning the temporal LES simulation of the far-field, we replaced the averaged quantities with total quantities as will be discussed below.

***Reviewer standalone comment round 2:*** *A more general comment: Your goal is to analyse the radiative effect of a contrail. However, in most figures, you show mean quantities, which are intensive quantities. If you are interested in the total effect, then total quantities should be analysed (i.e. extensive quantities), and not total quantities divided by the contrail volume. Moreover, mean quantities are hard to interpret as they show the combined effect of a change in the total quantity and in contrail dimensions. And to make matters worse, the latter depends on the choice of a threshold. As a side comment: For contrails, however, it makes no sense to integrate over all three space dimensions. Clearly, the total quantities scales with the length of the contrail in flight direction. Hence, intensive quantities are typically integrated over the contrail cross-sectional area. In the RANS domain, you divide the contrail into thin slices of width dx, sum up the quantity of interest in each slice, and then divide by dx. This yields the integrated quantity as a function of downstream distance x with units of m^-1. In the temporal LES, you can integrate over all three dimensions and divide by the length of your domain/contrail.*

**Authors answer round 2**: We agree with your comment. Average quantities have been replaced by total quantities. For the RANS domain, we plot total ice mass and total ice crystal number per meter of flightpath (See Fig.9). For the temporal LES, we followed your recommendations and we now plot total ice mass per flightpath $m_i$ (kg/m), total ice crystal number per flightpath $N_i$ (1/m) and the mean particle radius defined by: r_p,mean = $\left( \frac{3\, m_i}{4\, \pi\, N_i \rho_{sol}} \right)^{\frac{1}{3}}$.

**Reviewer comment round 1**: Your analysis focuses on intensive mean quantities, which depend on the contrail-cross-section of the contrail. It would be interesting to also see integrated quantities like the total ice crystal number and mass (which do not depend on the definition of the contrail boundary).

**Authors answer round 1:** We believe that the definition of the contrail boundary by rp>rs is valid enough to define the contrail boundary with no ambiguity. As mentioned in the previous answer, in our model if a particle radius is less or equal to its core soot radius, it is not an ice crystal. Therefore averaging on every cell where rp>rs consider all of the ice crystals in the computational domain.

**Reviewer comment round 2**: I think, you misunderstood the comment. The comment aimed at raising awareness, that the total effect of a young contrail is better described by extensive quantities that means quantities that are integrals over the contrail-cross section. The time series plots in Figs. 7, 14, 16,19, 21, 22,24 all show only intensive quantities. The newly included Fig. 9 is the only figure that shows total quantities. Unfortunately, they are apparently illdefined as stated further below. Hence, none of the current plots shows total values. Hence, it is nearly impossible to interpret your simulation data in terms on implications on the contrail effect.

**Authors answer round 2:** See previous answer. Intensive quantities have been replaced by extensive quantities.

The reviewer comments are in bold.

COMMENT 1:

**- Line 44: The ice crystals do not heat up adiabatically. The surrounding air does so:**
**-> *adiabatic heating replaced by heating (line 49)***
**You misunderstood what I meant. The air heats up and leads to an increase of the saturation pressure. Ice crystals may relax to the air parcel's temperature. But the important aspect is the adiabatic heating of the air. (Your original formulation stated that ice crystal heat up adiabatically, but this works only for gases!)**

This has been corrected (see line 54).

COMMENT 2:

**- I would prefer to formulate to something like "Contrail evolution is driven or governed by physical processes (not by conditions) which are affected by (conditions like) wind shear, stratification etc.**
**-> *Modifications implemented accordingly.***
**Could you list the physical processes instead of just saying 'certain number of physical processes'?**
The physical processes have been listed (line 64)

COMMENT 3:

**- Eqs. 2, 3 and 6 do not convey a lot of information. With which rates do those conversions occur? Would it be more informative to write the equations for the mass fractions of all or selected quantities? :**
**--> *The mass production rates are now given in Eq.2 and Eq.4***

**The transition from free state to adsorbed states occur through the reactions $SO_3 \rightarrow SO_3$,*ads* and $H_2SO_4 \rightarrow H_2SO_4$,*ads*. Sentences like this do not convey much information. It remains open with which rate the conversions occur.**
**What I miss, is a list of all prognostic equations; only the one for soot is given in Eq.1. It would help to see the analogous equations for all other species. They are more complicated, as they contain source and sink terms. Hence, it would be good to write them down.**

The full RANS and LES equations for a multi-species gas mixture and solved in our simulations are now given in Section 2.1.1 and 2.1.2. We give the general transport equation for the mass fraction of each species in the gas mixture, including ice water. The transport equation includes source term. The source term for adsorption is given in Eq.9.

COMMENT 4:

**Section 3.2 does not explicitly mention how water vapour is initialized. The sentence in line 281 may imply that absolute water vapour mixing ratio is held constant with altitude. Most other studies of the contrail vortex phase kept the relative humidity constant. Could you plot RHi(z) to see how much this quantity changes with altitude?**

*-> Ambient water vapour is initially constant in the computational domain. We decided to keep ambient water vapour constant instead of RH after informal discussions with climate scientist colleagues. ISSR measurements are needed to accurately define RH and water vapour profiles in the tropopause. RH profiles are now given in in Fig. 5 for the two stratification scenarios. With this hyphthesis very high values of RH are reached at the bottom of domain in the strong stratification case. However, no ice crystals descend at such altitude. This point has been developed clarified in line 301.*

*Fig. 5 needs a lot of space. The ratio of information content over space is quite low. Two lines with vertical profiles would be sufficient. Alternatively, you can show the RHi fields at the end of the simulation. Then, the contrail vertical extent would be directly visible.*

**How is the flight altitude chosen? It appears to be at some value >0? In Fig. 6 cruise altitude seems to be at z=0. Please clarify.**

**I do not think that ISSR measurements are needed for your application to accurately define RH and water vapour profiles. The spatial variability in nature is very high. Hence, there is no unique "precise profile". Your profiles should be plausible and not the most extreme examples of what could occur in reality. In this sense, your profile for the strong stratification case is not really appropriately chosen. Supersaturation values below z=-250m are just too high to occur in nature. It is not very comforting to see that one of two meteorological scenarios does not really make sense. I cannot rate how much your results are affected by using such high peak RH values.**

Fig. 5 has now been changed to two lines with vertical profiles of RH.  We have also added the RH contour inside the plume at the start of the LES simulation to help the reader visualize the initial position of the plume and RH values in the plume at the start of the simulation.

We recognize that we have not been clear enough for the relative humidity profile. First, it is important to differentiate plume initial altitude and flight altitude. For the LES simulation, the plume initial altitude is chosen in the plume area in the LES domain. For the RANS simulation, the flight altitude corresponds the z-coordinate of the wing trailing edge. For both cases, this corresponds to an altitude z ~ 0 in the corresponding reference frame even though it is not exactly the same value of z between the RANS reference frame and the LES reference frame. This is not contradictory because the LES reference frame and the RANS reference frame are different.  In the RANS domain, computations are carried out in the aircraft reference frame: the aircraft is horizontal, and the incoming airflow enters the domain with an angle equal to the angle of attack (=3°). In contrast, the LES domain is defined in the ground reference frame, where the aircraft is tilted by 3°. This is explained more in detail in Bouhafid et al. 2024 https://doi.org/10.1016/j.ast.2024.109512.

Regarding the plausibility of the strong stratification scenario, we first note that the elevated humidity levels below z = -300 m are not accessed by the contrail when using RANS initialization (see Fig. 13). This occurs due to the pronounced stratification and its stabilizing effect on the wake system. Under the more stable vortex pair conditions achieved with 2LO initialization, the contrail reaches levels below z = -250 m but remains above z = -300 m, where RHi approaches approximately 165%—a value that appears within the range of atmospheric observations in the North Atlantic flight corridor (see Ovarlez et al. https://doi.org/10.1029/1999JD900954). This work shows that a RHi value of 165%, while uncommon, is not exceptional in cloudy environment (see Plate 2 and Fig 1 of

https://doi.org/10.1029/1999JD900954). Thus, we believe that the zones in the LES domain where RHi is extraordinarily high (RHi>200%) do not influence our results as they are not reached by the contrail.

Finally, investigating scenarios with higher than usual RHi holds significant scientific value due to the complex physics involved and the quantitative insights into how stratification influences contrail evolution when its influence becomes dominant in controlling contrail dynamic. This could be seen as an interesting academic exercise.

COMMENT 5:

**- Fig. 9: the contrail height and width evolution of the RANS case with Nb =0.03 s^-1 looks a bit strange in the sense, that at t=4.5 the height suddenly stops to increase (which might be linked to vortex break up) and width increases. What process leads to such a large change in the width increase?**
*->The width increase is most likely due to the strongly turbulent nature of the secondary wake that will mix the ice crystals with the ambient air way more efficiently. This has been clarified in line 440.*
**Line 440: "contrail height stops increasing". Contrail height is typically used to describe at which altitude a contrail is located. Better say, that "the contrail vertical extent does not increase anymore".**

"Contrail height" has been replaced by "contrail vertical extent." for every occurrence in the manuscript.

COMMENT 6:

**- Section 4.1. It would be interesting to also see the time evolution of total ice crystal number Nice and mass Mice and possibly also of the ratio Nice/Ns. Mice and Nice are more straightforward to analyse and interpret than the derived mean radius ~(M ice /N ice ) 1/3 . Computing the mean radius via Mice and ice is probably better than evaluating rp in each grid cell and do a number-weighted average. How much do the computed values differ between the two formulas? The formula in line 325 might be interpreted in a way that rp depends only on IWC. I would prefer to include Nice in the formula**

*-> We have added the evolution of total ice mass and total ice crystal number in Fig. 9. Concerning the mean ice crystal radius, we are not sure to understand your comment. We believe that knowing the ice crystal radius in each cell of the contrail and doing a weighted average gives a good overview of the ice crystals size in the contrail. rp formula as a function of IWC and N_ice is given in Eq.7.*

**Point 1**
**Fig.9: Why does the ice crystal number continuously increase and the ice mass not? I think you misunderstood what I meant with Nice. Nice should be obtained by integrating over the cross-sectional area of the contrail at each downstream distance. Nice then gives the ice crystal number per meter of flightpath in units m^-1. Nice(x) gives then the ice crystal number for different downstream distances/contrail ages! I expect that Nice first increases during ice crystal formation, then it may reach a plateau and further downstream it might likely decrease due to sublimation processes. It seems that your Nice(x) are integrals not only over the cross-section but also from zero up to x along flight direction. I do not see what this quantity should tell us.**

**Point 2:**
**As mentioned, the weighting does not make sense for several quantities you show. As written, the number weighting makes in theory sense for computing a mean radius. However, this is more complicated than it should. Once you evaluate the total ice mass and number Mice and Nice, the mean radius can be derived via (Mice/Nice)1/3, which is more straightforward than your approach.**

Nice(x) is now defined for the RANS calculation as the integral over the cross-section at different downstream position. Nice(x) quickly reaches a plateau (around x/b=6). Moreover, we now use your definition of the mean radius for the LES simulations.

COMMENT 7:

**-Line 330: If all soot particles are activated and more ice crystals are present, then they should be smaller not larger if ice mass is similar.**
*-> This is true but this effect is not considered in our Eulerian model. More precisely, our model considers that soot surface activation is done only with sulfur compounds. Ice production term in Eq.4 is directly proportional to activation fraction. Consequently, the higher the activation fraction, the higher the ice production and the higher the ice crystal radius. This represents a limitation of our model, as it should also consider soot activation caused by the ice cap formed on soot particles, and not only activation due to sulfur particles. This point has now been clarified in line 355.*
**Unfortunately, I do not understand your argumentation. I think I understood how the activation works. But at a later stage, when ice crystal formation is completed, I do not see why this should lead to larger ice crystals. Assuming the same total amount of water vapor is depleted onto the ice crystals, having more ice crystals should, on average, result in smaller crystals compared to when the same water mass is distributed over fewer crystals. Could you try to explain your reasoning in a different way?**

Let us assume that the same total amount of water is depleted onto the ice crystals. In our model, ice will form on the portion of the soot surface that is activated. This portion of the soot surface is proportional to the activation fraction $\theta$. Mathematically, this implies that ice production rate is proportional to $\theta$ regardless of the ice crystal size. The lower $\theta$ is, the less ice is formed on particles on a given time interval and the slower ice crystals will grow. This results in smaller ice crystals. For the same total amount of water and same ambient relative humidity, ice crystals will be smaller in the case $\theta < 1$ than in the case $\theta = 1$. The relevant parameter here is the rate at which water vapor converts to ice in addition of total amount of water in the plume.

To put it shortly, the answer to your question lies in the fact that even when ice crystals are formed, the activated surface fraction value does not depend on ice crystal size. This is a strong hypothesis that leads to underestimating ice crystals size. This behavior is erroneous and constitutes one of the limits of our microphysical model. In reality, when ice crystals are formed and large enough, $\theta$ should be set to 1 to translate that activation is not longer important for ice crystals growth.

TECHNICAL CORRECTIONS:

**- Line 21: estimation of effective radiative ERF of contrails and other forcing agents.**
*-> Correction added.*
**Your corrected sentence is not well-formulated. "This work enabled the estimation of contrails' Effective Radiative Forcing (ERF) and other forcing agents such as CO2, NOx, aerosols and water vapor".**
**What does other "forcing agents such as CO2, NOx, aerosols and water vapor" refer to?**

**"Estimation of other forcing agents?" (no!)**
**"Estimation of contrail's ERF of other forcing agents?" (no!)**
**Moreover, Lee did not enable the estimation, they only reviewed, summarized and re-scaled existing studies.**
**My proposition: "This work provided an estimate of the Effective Radiative Forcing (ERF) of contrails and other forcing agents such as CO2, NOx, aerosols, and water vapor."**
Your proposition has been implemented. Thank you.

*Further comments on newly added text parts:*
**Line 68: '…then act as condensation nuclei' makes no sense. When ice crystals are already formed, they cannot act again as nuclei!**
You are right, this has been removed from the article.

**Fig.6 needs a lot of space without containing much information. You could cut the blue domain.**
The figure has been cropped.
**Line 350: changes instead of evolves**
Implemented to the paper.
**422: ice crystal number concentration**
Implemented to the paper.

**Line 580: full RANS initialization**
Implemented to the paper.